# Benchmarking Single-Modal Molecular Representations Across Diverse Modalities

## Abstract

Molecular representation learning (MRL) plays a vital role in high-precision drug discovery. Currently, people represent molecules in different modalities (such as sequences, graphs, and images), and have developed many MRL methods. However, three key challenges hinder further progress in the field of MRL: (i) Lack of systematic and unified evaluation on models of different modalities, resulting in unfair comparisons or being affected by randomness; (ii) The specific advantages between different molecular modalities are unclear; (iii) Lacking a unified platform to integrate data of different modalities and a large number of MRL methods. Therefore, we propose the first MRL platform supporting different modalities, called BenchMol, to integrate a large number of sing-modal MRL methods with different modalities and evaluate them systematically and fairly. BenchMol has four attractive features: (i) Rich modalities: BenchMol supports 7 major modalities of molecules, such as fingerprint, sequence, graph, geometry, image, geometry image, and video; (ii) Comprehensive methods: BenchMol integrates 23 mainstream MRL methods to process these modalities; (iii) New benchmarks: BenchMol constructs two new benchmarks based on PCQM4Mv2 and ChEMBL 34, called MBANet and StructNet, for a more systematic evaluation. (iv) Comprehensive evaluation: evaluation covers different aspects of molecules, such as basic attributes and molecular types. Through BenchMol, we conduct large-scale research on methods of different modalities and report many insightful findings. We hope that BenchMol can help researchers quickly use MRL methods with different modalities on the one hand; and on the other hand, provide meaningful insights into multi-modal MRL and help researchers choose appropriate representations in downstream tasks. We open-sourced BenchMol in Github.

## 1 Introduction

Molecular representation learning (MRL) is a prerequisite for high-precision drug discovery (Li et al., 2022; Catacutan et al., 2024). With the development of deep learning, researchers have developed a large number of MRL methods in recent years (Yi et al., 2022). According to the different representation forms of molecules, existing methods represent molecules in 7 different modalities (as shown in Figure 1) and use modality-specific techniques to extract molecular representations, namely molecular sequence (Kim et al., 2021; Ross et al., 2022), graph (Hu et al., 2020a; Liu et al., 2022a), geometry (Fuchs et al., 2020; Satorras et al., 2021), image (Xiang et al., 2023), geometry image (Xiang et al., 2024a) and video (Xiang et al., 2024b).

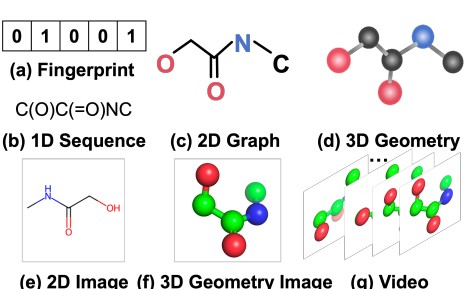

Figure 1: Schematic diagram of "C(O)C(=0)NC" with different modalities. The molecule is represented in 7 different modalities: (a) Fingerprint, (b) Sequence, (c) Graph, (d) Geometry graph, (e) 2D image, (f) 3D geometry image, and (g) video.

*Challenges.* Despite the remarkable success of MRL, there are still three key challenges in its development that hinder the further development of multi-modal MRL methods:

I. **Unfair comparison.** Differences in evaluation strategies lead to incomparable or unfair comparisons between methods. We summarize four differences in the evaluation process, including the method for dividing the dataset (Rong et al., 2020; Ross et al., 2022), the range of parameter optimization (Hu et al., 2020a; Wang et al., 2022), the standardization of labels (Ross et al., 2022; Zhou et al., 2023), and the selection of random seed (Zeng et al., 2022; Wang et al., 2022; Xia et al., 2023). For example, MoLFormer (Ross et al., 2022) uses random scaffold split to divide the dataset and is compared with MolCLR (Wang et al., 2022) and GraphMVP-C (Liu et al., 2022a) which use strict scaffold split. In general, it is easier to achieve good performance on datasets with random scaffold split than on datasets with scaffold split. Compared to previous studies (Hu et al., 2020a) that use the same hyperparameters for different tasks and run 10 replicates, MolCLR performs independent hyperparameter optimization for different tasks and runs 3 replicates. BARTSmiles (Chilingaryan et al., 2022), Uni-Mol (Zhou et al., 2023) and MoLFormer use label regularization to compare with other methods. Hu et al. and Xia et al. use consistent random seeds from 0 to 9 to initialize the model while a large number of studies (Rong et al., 2020; Zeng et al., 2022; Wang et al., 2022) do not explicitly state the random seeds, which may result in a benchmark deviation. Therefore, *it is necessary to build a platform to fairly evaluate methods with different modalities*, which will pave the way for researchers to explore scientific questions instead of falling into biases caused by experimental differences.

II. **Incomplete evaluation for different modality data.** MRL is evolving towards a multi-modal direction, relying on various technology stacks, such as sequence modalities based on Natural Language Processing (NLP) (Devlin et al., 2019; Lewis et al., 2020), graph modality based on graph deep learning (Xu et al., 2018), geometry modality based on geometry deep learning (Monti et al., 2017; Atz et al., 2021), and image, geometry image and video modalities based on Computer Vision (CV) (He et al., 2020; Kirillov et al., 2023). Currently, a large number of methods based on different modalities have been developed in the field of MRL (Hu et al., 2020a; Ross et al., 2022; Zeng et al., 2022). Intuitively, the data and encoding methods between different modalities are different, which may lead to their preference for different types of molecules. However, *this preference is still unclear and deserves further exploration*. In addition, researchers often focus on the task of molecular property prediction from MoleculeNet Wu et al. (2018) in the evaluation of MRL (Xiang et al., 2023; Xia et al., 2023; Xiang et al., 2024a). However, *a single evaluation is not comprehensive for studying preferences in molecular representation of different modalities*. Here, We introduce the Molecular Basic Attribute (MBANet) benchmark built from IEM (Xiang et al., 2024a) and the StructNet benchmark built from the ChemBL 34 database (Zdrazil et al., 2024), as shown in Figure 2, to further evaluate the ability of these models to identify essential molecular attributes and mine information from different types of molecules.

III. **Lack of a unified platform supporting diverse modalities.** MRL methods of different modalities are scattered in various corners of the Internet with different development environments and different running pipelines. *It is challenging to integrate various methods with different modalities into a unified platform to support multiple modality data*. Currently, there remains a blank in the molecular-based platform supporting diverse modalities. We hope to propose a unified molecular platform to provide meaningful insights for multi-modal MRL and facilitate the use and development of multi-modal molecules by researchers.

_Contributions._ In this work, we aim to provide the first MRL platform unified diverse modalities (called BenchMol) covering a large number of existing algorithms and re-evaluate existing methods in a fair and comprehensive manner. Our contributions are summary as follows:

- **Unified and flexible platform for supporting different molecular modalities.** BenchMol is a flexible and easy-to-use toolkit, which integrates 7 molecular modalities into a unified framework. Meanwhile, BenchMol provides a complete pipeline from raw data to the evaluation of the final model, which includes data preprocessing (57 modality extractors), predefined models (6 sequence models, 13 graph models, 9 geometry models and at least 900 visual models), training strategies (linear probing and fine-tuning) and a large number of evaluation metrics.

- **Novel benchmarks.** We propose two benchmarks, MBANet and StructNet, to explore the advantages of existing models in basic molecular information (12 atoms, 4 bonds and 8 attributes) and

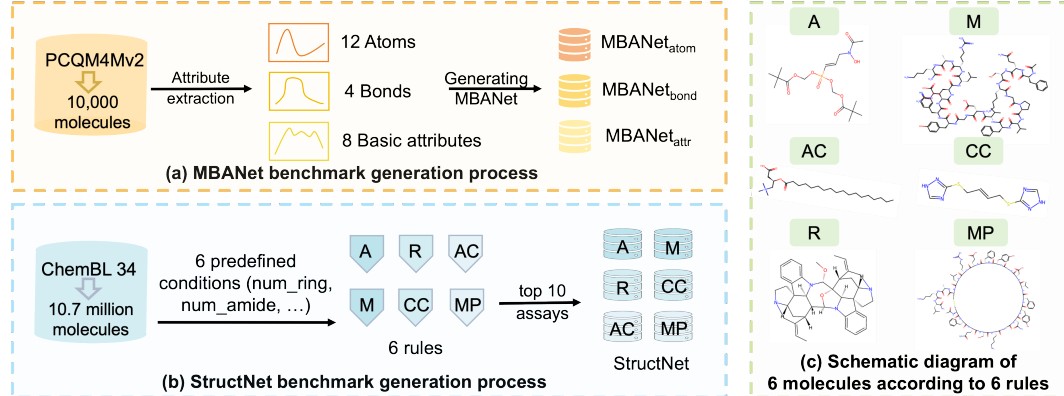

Figure 2: (a) Schematic diagram of MBANet. (b) Schematic diagram of StructNet. 6 conditions are num_ring, average_degree, num_amide, has_branch, num_amide and molecular weight. 6 rules are acyclic (A) rule, complete chain (CC) rule, acyclic chain (AC) rule, macrocyclic peptide (MP) rule, macromolecule (M) rule and reticular (R) rule. (c) 6 types of molecules in StructNet.

the preferences for 6 different types of molecules (acyclic chain, acyclic, complete chain, macro, macrocyclic peptide, reticular molecules), respectively.

- **Experiment comprehensively and fairly.** With BenchMol, we train at least 57,060 models, including 6,960 models on linear probing of 12 MoleculeNet, 900 models on fine-tuning of 3 datasets from MBANet, and 49,200 models on fine-tuning of 60 datasets from StructNet.

- **Meaningful insights.** Based on extensive experiments and rigorous comparisons, we provide many meaningful insights, including 9 main finds. Here, we highlight several important conclusions: (1) In tasks related to molecular properties, fingerprint or sequence modalities tend to be selected in non-pretrained models, while geometry graph modalities tend to be selected in pretrained models; (2) Video modality excels at atomic-level and attribute-level tasks and graph/geometry modality excels at bond-level tasks; (3) With regard to molecular preferences, the geometry modality prefers acyclic molecules, the fingerprint and graph modalities prefer cyclic molecules, and the vision-based modalities prefer macrocyclic and reticular molecules.

## 2 RELATED WORKS

**Molecular representation learning (MRL).** Existing MRL can be mainly divided into the following categories, including sequences, 2D topological graphs, 3D geometric graphs, 2D images, 3D geometry images, and videos. Sequence-based methods represent molecules as 1-dimensional strings (such as SMILES) and use NLP-related techniques to learn molecular representations (Ross et al., 2022; Zheng & Tomiura, 2024). Graph-based methods treat the atoms and bonds of molecules as nodes and edges in a graph and use GNN and its variants to extract features (Hu et al., 2020a; Wang et al., 2022). Image-based methods treat molecules as a flat image and use computer vision (CV)-related techniques for processing (Zeng et al., 2022; Xiang et al., 2023; Zhang et al., 2023). Subsequently, considering the importance of geometric information in molecules, geometric deep learning methods represent molecules as geometric graphs and extract information from them (Schütt et al., 2021; Liu et al., 2022b). Meanwhile, image-based methods render the geometric information of molecules into geometry images and perform feature extraction (Xiang et al., 2024a). Recently, video-based methods have been proposed, which represent the conformation of molecules as a video and extract features (Xiang et al., 2024b). Given that MRL is rapidly evolving towards different modalities, it is necessary to aggregate these methods with different modalities into a unified platform and accelerate the development and use of researchers.

**Molecular benchmark platforms.** OGB (Hu et al., 2020b) proposes a set of diverse, challenging and realistic benchmark datasets covering molecular graphs, which mainly focus on graph data. Molecule3D (Xu et al., 2021) develops a benchmark that includes a dataset with precise ground-state geometries of approximately 4 million molecules, and provides a few baseline methods based on DeeperGCN (Li et al., 2023) and DAGNN (Yang et al., 2021). Deng et al. (2023) focuses on the evaluation of molecular fingerprints and 2D graphs. MOLGRAPHEVAL (Wang et al., 2024) focuses on evaluating the impact of different pre-training strategies based on 2D graphs. Geom3D

(Liu et al., 2024) focuses on geometric data of different biological entities (such as molecules, proteins, crystalline materials). However, with the increase in modality data, the integration of multiple fields has raised concerns about fairness and comprehensiveness in evaluation protocols. Different from the above mentioned methods, BenchMol is the first molecular benchmark platform unified multiple molecular modalities, which integrates 7 different modalities (fingerprint, sequence, graph, geometry, image, geometry image, and video) and provides a easy-to-use interface for access.

## 3 PRELIMINARIES

There is a molecule $m$ with $n_a$ atoms and $n_b$ bonds and the representation of different modalities is as follows:

**Fingerprint.** Molecular Fingerprints $\mathcal{F}$ are a compact, fixed-size representation, where $\mathcal{F} \in \mathbb{R}^{n_{fp}}$ and $n_{fp}$ represents the dimension of molecular fingerprint. Currently, there are many fingerprints developed (Mason et al., 2001), such as 167-dimensional MACCS (Molecular ACCess System) (Durant et al., 2002), ECFP$x$ with custom dimension (Rogers & Hahn, 2010), 210-dimensional RDKit2D (Landrum et al., 2016). You can read (Hou et al., 2024) for more details about fingerprints.

**Sequence.** A molecule is regarded as a string sequence, such as SMILES (Weininger et al., 1988), SELFIES (Krenn et al., 2020) and IUPAC (Kuhn et al., 2004). These sequences are split into tokens by a tokenizer with word segmentation rules, which is formalized as $\mathcal{S} = \{s_0, s_1, ..., s_{n_a}\}$, where $s_{n_a}$ represents a token. Currently, the most commonly used molecular sequence is SMILES and a large number of technologies (Kim et al., 2021; Ross et al., 2022) are developed based on it. In this paper, we focus on the study of molecular SMILES because of its popularity.

**Graph.** A molecular graph $\mathcal{G} = (\mathcal{V}, \mathcal{E})$ consists of a set of nodes $\mathcal{V} \in \mathbb{R}^{n_a \times n_f}$ and edges $\mathcal{E} \in \mathbb{R}^{n_b}$, where $n_f$ represents the feature number of atom (e.g., atom type) (Hu et al., 2020a; Xia et al., 2023). Assume that there are two atoms $v$ and $u$ in $\mathcal{V}$, $\mathcal{E}$ represents an adjacency matrix $\boldsymbol{A} \in \mathbb{R}^{n_a \times n_a}$ that indicates whether $v$ and $u$ are connected, where $\boldsymbol{A}[v, u] = 0$ means there is no edge (bond) otherwise it means there is an edge (bond). In practical applications, since bonds have multiple chemical properties (e.g., bond types), the adjacency matrix $\boldsymbol{A}$ can be easily extended to $\boldsymbol{A}^* \in \mathbb{R}^{n_a \times n_a \times d_b}$, where $d_b$ represents the number of chemical properties of the bond (Liu et al., 2022a; Wang et al., 2022). For example, the type of the bond formed by nodes $v$ and $u$ can be formalized as $\boldsymbol{A}^*[v, u, i_t] = \{0, 1, 2, 3, 4\}$, where $i_t$ represents the index describing the bond type, 0 represents no bond, 1 represents a single bond, 2 represents an aromatic bond, and so on.

**Geometry.** Geometry graph introduces the 3-dimensional coordinates of atoms based on graph (Zhou et al., 2023; Satorras et al., 2021), which is formalized as $\hat{\mathcal{G}} = (\hat{\mathcal{V}}, \hat{\mathcal{E}})$, where $\hat{\mathcal{V}} \in \mathbb{R}^{n_a, n_f+3}$. Please note that in practical applications, models do not include edge information or use fully connected adjacency matrices to represent edge information when processing geometric graphs.

**Image, Geometry Image and Video.** Molecular images (Xiang et al., 2023), geometry images Xiang et al. (2023) and videos (Xiang et al., 2024b) are based on visual representations of molecules, which are atom- and bond-independent and are made up of a bunch of pixels. We describe the importance of the visual modality in Appendix C.3. Using RDKit (Landrum, 2013), the SMILES of a molecule can be converted into a molecular image $\mathcal{U}^I \in \mathbb{R}^{224 \times 224 \times 3}$ (Figure 1(e)). The geometry image (Figure 1(f)) and video (Figure 1(g)) of the molecule take into account the 3D structural information of the molecule and are generated using PyMOL (DeLano et al., 2002). See Appendix B for details of visual rendering. Formally, the geometry image and video can be represented as $\mathcal{U}^G \in \mathbb{R}^{4 \times 224 \times 24 \times \times 3}$ and $\mathcal{U}^V \in \mathbb{R}^{60 \times 224 \times 24 \times \times 3}$, respectively, where 4 and 60 represent the number of views in the geometry image and the number of frames in the video.

## 4 BENCHMOL: BENCHMARK PLATFORM WITH DIVERSE MODALITIES

### 4.1 OVERVIEW OF BENCHMOL

As shown in Figure 3, BenchMol consists of 5 modules. The *Dataset Collector Module* provides 3 types of benchmarks (Section 4.2). The *Modality Extractor Module* is used for data preprocessing, which converts raw data into input for models of specific modalities (4.3). The *Model Initializer Module* is used to initialize a large number of models (Section 4.4). The *Training Strategy Module*

is used to provide model trainers, including linear probing and fine-tuning (Section 4.5). The *Evaluation Metric Module* provides the indicators required for classification and regression tasks (Section 4.6). For a discussion of the motivation and potential impact of BenchMol, see Appendix C.1.

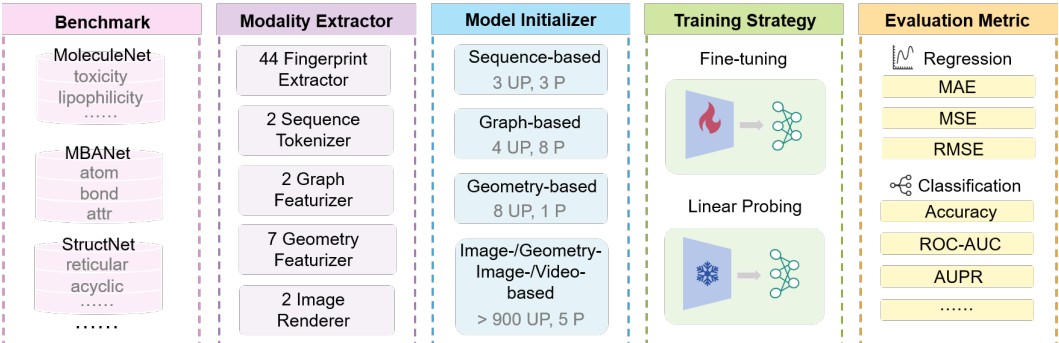

Figure 3: Overview of the proposed BenchMol.

## 4.2 DATASET COLLECTOR MODULE

In BenchMol, we provide three benchmarks, MoleculeNet (Wu et al., 2018), MBANet and StructNet. We contribute MBANet and StructNet to systematically analyze the preferences and performance of different modal methods. See Appendix C.2 for details of the motivation and practicality of the benchmarks. Next, we introduce the construction process of MBANet and StructNet.

**MBANet.** MBANet aims to study the ability of different methods to capture molecular basic information, including atom distributions, bond distributions, and basic attributes. As shown in Figure 2(a), we first sample 10,000 molecules from PCQM4Mv2 (Hu et al., 2017). Then, we count the number of atoms and bonds in each molecule, which are formalized as $\mathcal{K}^a = \{k_1^a, k_2^a, ..., k_{12}^a\}$ and $\mathcal{K}^b = \{k_1^b, k_2^b, k_3^b, k_4^b\}$, respectively, where $\mathcal{K}^a \in \mathbb{Z}^{12}$ and $\mathcal{K}^b \in \mathbb{Z}^4$ represent the count of 12 types of atoms (C, N, O, F, S, Cl, Br, P, Si, B, Se, Ge) and 4 types of bonds (SINGLE, AROMATIC, DOUBLE, TRIPLE). Subsequently, we further extracted 8 basic attributes, including {molecular weight, MolLogP, MolMR, BalabanJ, NumHAcceptors, NumHDonors, NumValenceElectrons, TPSA}, which are formalized as $\mathcal{K}^k \in \mathbb{R}^8$. Finally, we can generate three different types of datasets: MBANet$_{atom} = \{m, \mathcal{K}^a\}$, MBANet$_{bond} = \{m, \mathcal{K}^b\}$, and MBANet$_{attr} = \{m, \mathcal{K}^k\}$, where $m$ represents molecules. See Appendix D.1 for details and limitation analysis of the MBANet.

**StructNet.** StructNet is designed to evaluate the preference of models with varying modalities for molecule types. As shown in Figure 2(b), we first collect over 14.4 million the latest molecules from ChemBL 34. Then, we predefine 6 different conditions, including the number of rings, average degree, presence of branches, maximum number of rings, number of amides, and molecular weight. Based on the molecular SMILES sequences, we leverage RDKit to generate these conditions. Then, we formulate 6 rules to classify molecules into different types: acyclic, complete chain, acyclic chain, macrocyclic peptide, reticular, and macromolecule. As shown in Figure 2(c), the application of these rules enables us to identify molecules exhibiting diverse characteristics. For example, a molecule generated by the acyclic chain rule is a chain-like molecule without any rings. Ultimately, we group the molecules based on the Assay ChEMBL ID defined in ChemBL 34 and select the top 10 assays with the largest number of samples to construct StructNet. It is worth noting that these result in the creation of 60 datasets within StructNet, with 10 for each molecule type. See Appendix D.2 for details of the StructNet.

## 4.3 MODALITY EXTRACTOR MODULE

In order to obtain mode-specific input from the original molecular SMILES data, we define modality extractors as follows:

- **44 fingerprint extractors.** We predefine 6 types of 44 common molecular fingerprints in BenchMol, including circular- (ECFP$x$, FCFP$x$), path- (RDK$x$, HashTT), substructure- (MACCS),

longer-version-, pharmacophore- (TPATF), and physicochemistry-based (RDKit descriptors) fingerprints.

- **2 types of sequence tokenizers.** We include 2 common tokenizers for processing molecular SMILES, which come from CHEM-BERT (Kim et al., 2021) and MoLFormer.

- **2 types of graph featurizers.** We incorporate 2 common graph featurizers, which come from Hu et al. (2020a) and OGB library (Hu et al., 2020b).

- **7 types of geometry featurizers.** We integrate 7 geometry graph construction methods based on Hu et al. (2020a), OGB library, Geom3D, and Uni-Mol, which covers the input formats required by existing geometric deep learning methods. In particular, for Geom3D, we generate 4 combinations of featurizers by pairwise combining whether to use fully connected edges and whether to use only node features.

- **2 types of image renderers.** We build two visualization renderers based on RDKit and PyMOL to generate 2D images, 3D geometry images, and videos, which are referenced by ImageMol, IEM, and VideoMol, respectively.

## 4.4 MODEL INITIALIZER MODULE

After obtaining data of different modalities, we define several modality-specific factory to initialize model for extracting features. Table 1 shows the models based on different modalities supported in BenchMol. BenchMol not only supports existing pre-trained models, but also includes a large number of non-pre-trained basic models. Especially for vision-based models, BenchMol is compatible with the timm library (Wightman, 2019) and supports more than 900 vision models.

Table 1: The supported models in BenchMol.

| Modality Type | Pre-training Models | Models w/o Pre-training |
|---|---|---|
| **1D Sequence** | CHEM-BERT (Kim et al., 2021), CHEM-RoBERTa (Kim et al., 2021), MoLFormer (Ross et al., 2022) | BERT (Devlin, 2018), RoBERTa (Liu, 2019), Transformer_Rotate (Ross et al., 2022) |
| **2D Graph** | EdgePred (Hu et al., 2020a), ContextPred (Hu et al., 2020a), infomax (Hu et al., 2020a), masking (Hu et al., 2020a), GraphMVP (Liu et al., 2022a), MolCLR (Wang et al., 2022), CGIP-Graph (Xiang et al., 2023), MoleBERT (Xia et al., 2023) | GIN (Xu et al., 2018), GAT (Veličković et al., 2018), GCN (Li et al., 2021), GraphSAGE (Hamilton et al., 2017), DeeperGCN (Kipf & Welling, 2016) |
| **3D Geometry Graph** | Uni-Mol (Zhou et al., 2023) | SchNet (Schütt et al., 2017), DimeNet (Gasteiger et al.,), DimeNetPlusPlus (Gasteiger et al., 2020), TFN (Thomas et al., 2018), SE3_Transformer (Fuchs et al., 2020), EGNN (Satorras et al., 2021), SphereNet (Coors et al., 2018), PaiNN (Schütt et al., 2021) |
| **2D Image** | ImageMol (Zeng et al., 2022), CGIP-Image (Xiang et al., 2023), MaskMol (cheng et al., 2024) | |
| **3D Geometry Image** | IEM (Xiang et al., 2024a) | More than 900 vision models based on timm (Wightman, 2019) |
| **Video** | VideoMol (Xiang et al., 2024b) | |

## 4.5 TRAINING STRATEGY MODULE

In training strategy module, we provide two training strategies, linear probing and fine-tuning. In linear probing, to improve efficiency, we define a modality-specific feature extractor and pre-extract features based on a given pre-trained model. Then, we define a trainer to train a fully connected layer directly on the features. In fine-tuning, given a model name, BenchMol will train the entire model on the given task.

## 4.6 EVALUATION METRIC MODULE

BenchMol supports multiple evaluation metrics for classification and regression tasks. For classification tasks, the metrics include Accuracy, Area Under Receiver Operating Characteristic Curve

(ROC-AUC), F1-Score, Area Under the Precision-Recall Curve (AUPR), Precision, Recall, Kappa, Matthews. For regression tasks, the metrics include Mean Absolute Error (MAE), Mean Squared Error (MSE), Root-Mean Squared Error (RMSE), Spearman's Rank Correlation Coefficient, Pearson's Correlation Coefficient, Coefficient of Determination ($R^2$). Users have the liberty to select specific metrics for evaluating the model.

### 4.7 USE OF BENCHMOL

BenchMol is a flexible and easy-to-use framework for MRL and you can find detailed user instructions in Appendix A. Through simple and direct code invocations (syntax is `from benchmol import package`), users can easily implement the entire pipeline of MRL from initial data loading to the final model evaluation. Specifically, the use case of image modality is shown in Appendix A.6. The entire pipeline just mentioned can be completed with only 4 lines of effective code.

## 5 EXPERIMENTS

### 5.1 EXPERIMENTS SETTINGS

**Settings.** To ensure the fairness and comprehensiveness of the experimental results, unless otherwise stated, all experiments are performed under strictly consistent settings. Specifically, we use the same hyperparameter search range and report the test set results with the best validation performance on 12 molecular property prediction (MPP) benchmarks from MoleculeNet, 3 attribute datasets from MBANet, and a total of 60 molecular activity datasets of 6 different molecular types from StructNet. Meanwhile, we repeat the experiment 10 times with the same and large number of random seeds from 0 to 9 and report the mean and standard variance. See the Appendix E.1, the Appendix E.2 and the Appendix E.3 for details of baselines, hyperparameter search and training loss. The computational efficiency is discussed in the Appendix J.

**Data Split and Metrics.** All evaluation datasets are split into 80% training, 10% validation and 10% test sets. The 12 MPP datasets include 8 classification datasets and 4 regression datasets with a strict scaffold split (Hu et al., 2020a). We follow the suggestions of MoleculeNet and GraphMVP to use the ROC-AUC metric for classification tasks and the RMSE metric for regression tasks. In the remaining MBANet and StructNet benchmarks, we uniformly use RMSE for evaluation. We split MBANet benchmark by using ordered split. In StructNet benchmark, except for acyclic rule and acyclic chain rule which are random split, the others are strict scaffold split. This is because acyclic rule and acyclic chain rule cannot extract the scaffold.

Table 2: Benchmarking 7 different modality methods on 8 classification tasks with average ROC-AUC (%) and 4 regression tasks with average RMSE performance from 12 MPP datasets. The modality types from top to bottom are fingerprint, sequence, graph, geometry graph, image, geometry image, and video. L means the number of layers, -I and -G mean the modalities are image and geometry image, respectively. Note that the geometry images and videos use the BGR format. The green background represents top-6 models in performance.

| Model | Classification (↑) | Regression (↓) |
| --- | --- | --- |
| mcfp4_2048 | 71.66 | 1.255 |
| ecfp4_2048 | 69.81 | 1.300 |
| maccs | 70.54 | 1.302 |
| physchem | 63.16 | 1.454 |
| atompair_2048 | 70.28 | 1.189 |
| rdkDes | 63.19 | 1.648 |
| Chem-BERT-8L | 73.41 | 1.093 |
| MolFormer | 69.58 | 1.293 |
| EdgePred | 60.73 | 1.665 |
| ContextPred | 66.90 | 1.461 |
| infomax | 66.35 | 1.409 |
| masking | 62.53 | 1.490 |
| MolCLR | 65.50 | 1.369 |
| MoleBERT | 72.28 | 1.320 |
| GraphMVP | 65.78 | 1.401 |
| CGIP-Graph | 67.05 | 1.552 |
| Uni-Mol (10 conf) | 74.13 | 1.144 |
| ImageMol | 62.63 | 1.507 |
| MaskMol | 63.03 | 1.441 |
| CGIP-Image | 61.94 | 1.556 |
| IEM-I | 60.95 | 1.577 |
| IEM-G (10 conf) | 70.29 | 1.212 |
| VideoMol | 69.03 | 1.222 |

### 5.2 LINEAR PROBING ON MOLECULENET

To evaluate the quality of features, we use the linear probing strategy to evaluate the performance of molecular encoders on 12 MPP datasets. Specifically, BenchMol extracts features from modality-specific data using a given encoder and trains and evaluates a single-layer fully connected network.

Table 3: Effect of different numbers of conformations on 8 classification tasks (CLS) with ROC-AUC and 4 regression tasks (REG) with RMSE from 12 MPP datasets. -R means no pre-training and -G means the modalities are geometry image. The geometry images use the BGR format.

| | Uni-Mol-R | | Uni-Mol | | ResNet18-G-R | | IEM-G | |
|---|---|---|---|---|---|---|---|---|
| | CLS | REG | CLS | REG | CLS | REG | CLS | REG |
| 1 conf | **65.80** | 1.308 | 73.75 | 1.162 | 58.73 | 1.568 | 64.66 | 1.406 |
| 10 conf | 65.53 | **1.276** | **74.13** | **1.144** | 63.68 | **1.485** | 70.29 | **1.212** |
| δ | ↓0.41% | ↑2.51% | ↑0.52% | ↑1.55% | ↑8.43% | ↑5.29% | ↑8.71% | ↑13.80% |

**Findings.** Table 2, Table 3 and Table 4 show the performance of encoders with different modalities in BenchMol. More detailed results about MPP in Appendix F and Tables S20, S21, S22, S23, S24. We summarize these findings as follows:

1) **Models from 6 modalities are in the top 6 in performance.** In Table 2, the top 6 performances on the classification task are Uni-Mol (10 conf), Chem-BERT-8L, MoleBERT, mcfp4_2048, maccs, IEM-G (10 conf) and the top 6 performances on the regression task are Chem-BERT-8L, Uni-Mol (10 conf), AtomPair, IEM-G (10 conf), VideoMol, mcfp4_2048. Our analysis reveals that the top 6 include 6 modalities, except for the 2D image modality, indicating their advancement in linear probing. For the image modality, we find that it relies more on the fine-tuning stage. The Table S24 shows that ImageMol has a significant performance improvement from 62.5% to 71.9% after fine-tuning, with a performance improvement of 15.0%. There are two possible reasons: one is that there is too little information in 2D images to learn generalized knowledge in the pre-training stage, and the second is that the pre-training task still needs to be further improved. Our findings suggest using geometry images or videos to achieve better performance in vision-based representations. In Appendix K.6, we also study the impact of RGB and BGR formats.

Table 4: The average ROC-AUC (%) and RMSE performance of non-pretrained methods on 8 classification tasks (CLS) and 4 regression tasks (REG) from 12 MPP datasets. The modality types from top to bottom are sequence, graph, geometry graph, image, geometry image, and video. L means the number of layers and -G means the modality is geometry image. Note that the geometry images and videos use the BGR format.

| Model | CLS (↑) | REG (↓) |
|---|---|---|
| BERT-8L-R | 68.94 | **1.264** |
| MolFormer-R | **70.40** | 1.319 |
| GIN-R | 63.21 | 1.576 |
| Uni-Mol-R | 65.80 | 1.308 |
| Uni-Mol-R (10 conf) | 65.53 | 1.276 |
| ResNet18-R | 55.34 | 1.682 |
| ResNet18-G-R | 58.73 | 1.568 |
| ResNet18-G-R (10 conf) | 63.68 | 1.485 |
| VideoMol-R | 61.16 | 1.526 |

2) **The visual modality contributes the greatest diversity in dual-modal fusion.** The success of multi-modal fusion depends on the diversity of prediction results from different modalities Dong et al. (2020). Here, we evaluate the difference in prediction between the two modalities using RMSE and Pearson correlation coefficient on 8 classification datasets from MoleculeNet. As shown in Table S46, We find that the top 6 with the largest differences in RMSE and Pearson coefficient are all related to vision-based modalities (image, geometry image and video), which suggests that combining other modalities with the visual modality will hopefully increase the diversity of predictions of multi-modal models. We also discuss the modal diversity on HIV dataset in Appendix K.1.

3) **Multiple conformations can significantly improve the performance of the geometry image.** Since molecules have multiple conformations, we compare here the performance differences between single and multiple conformations. Here we follow the suggestion of Zhou et al. and use 10 conformations. We use geometry-based methods (Uni-Mol-R, Uni-Mol) and geometery image-based methods (ResNet18-G-R, and IEM-G). Table 3 indicates that Uni-Mol-R and Uni-Mol have almost consistent performance between 1 conformation and 10 conformations, namely 65.8% v.s. 65.5% and 73.75% v.s. 74.13% (For Uni-Mol ablation on data scale see Appendix K.2). However, we observe that ResNet18-G-R and IEM-G have significant performance gains from 1 conformation and 10 conformations, i.e. 58.73% v.s. 63.68% and 64.66% v.s. 70.29%. For a detailed analysis of why multi-conformation has a large performance gain for geometric images, see Appendix K.3.

4) **Inductive bias of identifying substructures in sequence is beneficial for predicting molecular properties.** Table 4 shows the performance of non-pretrained models. Here, we exclude hand-crafted feature-based fingerprints and focus solely on discussing methods based on automatic

feature extraction. We find that BERT-8L-R and MolFormer-R can achieve high performance on 12 MPP tasks without any pre-training, surpassing many other modality pre-training methods (such as GraphMVP, MolCLR, ImageMol, etc.). This provides evidence that the inductive bias based on the sequence is consistent with the molecule. Even without any training, the extracted features can retain the original molecular information. See Appendix K.4 and Appendix K.5 for a more detailed analysis.

Table 5: The RMSE performance on MBANet$_{atom}$, MBANet$_{bond}$, MBANet$_{attr}$, which are abbreviated as MBA$_{atom}$, MBA$_{bond}$, MBA$_{attr}$. Sequence, graph, geometry graph, geometry image, and video represent BERT-6L, GIN-R, TFN, ResNet18-I-R, ResNet18-G-R, and ResNet18-G-R, respectively.

| Modalities | MBA$_{atom}$ | MBA$_{bond}$ | MBA$_{attr}$ |
|---|---|---|---|
| Sequence | 0.522 | 2.641 | 11.448 |
| Graph | 0.340 | 0.602 | 8.514 |
| Geometry Graph | 0.177 | **0.309** | 3.091 |
| Image | 0.350 | 1.630 | 6.951 |
| Geometry Image | 0.268 | 1.586 | 4.848 |
| Video | **0.156** | 1.048 | **2.660** |

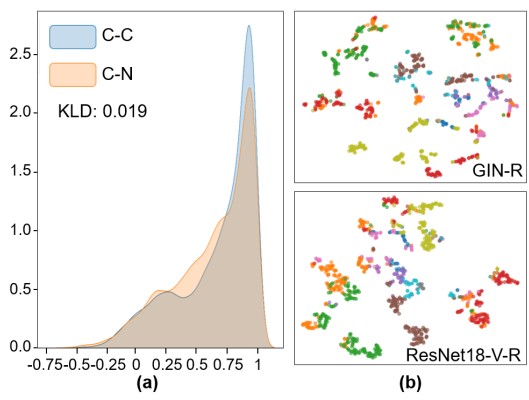

Figure 4: (a) Distribution of cosine similarity of C-C and C-N. (b) t-SNE visualization of GIN-R and ResNet18-V-R using labels of $k$-Means.

## 5.3 FINE-TUNING ON MBANET

**Findings.** The MBANet benchmark defines tasks related to molecular attributes (atom, bond, molecular weight, etc.). Here we focus on deep learning representation and ignore fingerprint methods because some fingerprints directly contain this information. For fairness, Table 5 shows the RMSE performance of non-pretrained models with different modalities on MBANet$_{atom}$, MBANet$_{bond}$, and MBANet$_{attr}$. The best performance in each modality is selected as representative of the performance of that modality and the results of all 30 methods in in Appendix G and Tables S25±S26, S27, S28±S29. We summarize the following findings:

5) **Video modality excels at tasks related to atoms and basic attributes**. In Table 5, video modality achieves the best performance on MBANet$_{atom}$ and MBANet$_{attr}$ and geometry graph modal achieve the best performance on MBANet$_{bond}$. In particular, as shown in Table S28, we find that the video modality has obvious advantages over other modalities in learning simple MW (Molecular Weight), MR (Molar Refractivity), VE (Valence Electrons) and TPSA (Topological Polar Surface Area).

6) **The inductive bias of the graph weakens the ability to discriminate at atoms.** The inductive bias of graph message passing increases the similarity between atoms of different types and makes the discrimination between different atoms confusing. To prove this point, we use the two most common atomic relationships (C-C, C-N). Specifically, we use GIN-R to extract the atomic features of C and N in the molecule respectively and calculate the cosine similarity of C-C and C-N. Figure 4(a) shows the distribution of cosine similarity. We find that the distributions of C-C and C-N are very similar with a low Kullback-Leibler Divergence (KLD) (Kullback & Leibler, 1951) of 0.019, which means that GIN-R may be limited in distinguishing C and N atoms.

7) **Video modality are easier to learn local information of molecules than graph modality.** We choose GIN-R (graph modality) and ResNet18-V-R (video modality) without pre-training models for fairness. To study the ability of GIN-R and ResNet18-V-R in extracting local information, we use t-SNE Van der Maaten & Hinton (2008) to visualize their representations on MBANet$_{atom}$ and use $k$-Means (MacQueen, 1967) to cluster the labels of the samples into 10 clusters, and the labels of each cluster are used as the labels for t-SNE visualization. Figure 4(b) shows that ResNet18-V-R with Davies-Bouldin index (DBI) (Davies & Bouldin, 1979) of 2.57 has better clustering effect than GIN-R with DBI of 4.69 (DBI is an indicator for evaluating clustering and the smaller the value, the better), indicating the advantage of visual modality in learning molecular locality. See Appendix K.7 for a more detailed analysis.

In Appendix H, we further expand the data scale of MBANet and verify the validity of the findings.

### 5.4 Fine-tuning on StructNet

**Findings.** Table 6 shows the average performance of methods with different modalities on StructNet. We also report detailed results on in Appendix I and Tables S31±S32, S33±S34, S35±S36, S37±S38, S39±S40, S41±S42. We summarize the following findings:

8) **The geometry graph modality prefers acyclic (AC and A) molecules; The fingerprint and graph modalities prefer cyclic (CC and M) molecules; The visual-based modalities (Image, Geometry Image, and Video) prefer macrocyclic peptide (MP) and reticular (R) molecules.** Table 6 shows the experimental results of different

Table 6: The average RMSE performance on AC (acyclic chain), A (acyclic), CC (complete chain), M (macro), MP (macrocyclic peptide) and R (reticular) of StructNet. We select the methods with the best average performance on 10 datasets from different modalities for presentation. Geom Image represents Geometry Image. The green background represents top-3 performance.

|  | AC | A | CC | M | MP | R |
|---|---|---|---|---|---|---|
| Fingerprint | 10.619 | 12.508 | 9.228 | 18.290 | 9.436 | 2.478 |
| Sequence | 10.439 | 12.520 | 9.307 | 18.918 | 9.385 | 2.485 |
| Graph | 10.459 | 12.536 | 9.246 | 18.109 | 10.956 | 2.496 |
| Geometry | 10.284 | 12.192 | 9.249 | 18.596 | 9.424 | 2.471 |
| Image | 10.550 | 12.473 | 9.259 | 19.136 | 9.321 | 2.454 |
| Geom Image | 10.430 | 12.482 | 9.306 | 18.846 | 9.343 | 2.467 |
| Video | 10.441 | 12.444 | 9.352 | 18.923 | 9.339 | 2.469 |

modalities without pre-training. First, we find that geometry graph achieves the best performance on acyclic chain molecules and acyclic molecules, which indicates that it prefers acyclic molecules. Then, we find that fingerprint and graph modalities achieve good performance on complete chain molecules and macro molecules. We speculate that it may be suitable for cyclic molecular structures and further counted the number of rings in complete chain molecules and macro molecules. we find that more than 93.6% of the complete chain molecules and all macro molecules are cyclic, which indicates that graph prefers cyclic structures. Finally, we find that vision-based modalities can achieve the best performance on macrocyclic peptide molecules and reticular molecules. The top 3 in performance are image, video and geometry image modalities, which indicates that vision-based modalities prefer macrocyclic peptide molecules and reticular molecules. We also provide the details of significance test in Appendix K.8 to validate the robustness of conclusions.

9) **Pre-training tasks may fail for certain types of molecules.** We are surprised to find many pre-training tasks may fail. For example, graph-based pre-training tasks achieve worse performance than unpre-trained GIN in complete chain molecules in Table S35 and image-based pre-training tasks achieve worse performance than unpre-trained ResNet18 in reticular molecules in Table S41. This suggests that in molecules with certain specific types, we need to design special pre-training for them to improve performance.

## 6 Conclusion

We first proposed a unified and flexible platform supporting different molecular modalities, called BenchMol, to promote reproducibility in the molecular representation learning (MRL) community, which provides the entire pipeline from raw data to final model evaluation and ensures fair and comprehensive benchmarking. Subsequently, we proposed two new benchmarks, MBANet and StructNet, to explore the performance and preferences of existing models on different modalities. Finally, we used BenchMol to train at least 57,060 models and provided many meaningful insights. BenchMol reviews and integrates mainstream MRL models with 7 different modalities, allowing researchers to easily understand these models and quickly iterate on them.

**Limitations and Future Works:** Currently, BenchMol does not support multi-modal fusion. In the future, we plan to continue maintaining BenchMol and upgrading it to a multi-modal fusion platform. This enhanced platform will be capable of further improving molecular representation and tackling interaction-based tasks such as drug-drug interaction (DDI), drug-target interaction (DTI), and protein-protein interaction (PPI). Furthermore, while BenchMol currently focuses on evaluating its performance in MRL, it is important to note that its applicability is not restricted to the molecular domain. We hope to leave the exploration of more fields to the community for verification.

## 7 REPRODUCIBILITY STATEMENT

For reproducibility, we open sourced BenchMol and made all data publicly accessible at Github.

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

# Appendix

## Table of Contents

## A BENCHMOL TOOLKIT USER GUIDE

### A.1 OVERVIEW

Here, we will go through the usage instructions of BenchMol in detail to demonstrate the flexibility and user-friendliness of BenchMol. In addition to the two contributed benchmarks (MBANet and StructNet), user interfaces of BenchMol are divided into four modules: Modality Extractor, Model Initializer, Training Strategy, and Evaluation Metric. Since words alone cannot fully demonstrate the full functionality of BenchMol, we next will focus on describing the main features here.

### A.2 MODALITY EXTRACTOR

Given a molecule, BenchMol can transform this molecule into different modalities, including fingerprint (can be viewed as a row-type or table-type modality), sequence, graph, geometry, image, and video. We describe several main modality extraction steps in detail below:

- **Fingerprint Modality.** Listing 1 shows the script for extracting 44 fingerprints. By changing the parameter fp_name, different fingerprint types can be extracted.

- **Sequence Modality.** BenchMol automatically extracts sequence modality from SMILES by using a predefined function *collate()*, which tokenizes a batch of molecular data.

- **Graph and Geometry Modalities.** Listing 2 shows the script for extracting geometry modality from molecular sdf file. By changing the parameter graph_feat_extractor, different geometry types can be extracted. For graph modality, the extraction process is similar to that of geometry modality, which defines two methods to generate graphs.

- **Image Modality.** BenchMol defines a function *loadSmilesAndSave(smiles, path)* which takes a smiles as input saves the molecule image to a path.

- **Geometry Image and Video Modalities.** BenchMol defines a method with parameter img_type to extract different types of visual modalities by changing the value of img_type.

```python
from benchmol.data_process.molecules import FPGeneration

fg = FPGeneration()
features = fg.get_fingerprints(
    df, fp_name="maccs", smiles_column_name="smiles"
)
```

Listing 1: Transforming process from molecules to fingerprint modality.

```python
if graph_feat_extractor == "ogb":
    x, edge_index, edge_attr, coords = mol_to_3d_graph_data_ogb(sdf_path)
elif graph_feat_extractor == "jure":
    x, edge_index, edge_attr, coords = mol_to_3d_graph_data_jure(sdf_path
    )
elif graph_feat_extractor == "geom3d":
    x, edge_index, edge_attr, coords = mol_to_graph_data_obj_simple_3D(
    sdf_path, pure_atomic_num=False)
elif graph_feat_extractor == "geom3d_pure_atomic_num":
    x, edge_index, edge_attr, coords = mol_to_graph_data_obj_simple_3D(
    sdf_path, pure_atomic_num=True)
elif graph_feat_extractor == "geom3d_full_edge":
    x, edge_index, edge_attr, coords =
    mol_to_graph_data_obj_simple_3D_full_edge(sdf_path, pure_atomic_num=
    False)
elif graph_feat_extractor == "geom3d_pure_atomic_num_full_edge":
    x, edge_index, edge_attr, coords =
    mol_to_graph_data_obj_simple_3D_full_edge(sdf_path, pure_atomic_num=
    True)
elif graph_feat_extractor == "unimol":
    atoms, coords, smi, scaffold = unimol_data(sdf_path)
else:
```

```
16      raise Exception("graph_feat_extractor {} is undefined".format(
        graph_feat_extractor))
```

Listing 2: Transforming process from molecules to geometry modality.

## A.3   MODEL INITIALIZER

As shown in Listing 3, BenchMol define 4 factories to initialize models for handling different modalities, including SmilesModelFactory, GraphModelFactory, GeometryModelFactory, and ImageModelFactory. Through these factory classes, we can easily initialize various models by giving the model name and necessary parameters (such as the configuration of the neck model related to the task).

```
1  # SmilesModelFactory for sequence modality
2  class SmilesModelFactory(torch.nn.Module):
3      def __init__(self, model_name, head_arch, num_tasks, vocab_path,
        d_dropout=0, head_arch_params=None, pretrain_path=None, device="cpu",
         **kwargs):
4          ...
5      def forward(self, batch):
6          ...
7      def get_model(self):
8          ...
9
10 # GraphModelFactory for graph modality
11 class GraphModelFactory(torch.nn.Module):
12      def __init__(self, model_name, head_arch, num_tasks, head_arch_params
        =None, pretrain_gnn_path=None, model_key=None, num_layer=5, emb_dim
        =300, JK="last", dropout=0.5, graph_pooling="mean", gnn_type="gin",
        update_predictor=True, **kwargs):
13          ...
14      def forward(self, batch):
15          ...
16      def get_model(self, update_predictor=True):
17          ...
18
19 # GeometryModelFactory for geometry modality
20 class GeometryModelFactory(torch.nn.Module):
21      def __init__(self, model_name, head_arch, num_tasks, head_arch_params
        =None, pretrain_gnn_path=None, model_key=None, emb_dim=300, args=None
        , **kwargs):
22          ...
23      def forward(self, batch):
24          ...
25      def get_model(self, args, num_tasks, node_class, edge_class):
26          ...
27
28 # ImageModelFactory for image/geometry-image/video modality
29 class ImageModelFactory(torch.nn.Module):
30      def __init__(self, model_name, head_arch, num_tasks, pretrained=False
        , head_arch_params=None, **kwargs):
31          ...
32      def forward(self, x):
33          ...
34      def get_model(self):
35          ...
```

Listing 3: The model factories of different modalities.

## A.4   TRAINING STRATEGY

BenchMol provides two training strategies, linear probing and fine-tuning.

**Linear Probing**. To improve the efficiency of linear probing, BenchMol designs 6 extractors to extract feature representations from molecules of different modalities. A core idea of these extractors is that given a pre-trained model and a molecule, the feature extractor will return features of a specific modality. As shown in 4, BenchMol defines an interface FeatureExtractor, which is an abstract class with two abstract methods: extract_features() and return_features(). Subsequently, we defined an implementation class for each modality to extract features from the corresponding modality. The specific details class is as follows:

- **FingerprintExtractor.** The features of fingerprint can be extracted by giving the fingerprint name and SMILES sequence.
- **SmilesFeatureExtractor.** The sequence features are generated from molecular SMILES sequences.
- **GraphFeatureExtractor.** The graph features can be extracted from graph modality given a pre-trained graph model.
- **ImageFeatureExtractor.** The image feature can be extracted from image modality given a pre-trained image model.
- **MCImageFeatureExtractor.** The features of geometry image and video can be extracted given a corresponding pre-trained model.
- **MVImageFeatureExtractor.** The extractor is used to extract features from molecular geometry images or videos with multiple conformers.

```python
@dataclasses.dataclass
class FeatureExtractor(abc.ABC):

    @abc.abstractmethod
    def extract_features(self):
        pass

    @abc.abstractmethod
    def return_features(self):
        pass
```

Listing 4: General interface of modality extractor.

**Fine-Tuning.** Fine-tuning refers to training the pre-trained model and the network related to the downstream task at the same time. With BenchMol, fine-tuning of different modes can be achieved very easily. Appendix A.6 shows fine-tuning on image modality. Similar to the image modality, we can also easily implement fine-tuning of other modalities.

A.5    EVALUATION METRIC

BenchMol provides a large number of metrics for classification and regression tasks and users can switch between different evaluation metrics at will by specifying the eval_metric parameter. The following shows the evaluation indicators supported by different tasks:

- **Classification task:** accuracy, ROC-AUC, F1-score, AUPR, Precision, Recall, Kappa coefficient, Matthews coefficient.
- **Regression task:** MAE, MSE, RMSE, Ppearman coefficient, Pearson coefficient, $R^2$ coefficient.

A.6    USE CASE FOR IMAGE MODALITY

We assume that there is a requirement: there are a batch of molecular images and corresponding binary labels of active or inactive. We first need to divide these data into training set, validation set and test set. Then, we need to use a ViT network to train on the training set and use the ROC-AUC metric to evaluate on the validation set. Finally, we need to obtain the results of the test set based on the best performance on the validation set. We can easily achieve this with BenchMol. Listing 5 shows instructions for using BenchMol in image modality, which shows the flexibility and ease of use of BenchMol. We can see that only 4 lines of effective code are needed to complete the training of the model, which are lines #10, #15, #23, and #28.

```python
from benchmol.dataloader.image_dataset import TrainValTestFromCSVFactory
from benchmol.trainer import Trainer
from benchmol.model_pools import ImageModelFactory

# Take the image modality and classification task as an example
modality="image"
task_type = "classification"

# define Model with backbone as ViT and neck as arch4 for n tasks
model = ImageModelFactory(
    model_name="vit_small_patch16_224", head_arch="arch4",
    head_arch_params={"inner_dim": 128, "dropout": 0.2, "activation_fn":
    "gelu"}, num_tasks=2
)

# define Dataset
factory = TrainValTestFromCSVFactory(
    dataroot, csv_path, data_type="image", image_dir_name="image",
    task_type=task_type, batch_size=16, num_workers=8
)
train_loader = factory.get_dataloader(split="train")
valid_loader = factory.get_dataloader(split="valid")
test_loader = factory.get_dataloader(split="test")

# define Trainer
trainer = Trainer(
    model, modality, train_loader, valid_loader, test_loader, task_type,
    criterion=nn.BCEWithLogitsLoss(reduction="none"), optimizer=Adam(
    model.parameters(), lr=0.001, weight_decay=1e-5), device="cuda:0"
)

# training and evaluation
results = trainer.train(num_epochs=100, eval_metric="ROCAUC",
    valid_select="max", min_value=-np.inf, save_finetune_ckpt=True,
    save_dir="./experiments/")

# Output model results
print("results: {}\n".format(results))
```

Listing 5: The use case on image modality with classification task and ROC-AUC metric.

# B    VISUAL RENDERING OF MOLECULE

We describe in detail how to render molecules as images, geometry images, and videos:

- **Rendering of image:** Following CGIP (Xiang et al., 2023), the molecualr image is generated by the *MolsToGridImage()* method of RDKit. This method takes the SMILES sequence of a molecule as input and generates an image of length 224, width 224, and 3 channels.

- **Rendering of geometry image:** Molecular images represent the 2D planar structure of molecules. However, molecules have three-dimensional conformational information. Following IEM (Xiang et al., 2024a), we use PyMOL to render the 3D structure of the molecule. Since the 3D structure is easily obscured when displayed on a single image, 4 viewing angles are used to render the molecule from different angles. Therefore, through multi-view rendering with PyMOL, we can obtain 4 geometric images with different vies for a molecule, which can be formulated as a matrix of $4 \times 224 \times 224 \times 3$.

- **Rendering of video:** VideoMol (Xiang et al., 2024b) represents the 3D image of a molecule as a video. Specifically, a molecular video is constructed by rotating a molecule along the $x$-axis, $y$-axis, and $z$-axis. During the rotation process, VideoMol captures 60 frames at equal intervals to represent the molecular video. Therefore, we use PyMOL to render the molecule with 3D information and generate a 60-frame video, which can be formulated as a matrix of $60 \times 224 \times 224 \times 3$.

# C    DISCUSSION

## C.1    MOTIVATION AND POTENTIAL IMPACT OF BENCHMOL

**Motivation.** The main motivations for BenchMol are to address the following research gaps in the field of chemical machine learning: (1) There is a lack of fair and comprehensive evaluation of methods across different modalities; (2) The strengths and differences of different modalities are still unknown, which limits the development of multi-modal fusion; and (3) There is a lack of a unified and easy-to-use platform to integrate methods across different modalities. With BenchMol, researchers can easily use and compare various molecular representation learning methods.

**Potential Impact.** The two benchmarks MBANet and StructNet proposed in this paper have important impacts on promoting certain chemical problems, which are summarized as follows:

- MBANet aims to evaluate the ability of different methods to capture basic molecular information that is critical for many chemical problems. For example, the distribution of atoms and bonds is crucial for understanding the three-dimensional structure of a molecule [1] and aids in molecular dynamics simulations [2]. Predicting the basic attributes of a molecule helps design molecules with specific properties [3]. Attributes such as molecular weight, MolMR (molecular refractive index), and NumHDonors (number of hydrogen bond donors) can be used to infer the biological activity of a molecule and its interaction with its target [4].

- StructNet aims to explore the preferences of various methods for molecules of different structural types and it is of great significance for certain specific targets. For example, molecules targeting KRAS targets are often macromolecules, and the model needs to learn and predict in the sample space of macromolecules [5]; a class of antiviral and antimalarial drugs usually have chain-like molecular structures, while molecules targeting fibroblast activation protein (FAP) usually exhibit non-cyclic structures. These chemical preferences indicate that it is important to select appropriate molecular modalities to work more effectively in different scenarios.

## C.2    MOTIVATION AND PRACTICALITY OF BENCHMARK DATASETS

**MBANet.** The motivation and practicality of MBANet are as follows:

- **Motivation.** Currently, a large number of benchmarks focus on mapping molecules to complex properties or biological activities. However, it is a complex process for models to learn to map molecules to complex properties or biological activities, which may be related to the regulatory network of molecules from a microscopic perspective. This complex process is not conducive to describing the model's understanding of the basic properties of molecules. In this paper, we hope to clarify the understanding of the most basic attributes of molecules by different modalities. This basic attribute reflects the properties directly related to the molecule and has nothing to do with the complex regulatory network. This has always been a research gap but is also crucial for the model to understand molecules. Therefore, we design MBANet to evaluate the model's ability to understand the basic attributes of molecules.

- **Practicality.** From a practical point of view, the basic attributes evaluated by MBANet are closely related to complex properties or biological activities. For example, LogP is an important indicator of BBBP and it and TPSA are two key physicochemical parameters in drug design (Prasanna & Doerksen, 2009), affecting drug absorption, distribution, metabolism, excretion and toxicity.

We hope that this simple, decoupled basic task can provide reference and more thinking for modality selection on related active tasks. In addition, we will consider providing more complex prediction tasks for MBANet in future work to make it more comprehensive. However, in this paper, we prefer to study this unique aspect.

**StructNet.** We describe the motivation and practicality of classifying molecules into different types based on their 2D structural patterns (e.g., acyclic, acyclic chain, cyclic chain, macrocyclic peptide, macromolecule, and reticular molecule). Molecular structure is intimately linked to molecular properties, so that certain drug targets exhibit preferences for specific molecular structures, and some therapeutic applications correspond to molecules with particular structures. For instance:

- **Macrocyclic peptides** are increasingly being recognized for their therapeutic potential in targeting aberrant protein-protein interactions (PPIs), with several macrocyclic peptide-based oncology drugs already approved by the U.S. Food and Drug Administration (FDA) for clinical application (Yang et al., 2022).

- Due to the mechanism of action of the KRAS target involving covalent binding with a cysteine generated by the mutation of glycine at the 12th position in proteins, the designed molecules tend to have a larger molecular weight, resulting in a relatively larger binding area, more stable binding, and stronger specificity. Therefore, molecules targeting KRAS targets tend to be **macromolecules** (Cox & Der, 2024).

- **Acyclic single-chain** fragment variable (scFv) molecules target fibroblast activation protein (FAP) (Baum et al., 2007) and have applications in CAR-T cell therapy (Niu et al., 2024; Loureiro et al., 2023).

- As structural analogues to the **chain molecules** defined herein, acyclonucleoside phosphonates (ANPs) exhibit a distinctive acyclic structure (Bessières et al., 2024), which constitute a significant class of compounds with antiviral and anticancer properties (Holỳ, 2006), and they harbor substantial potential as candidates for antimalarial drug development (Cheviet et al., 2020).

Thus, we have introduced StructNet, which categorizes based on 2D structural patterns, with the aim of offering insights and suggestions on model selection for drug design targeting specific structures and the future optimization directions for various models.

### C.3 THE IMPORTANCE OF MOLECULAR VISUAL MODALITIES

Currently, molecular visual modalities consist of image, geometry image, video. Compared with previous molecular representations (such as SMILES or graph), the importance of molecular visual modalities is reflected in the following four aspects:

- **Direct representation of structural and geometric information**: Compared with previous SMILES or graph representations, molecular visual modalities naturally retain structural and geometric properties such as atom type, chemical bond type, bond angle, spatial conformation, etc. through pixel information (color, texture, etc.). Molecular videos can further describe the dynamic information of molecules, which is particularly important for tasks involving molecular geometry or dynamic behavior;

- **Stronger interpretability**: Molecular visual modalities show the molecular structure and its dynamic behavior in an intuitive form, which is convenient for humans to understand and analyze the features learned by the model. For example, through technologies such as GradCAM (Selvaraju et al., 2017), researchers can intuitively understand how the model makes decisions;

- **Enriching molecular representation technology**: Molecular visual modalities can use another technology stack (computer vision) to extract potential features from molecular images, geometry images or videos, enriching existing molecular representation technology;

- **Independence of the number of atoms and bonds**: Molecular visual modalities are pixel-based representations, which are independent of the number of atoms and bonds. At present, drug discovery is increasingly biased towards large molecules. Significantly different from SMILES and graph, visual modalities have the unique advantage that the computational cost does not increase with the number of atoms and bonds, which will be of great benefit in drug development of large molecules.

Molecular visual modalities are particularly suitable for the following tasks, which rely on the spatial configuration or dynamic characteristics of molecules:

- **General molecular representation**: Image-, geometry-image- and video-based methods have achieved excellent performance in various drug discovery tasks, just like other molecular representation methods (as shown in Table S20 and Table S21, etc.). Therefore, they can be used for general representation of drug discovery tasks;

- **Multi-modal fusion**: Obviously, visual modalties are different from previous molecular representation learning methods. Through multi-modal fusion or cross-modal contrastive learning tasks, more diversity will be provided to further improve the performance of drug discovery. As shown in Appendix K.1, Table S47 and Table S48, video has the largest differences with other modalities;

- **Tasks related to atomic molecular distribution and basic properties**: As shown in Table 5, video modality can achieve the best performance on atom-level and attr-level tasks of MBANet. Therefore, the video modality makes up for the lack of understanding of atomic distribution and basic properties of other modal representations;

- **Virtual screening of macrocyclic or reticular structures**: As shown in Table 6, images and videos can achieve the best performance on MP and R tasks on StructNet. Therefore, images or videos make up for the lack of understanding of macrocyclic peptide (MP) and reticular (R) structures by other modalities, such as virtual screening of PPI targets (Cox & Der, 2024).

In general, molecules exist in the physical world. Currently, due to the limitations of molecular imaging technology, molecular images/geometry images/videos are obtained by economical image rendering technology. However, with the continuous advancement of molecular imaging technology, it is promising to directly represent molecules and inference about them in a visual way.

# D    BENCHMARKS

Here, we present the details of the two proposed sets of benchmarks: MBANet and StructNet.

## D.1    MBANET

MBANet is used to evaluate the performance of deep learning models in understanding the basic information of molecules, aiming to measure whether the model can effectively capture the low-level core features related to molecules. In the fields of cheminformatics and drug development, the basic information of molecules (such as atomic information, bond information, molecular weight, TPSA and other basic attributes) is crucial for predicting molecular activity, drug-target interactions, and chemical reactivity. Therefore, MBANet attempts to examine the model's capabilities at these basic levels and explore whether the model can accurately capture these basic features by learning molecules.

MBANet has a total of 10,000 molecules and includes three groups of prediction tasks, namely prediction of atoms $MBANet_{atom}$, bonds $MBANet_{bond}$ and basic attributes $MBANet_{attr}$. We check MBANet for duplication according to canonical SMILES and find that only 25 molecules are duplicated in canonical SMILES. Therefore, the impact on the evaluation is negligible. In the MBANet benchmark, the model needs to predict the atomic distribution, bond distribution, and basic attributes given a molecule. The Figure S1, Figure S2, and Figure S3 show the distribution information of $MBANet_{atom}$, $MBANet_{bond}$ and $MBANet_{attr}$ respectively. In particular, we describe the meaning of each attribute in $MBANet_{attr}$ in detail as follows:

- **Molecular Weight:** The sum of the relative atomic masses of all atoms in a molecule, usually expressed in daltons (Da) or grams per mole (g/mol). It is a fundamental property that affects a compound's physical and chemical behavior.

- **MolLogP (LogP):** The logarithmic value of the distribution coefficient ratio (P) of a compound in n-octanol (oil) and water. It indicates the hydrophobicity or lipophilicity of the molecule, with higher values suggesting greater affinity for lipid environments.

- **MolMR (Molecular Refractivity):** A calculated property that reflects the volume and polarizability of a molecule. It is used to estimate the interactions of the molecule with its environment, including its ability to penetrate biological membranes.

- **BalabanJ (Balaban's J Index):** A topological index that measures the complexity of a molecular structure. It is used in quantitative structure-activity relationship (QSAR) studies to correlate molecular structure with biological activity.

- **NumValenceElectrons (Number of Valence Electrons):** The total number of valence electrons present in the atoms of a molecule. This property is important for understanding the chemical reactivity and bonding behavior of the compound.

- **TPSA (Topological Polar Surface Area):** A measure of the polar surface area of a molecule, calculated based on its structure. TPSA is often used to predict a compound's absorption, permeability, and bioavailability.

- **NumHAcceptors (Number of Hydrogen Bond Acceptors):** The count of atoms in a molecule that can accept hydrogen bonds, typically involving oxygen and nitrogen atoms. This property is crucial for evaluating molecular interactions in biological systems.
- **NumHDonors (Number of Hydrogen Bond Donors):** The count of atoms that can donate hydrogen bonds, usually hydrogen atoms attached to electronegative atoms like nitrogen or oxygen. This property influences a molecule's interaction with biological targets.

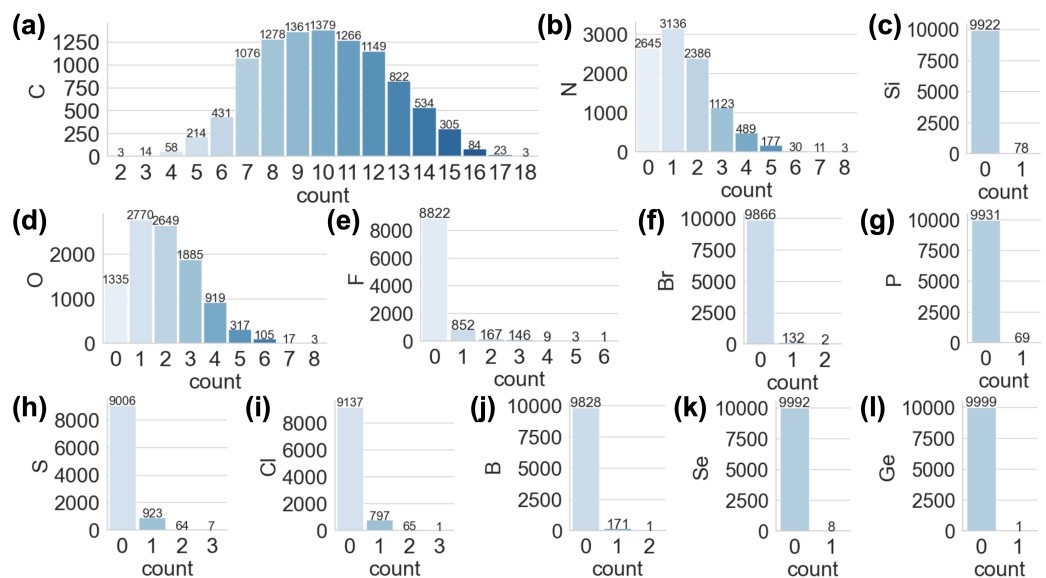

Figure S1: The distribution figures of MBANet$_{atom}$. (a)-(i) represent the distribution information of C, N, Si, O, F, Br, P, S, Cl, B, Se, Ge respectively.

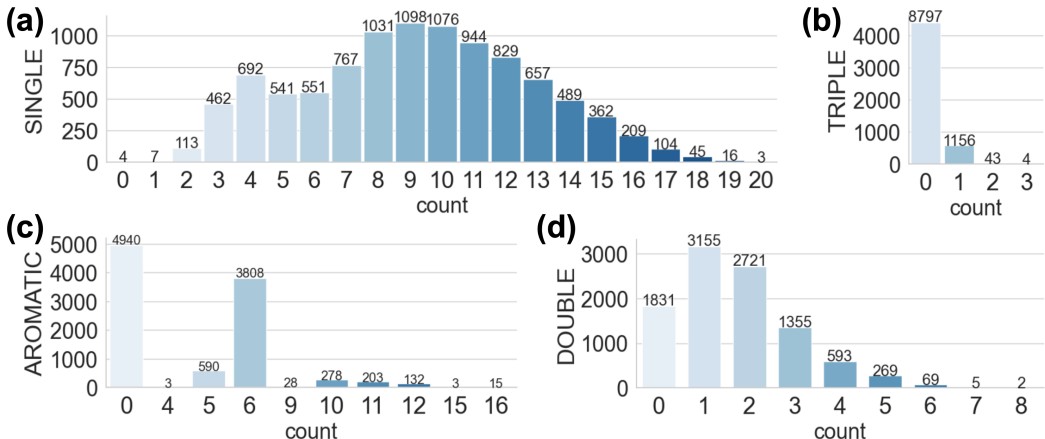

Figure S2: The distribution figures of MBANet$_{bond}$. (a)-(d) represent the distribution information of single, triple, aromatic, double bonds respectively.

**MBANet Task Settings and Limitation Analysis.** The reasons why we incorporate atoms with skewed distribution histograms into MBANet and design MBANet as a regression task and the evaluation metric of RMSE are as follows:

- **Reasons for including atoms (Si, Br, P, S, Cl, B, Se, Ge) and bonds (TRIPLE) with skewed distribution histograms.** This consideration is mainly to reflect the wide applicability of the evaluation. Even if the data distribution of some atoms and bonds is skewed, they reflect the actual chemical distribution. For the special case of Ge, it is a good choice to remove the task of

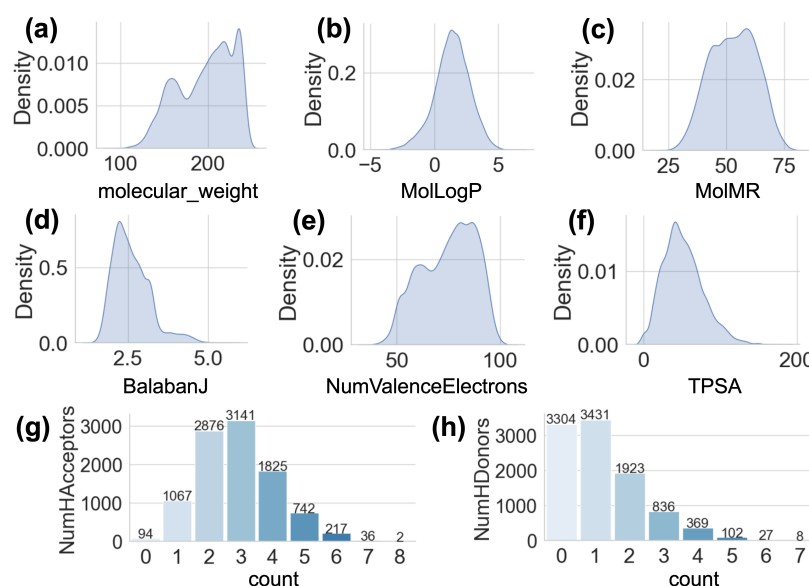

Figure S3: The distribution figures of MBANet$_{attr}$. (a)-(h) represent the distribution information of molecular weight, MolLogP, MolMR, BalabanJ, NumValenceElectrons, TPSA, NumHAcceptors, NumHDonors respectively.

this atom. But we still prefer to keep it. In detail, in our training process, the absence of Ge atoms is also a supervision label, that is, the number of Ge atoms is 0. Therefore, when training, the model needs to predict that there is no Ge in a molecule, which also reflects the learning ability of the model in this case. In addition, we can also see from Table S25 that Ge has little effect on the results, which will not affect the main conclusions given in the paper. Therefore, we tend to keep these atoms and bonds with skewed distributions;

• **Reasons for choosing regression tasks**. Given that the chemical space of molecules is extremely wide, it is difficult for us to exhaustively list all possible numbers of atoms and bonds because they are still expanding. Therefore, using regression tasks is a more intuitive choice, which is not limited by the number of specific classification tasks and is more suitable for handling such complex situations.

• **Reasons for using RMSE evaluation**. The reason why we choose to use RMSE (Root Mean Squared Error) instead of Spearman or Pearson correlation coefficient in regression tasks is mainly because the correlation coefficient only reflects the relative ranking or linear correlation between the predicted value and the true value and cannot directly quantify the actual error of the prediction. In the task of predicting the distribution of atoms and bonds, we need the model to accurately predict the number of certain atoms or bonds in a molecule. Therefore, we prefer to use RMSE to evaluate the performance of the model.

Although atoms with skewed distribution histograms have little impact on the final conclusion, we have to acknowledge that MBANet may be affected by atoms with extremely skewed count histograms, which may introduce noise to the final metrics.

### D.2 STRUCTNET

StructNet has a total of 6 different molecular types, including reticular (R)-, acyclic (A)-, complete chain (CC)-, acyclic chain (AC)-, macrocyclic peptide (MP)-, macro (M)-molecules. The scenarios corresponding to these six different types of molecules are of great significance in drug discovery. For example, molecules targeting KRAS targets tend to be large molecules and the model needs to learn and predict in the sample space of large molecules (Cox & Der, 2024). Here, we first give the specific 6 screening rules for each type of molecule, as follows:

- **R rule for reticular molecules.** Reticular molecules exhibit better structural stability, drug loading capacity, and controllable release due to their complex cross-linked pore structure and larger specific surface area, making them suitable for multiple fields such as drug carriers, tissue engineering, and drug controlled release systems. Because reticular molecules are highly cross-linked structures composed of multiple interwoven molecular chains, similar to a network mesh. The more cross-linking (chemical bonds) between atoms, the higher the average degree. So we define molecules with average degree greater than 2.33 as reticular molecules.

- **A rule for acyclic molecules.** Acyclic molecules usually have higher reactivity and spatial adaptability, and can better bind to active sites that cyclic structures cannot reach smoothly due to steric hindrance in cyclic structures. Thus, We retain molecules with ring number equal to 0.

- **CC rule for complete chain molecules.** Compared to other structural molecules, chain molecules serve as drug molecules or carriers, based on their better flexibility and adaptability, high specificity and affinity, targeting specific biological targets to achieve precise treatment or drug delivery. Chain molecule is a long chain structure composed of a main molecular chain with few branches, where most of the atoms have a degree of 2. Therefore, We retain molecules where the ratio of branched atoms (atoms with degree greater than 2) to the total number of atoms is less than 0.2.

- **AC rule for acyclic chain molecules.** Strictly speaking, chain molecules do not have cyclic structures. Therefore, based on custom chain molecules, we retain molecules where the ratio of branched atoms (atoms with degree greater than 2) to the total number of atoms is less than 0.255 and the number of rings is equal to 0 as acyclic chain molecules.

- **MP rule for macrocyclic peptide molecules.** As one of the hot topics in the field of drug development today, macrocyclic peptides have high targeting, excellent pharmacokinetic properties, and low immunogenicity. Some macrocyclic peptides even have strong penetration ability. These advantages make macrocyclic peptides have broad application prospects in drug development and treatment. Therefore, we chose macrocyclic peptides as a class of structural molecules to study. We retain molecules with a maximum number of rings greater than 12 and a number of peptide bonds greater than 0 according to the definition of macrocyclic peptide, which refer to compounds connected by peptide bond (amide bond) and possessing a large cyclic structure.

- **M rule for macro molecules.** Large molecule targeted drugs, such as monoclonal antibodies, generally act on targets on the cell surface and have strong specificity. Moreover, compared to traditional small molecule targeted drugs, large molecule drugs have a longer half-life, which greatly reduces the frequency of medication. Therefore, We keep molecules with molecular weight greater than 900 as macro molecules.

After filtering according to the above rules, for each molecular type, the top 10 assay with the largest number of molecules are selected to further construct the StructNet, which means that each molecular type has 10 datasets. In particular, for multiple trials on the same SMILES, we only keep one and use the mean of multiple trials as the label. We show the statistics of the datasets with different molecular types in Tables S1, S2, S3, S4, S5 and S6.

Table S1: The 10 datasets composed of **reticular** molecules in StructNet, called R, with scaffold split.

| Assay ChEMBL ID | Train/Valid/Test | Standard Type |
|---|---|---|
| CHEMBL4888485 | 512/64/65 | Inhibition(%) |
| CHEMBL1614458 | 391/49/49 | Potency(nM) |
| CHEMBL1614459 | 277/35/35 | Potency(nM) |
| CHEMBL1613914 | 241/30/31 | Potency(nM) |
| CHEMBL1614421 | 161/54/54 | Potency(nM) |
| CHEMBL1614087 | 155/52/52 | Potency(nM) |
| CHEMBL1614249 | 148/49/50 | Potency(nM) |
| CHEMBL1614236 | 139/46/47 | Potency(nM) |
| CHEMBL1614544 | 130/44/44 | Potency(nM) |
| CHEMBL1614038 | 127/43/43 | Potency(nM) |

Table S2: The 10 datasets composed of **acyclic** molecules in StructNet, called A, with random split.

| Assay ChEMBL ID | Train/Valid/Test | Standard Type |
|---|---|---|
| CHEMBL4513082 | 271/34/34 | Inhibition(%) |
| CHEMBL4495582 | 271/34/34 | Inhibition(%) |
| CHEMBL1614458 | 246/31/31 | Potency(nM) |
| CHEMBL4303805 | 164/56/56 | Inhibition(%) |
| CHEMBL4808149 | 164/56/56 | Inhibition(%) |
| CHEMBL4296187 | 164/56/56 | Inhibition(%) |
| CHEMBL4808150 | 163/55/55 | Inhibition(%) |
| CHEMBL4296188 | 153/52/52 | Inhibition(%) |
| CHEMBL4649955 | 153/51/51 | Percent Effect(%) |
| CHEMBL4649949 | 153/51/51 | Percent Effect(%) |

Table S3: The 10 datasets composed of **complete chain** molecules in StructNet, called CC, with scaffold split.

| Assay ChEMBL ID | Train/Valid/Test | Standard Type |
|---|---|---|
| CHEMBL4649949 | 1410/176/177 | Percent Effect(%) |
| CHEMBL4649948 | 1410/176/177 | Percent Effect(%) |
| CHEMBL4649955 | 1393/174/175 | Percent Effect(%) |
| CHEMBL4888485 | 1336/167/167 | Inhibition(%) |
| CHEMBL4296187 | 1005/126/126 | Inhibition(%) |
| CHEMBL4296188 | 956/120/120 | Inhibition(%) |
| CHEMBL4296802 | 907/113/114 | Inhibition(%) |
| CHEMBL1614459 | 852/106/107 | Potency(nM) |
| CHEMBL1614458 | 689/86/87 | Potency(nM) |
| CHEMBL1614530 | 540/68/68 | Potency(nM) |

Table S4: The 10 datasets composed of **acyclic chain** molecules in StructNet, called AC, with random split.

| Assay ChEMBL ID | Train/Valid/Test | Standard Type |
|---|---|---|
| CHEMBL1614458 | 145/49/49 | Potency(nM) |
| CHEMBL4513082 | 137/47/47 | Inhibition(%) |
| CHEMBL4495582 | 137/47/47 | Inhibition(%) |
| CHEMBL4296187 | 124/42/42 | Inhibition(%) |
| CHEMBL4296188 | 115/39/39 | Inhibition(%) |
| CHEMBL1614361 | 110/38/38 | Potency(nM) |
| CHEMBL4303805 | 108/36/36 | Inhibition(%) |
| CHEMBL4649955 | 107/36/36 | Percent Effect(%) |
| CHEMBL4649949 | 107/36/36 | Percent Effect(%) |
| CHEMBL4649948 | 107/36/36 | Percent Effect(%) |

Next, we describe the details of the standard type. The standard type in the ChEMBL database refers to the type of biological or chemical measurement that is being recorded in the database for a particular bioactivity or assay, which defines the specific biological endpoint or property that has been measured for a compound. The standard types included in StructNet benchmarks are as follows:

Table S5: The 10 datasets composed of **macrocyclic peptide** (MP) molecules in StructNet, with scaffold split.

| Assay ChEMBL ID | Train/Valid/Test | Standard Type |
|---|---|---|
| CHEMBL4888485 | 384/48/48 | Inhibition(%) |
| CHEMBL2354301 | 336/42/42 | AC50(nM) |
| CHEMBL3880198 | 168/56/57 | Ki(nM) |
| CHEMBL4420271 | 150/50/51 | Inhibition(%) |
| CHEMBL4419595 | 150/50/51 | Inhibition(%) |
| CHEMBL4420282 | 136/45/46 | IC50(nM) |
| CHEMBL3214979 | 129/44/43 | AC50(nM) |
| CHEMBL4420277 | 124/42/42 | Inhibition(%) |
| CHEMBL4419601 | 124/42/42 | Inhibition(%) |
| CHEMBL4419606 | 124/42/42 | IC50(nM) |

Table S6: The 10 datasets composed of **macro** molecules in StructNet, called M, with scaffold split.

| Assay ChEMBL ID | Train/Valid/Test | Standard Type |
|---|---|---|
| CHEMBL4420282 | 1125/141/141 | IC50(nM) |
| CHEMBL4419606 | 985/123/124 | IC50(nM) |
| CHEMBL4420281 | 580/72/73 | Inhibition(%) |
| CHEMBL3881498 | 569/71/72 | Inhibition(%) |
| CHEMBL4419605 | 568/71/72 | Inhibition(%) |
| CHEMBL4420271 | 555/69/70 | Inhibition(%) |
| CHEMBL4419595 | 555/69/70 | Inhibition(%) |
| CHEMBL3881499 | 548/69/69 | IC50(nM) |
| CHEMBL4420273 | 418/52/53 | Inhibition(%) |
| CHEMBL4419597 | 418/52/53 | Inhibition(%) |

- **Inhibition** means the inhibition rate under certain conditions, and its unit is %.

- **Potency** means the dosage at which a drug achieves a certain pharmacological effect, and its unit is nM.

- **Percent Effect** means the percentage of physiological effects caused by a drug, measuring the efficacy of the drug, and its unit is %.

- **AC50** means 50% the maximum active concentration. The concentration at which a drug reaches 50% of its maximum effect under specific conditions. its unit is nM.

- **Ki** means inhibition constant, which is the concentration of the free inhibitor corresponding to the binding of 50% of the enzyme to the inhibitor. its unit is nM.

- **IC50** means 50% inhibitory concentration, which is the concentration of the inhibitor required to achieve a 50% inhibitory effect. its unit is nM.

In StructNet, an Assay ChEMBL ID corresponds to an assay. For example, CHEMBL1614458 represents a biological assay: qHTS Assay for Inhibitors of Aldehyde Dehydrogenase 1 (ALDH1A1). CHEMBL1614459 is qHTS Assay for Lipid Storage Modulators in Drosophila S3 Cells. CHEMBL1613914 is qHTS Assay for Inhibitors of Human Jumonji Domain Containing 2E (JMJD2E). You can easily find the description of all assays in StructNet by following the link https://www.ebi.ac.uk/chembl/assay_report_card/{using your Assay ChEMBL ID in here}/.

It is worth noting that when constructing StructNet from ChEMBL, considering that the isomerism phenomenon of drug molecules may significantly affect the measurement results of its biological activity, we only use SMILES as the merging condition of multiple experiments instead of canonical SMILES. We filtered through RDKit's isomericSmiles condition and find that all SMILES are isomers, so we retain the differences between SMILES. We also present the results of repeatability detection using canonical SMILES. As shown in Tables S7, S8, S9, S10, S11, S12, we find that in

most datasets, the duplication of canonical SMILES is not obvious. Notably, we perform an isomer check for SMILES using RDKit on all molecules where canonical SMILES was repeated. The results show that these SMILES repeated in canonical formats sequences are isomers.

Table S7: Statistics after deduplication using canonical SMILES in **reticular** molecules from StructNet, called R. #Mol represents the total number of molecules. #Unique represents the number of molecules after removing duplicates. Ratio represents $1 - \frac{\#\text{Unique}}{\#\text{Mol}} \times 100$.

| Assay ChEMBL ID | #Mol / #Unique / #Ratio |
|---|---|
| CHEMBL1613914 | 302 / 300 / 0.7% |
| CHEMBL1614038 | 213 / 213 / 0% |
| CHEMBL1614087 | 259 / 257 / 0.8% |
| CHEMBL1614236 | 232 / 232 / 0% |
| CHEMBL1614249 | 247 / 246 / 0.4% |
| CHEMBL1614421 | 269 / 264 / 1.9% |
| CHEMBL1614458 | 489 / 486 / 0.6% |
| CHEMBL1614459 | 347 / 346 / 0.3% |
| CHEMBL1614544 | 218 / 215 / 1.4% |
| CHEMBL4888485 | 641 / 637 / 0.6% |

Table S8: Statistics after deduplication using canonical SMILES in **acyclic** molecules from StructNet, called A. #Mol represents the total number of molecules. #Unique represents the number of molecules after removing duplicates. Ratio represents $1 - \frac{\#\text{Unique}}{\#\text{Mol}} \times 100$.

| Assay ChEMBL ID | #Mol / #Unique / #Ratio |
|---|---|
| CHEMBL1614458 | 308 / 302 / 1.9% |
| CHEMBL4296187 | 274 / 274 / 0% |
| CHEMBL4296188 | 257 / 257 / 0% |
| CHEMBL4303805 | 276 / 273 / 1.1% |
| CHEMBL4495582 | 339 / 331 / 2.4% |
| CHEMBL4513082 | 339 / 331 / 2.4% |
| CHEMBL4649949 | 255 / 255 / 0% |
| CHEMBL4649955 | 255 / 255 / 0% |
| CHEMBL4808149 | 274 / 269 / 1.8% |
| CHEMBL4808150 | 273 / 268 / 1.8% |

Table S9: Statistics after deduplication using canonical SMILES in **complete chain** molecules from StructNet, called CC.

| Assay ChEMBL ID | #Mol / #Unique / #Ratio |
|---|---|
| CHEMBL1614458 | 862 / 855 / 0.8% |
| CHEMBL1614459 | 1065 / 1064 / 0.1% |
| CHEMBL1614530 | 676 / 676 / 0% |
| CHEMBL4296187 | 1257 / 1257 / 0% |
| CHEMBL4296188 | 1196 / 1196 / 0% |
| CHEMBL4296802 | 1134 / 1134 / 0% |
| CHEMBL4649948 | 1763 / 1762 / 0.1% |
| CHEMBL4649949 | 1763 / 1762 / 0.1% |
| CHEMBL4649955 | 1742 / 1741 / 0.1% |
| CHEMBL4888485 | 1670 / 1669 / 0.1% |

Table S10: Statistics after deduplication using canonical SMILES in **acyclic chain** molecules from StructNet, called AC.

| Assay ChEMBL ID | #Mol / #Unique / #Ratio |
|---|---|
| CHEMBL1614361 | 186 / 184 / 1.1% |
| CHEMBL1614458 | 243 / 240 / 1.2% |
| CHEMBL4296187 | 208 / 208 / 0% |
| CHEMBL4296188 | 193 / 193 / 0% |
| CHEMBL4303805 | 180 / 179 / 0.6% |
| CHEMBL4495582 | 231 / 228 / 1.3% |
| CHEMBL4513082 | 231 / 228 / 1.3% |
| CHEMBL4649948 | 179 / 179 / 0% |
| CHEMBL4649949 | 179 / 179 / 0% |
| CHEMBL4649955 | 179 / 179 / 0% |

Table S11: Statistics after deduplication using canonical SMILES in **macrocyclic peptide** molecules from StructNet, called MP.

| Assay ChEMBL ID | #Mol / #Unique / #Ratio |
|---|---|
| CHEMBL2354301 | 420 / 131 / 68.8% |
| CHEMBL3214979 | 216 / 87 / 59.7% |
| CHEMBL3880198 | 281 / 243 / 13.5% |
| CHEMBL4419595 | 251 / 221 / 12% |
| CHEMBL4419601 | 208 / 180 / 13.5% |
| CHEMBL4419606 | 207 / 181 / 12.6% |
| CHEMBL4420271 | 251 / 221 / 12% |
| CHEMBL4420277 | 208 / 180 / 13.5% |
| CHEMBL4420282 | 227 / 198 / 12.8% |
| CHEMBL4888485 | 480 / 480 / 0% |

Table S12: Statistics after deduplication using canonical SMILES in **macro** molecules from StructNet, called M.

| Assay ChEMBL ID | #Mol / #Unique / #Ratio |
|---|---|
| CHEMBL3881498 | 712 / 696 / 2.2% |
| CHEMBL3881499 | 686 / 666 / 2.9% |
| CHEMBL4419595 | 694 / 657 / 5.3% |
| CHEMBL4419597 | 523 / 507 / 3.1% |
| CHEMBL4419605 | 711 / 693 / 2.5% |
| CHEMBL4419606 | 1232 / 1187 / 3.7% |
| CHEMBL4420271 | 694 / 657 / 5.3% |
| CHEMBL4420273 | 523 / 507 / 3.1% |
| CHEMBL4420281 | 725 / 707 / 2.5% |
| CHEMBL4420282 | 1407 / 1358 / 3.5% |

# E EXPERIMENT SETTINGS

## E.1 BASELINES

To comprehensively evaluate the performance of different modality-based methods, we selected a large number of common baselines for each modality. However, since BenchMol supports many baseline methods, it is difficult for us to cover all baseline methods. Therefore, we select methods based on whether they are representative and leave the remaining methods to the community for verification and exploration. Even so, we have selected a large number of baselines and the conclusions we draw are also statistically significant.

Table S13: The 6 common fingerprints for benchmarking evaluation.

| No. | Baselines | Descriptions |
|---|---|---|
| 1 | mcfp4_2048 | MCFP4 (Morgan Connectivity Fingerprints) with dimension 2048 and radius 2, which is a variant of ECFP4 based on the Morgan algorithm and is a classical approach for generating circular fingerprints that focus on capturing connectivity patterns in molecules |
| 2 | ecfp4_2048 | ECFP4 (Extended Connectivity Fingerprints) with dimension 2048 and radius 2, which is a circular topological fingerprint that encodes information about a molecule's atomic connectivity and its local environment |
| 3 | maccs | 166-dimensional MACCS (Molecular ACCess System) keys are a type of molecular fingerprint that is widely used in cheminformatics to represent molecular structures |
| 4 | physchem | physchem (physicochemical) fingerprints is based on molecular physicochemical properties |
| 5 | atompair_2048 | 2048-dimensional AtomPair fingerprints, which represents molecules based on pairs of atoms and the shortest path (in bonds) between them, encoding both the atom types and their relative positions within a molecule |
| 6 | rdkDes | rdkDsc (RDKit Descriptors) are a set of predefined molecular descriptors calculated using the RDKit cheminformatics toolkit |

Table S14: The sequence-based baselines for benchmarking evaluation.

| No. | Baselines | Descriptions |
|---|---|---|
| 1 | BERT-6L | 6-layer BERT model, which comes from Chem-BERT with random initialization |
| 2 | BERT-8L/BERT-8L-R | 8-layer BERT model, which comes from Chem-BERT with random initialization |
| 3 | RoBERTa-12L | 12-layer RoBERTa model, which comes from Chem-RoBERTa with random initialization |
| 4 | molformer-R/MolFormer-R | MolFormer with random initialization |
| 5 | Chem-BERT-6L | 6-layer Chem-BERT |
| 6 | Chem-BERT-8L | 8-layer Chem-BERT |
| 7 | CHEM-RoBERTa-12L | 12-layer CHEM-RoBERTa |
| 8 | Molformer/MolFormer | MolFormer |

We describe the selected baselines for different modalities below:

- **Fingerprint.** Table S13 shows 6 commonly used molecular fingerprinting methods.

- **Sequence.** The baselines here use molecular sequences as input. As shown in Table S14, we use 8 common sequence-based models for benchmarking evaluation.

- **Graph.** The baselines here use molecular graphs as input. The Table S15 shows 9 graph-based baselines for benchmarking evaluation.

- **Geometry Graph.** The baselines here use molecular geometry graph as input. In the paper, the word "geometry" alone refers to the geometry graph. The Table S16 shows 9 geometry-based baselines for benchmarking evaluation.

- **Image.** The baselines here use molecular images as input. The Table S17 shows 5 image-based baselines for benchmarking evaluation.

- **Geometry Image.** The baselines here use molecular geometry images as input. The Table S18 shows 6 geometry image-based baselines for benchmarking evaluation.

- **Video.** The baselines here use molecular videos as input. The Table S19 shows 4 video-based baselines for benchmarking evaluation.

E.2 HYPERPARAMETERS SEARCH

In molecular property prediction, we use a linear probing strategy to train all models. To be fair, the hyperparameter search ranges for all models are the same. We set the batch size to 32 and perform grid search on learning rates of 0.001, 0.005, 0.01, and 0.05. In the MBANet task, all models use the same hyperparameters (batch size of 8 and learning rate of 0.005) to train the models. In StructNet, we set the batch size to 8 and perform grid search on learning rates of 0.001, 0.005.

To eliminate the influence of randomness, we used a uniform set of 10 random seeds ranging from 0 to 9 to calculate the mean and variance in all experiments in this paper.

Table S15: The 9 graph-based baselines for benchmarking evaluation.

| No. | Baselines | Descriptions |
|---|---|---|
| 1 | GIN-R | GIN model with random initialization |
| 2 | EdgePred | |
| 3 | ContextPred | |
| 4 | infomax | |
| 5 | masking | |
| 6 | MolCLR | - |
| 7 | MoleBERT | |
| 8 | GraphMVP | |
| 9 | CGIP-Graph | |

Table S16: The 9 geometry-based baselines for benchmarking evaluation.

| No. | Baselines | Descriptions |
|---|---|---|
| 1 | Uni-Mol-R (1 conf) | Uni-Mol with random initialization, which uses 1 conformation during fine-tuning |
| 2 | Uni-Mol (1 conf) | pre-trained Uni-Mol, which uses 1 conformation during fine-tuning |
| 3 | Uni-Mol-R (10 conf) | Uni-Mol with random initialization, which uses 10 conformations during fine-tuning |
| 4 | Uni-Mol (10 conf) | pre-trained Uni-Mol, which uses 10 conformations during fine-tuning |
| 5 | SchNet | |
| 6 | EGNN | |
| 7 | TFN | - |
| 8 | SE3_Transformer | |
| 9 | PaiNN | |

### E.3 TRAINING LOSS

In MoleculeNet benchmark, there are 8 classification tasks and 4 regression tasks. For classification tasks, we use cross entropy loss. For regression tasks, we use MSE loss. In MBANet and StructNet benchmarks, we use MSE loss because they are regression tasks.

## F  MORE RESULTS ON MOLECULENET

### F.1  DETAILED RESULTS

Table S20 and Table S21 show the ROC-AUC performance on 8 classification tasks and RMSE performance on 4 regression tasks from MoleculeNet, respectively. In classification tasks, as shown in Table S20, we find that fingerprint-based atompair_2048 achieves the best performance on Sider and BACE, which shows that fingerprinting is a simple and effective method for property prediction. The sequence-based MolFormer-R achieves state-of-the-art performance on BBBP and ClinTox without any pre-training, which indicates that sequence have a strong inductive bias for molecular property prediction. The graph-based MoleBERT, geometry-based Uni-Mol (10 conf), and geometry image-based IEM-G (10 conf) achieved the best performance on Tox21, ToxCast and MUV, HIV, respectively. These findings suggest that different modalities have certain preferences for different tasks, which can further establish guiding ideas for multi-modal learning of molecules.

In addition, in Table S21, we find the superiority of fingerprints on the regression task of property prediction, which achieves the best performance on 3 (Lipo, Malaria and CEP) out of 4 datasets.

### F.2  RESULTS OF GEOMETRY IMAGE AND VIDEO WITH RGB FORMAT

Images are available in RGB and BGR formats, and it is meaningful to study the difference between RGB and BGR for images. Here, we report the results using RGB format images as input. Table S22 and Table S23 show the results of 6 vision-based methods on 8 classification tasks and 4 regression tasks from MoleculeNet using RGB-format images. We find that using RGB or BGR as the input of the visual modality has little impact on performance.

Table S17: The 5 image-based baselines for benchmarking evaluation.

| No. | Baselines | Descriptions |
|---|---|---|
| 1 | ResNet18-R | image-based ResNet18 without pre-training |
| 2 | ImageMol | |
| 3 | MaskMol | |
| 4 | CGIP-Image | - |
| 5 | IEM -I | |

Table S18: The 6 geometry image-based baselines for benchmarking evaluation.

| No. | Baselines | Descriptions |
|---|---|---|
| 1 | ResNet18-G-R | geometry image-based ResNet18 without pre-training |
| 2 | IEM-G | IEM using geometry images as input |
| 3 | ResNet18-G-R (10 conf) | geometry image-based ResNet18 without pre-training, which uses 10 conformations during fine-tuning |
| 4 | IEM-G (10 conf) | IEM using geometry images as input, which uses 10 conformations during fine-tuning |
| 5 | ViT-G-R | ViT without pre-training, which is the bachbone of VideoMol and uses geometry image as input |
| 6 | VideoMol-G | pre-trained VideoMol, which uses geometry image as input |

## F.3   Results of Fine-tuning

Fine-tuning is a common strategy to maximize the performance of pre-trained models on downstream tasks. To compare the performance difference between linear probing and fine-tuning, we fine-tune Uni-Mol, Molformer, ImageMol and VideoMol using their public source code. Specifically, we use the officially released no-H pre-trained weight and the corresponding optimal hyperparameters to fine-tune Uni-Mol [1].

The Table S24 shows the fine-tuning performance of Uni-Mol, Molformer, ImageMol, and VideoMol on 6 classification tasks (Tox21, ToxCast, Sider, HIV, BBBP and BACE) from MoleculeNet. Overall, except for Molformer, fine-tuning on other methods helps improve the performance compared to linear probing. In particular, we find that after fine-tuning, the image modality improves performance by 14.98% compared to linear probing. The significant performance improvement indicates that the image modality currently relies on detailed fine-tuning to further improve performance.

## G   More Results on MBANet

### G.1   Atom

Table S25 shows the average RMSE performance of a large number of baselines on MBANet$_{atom}$ for 12 atom distribution prediction tasks (C, N, O, F, S, Cl, Br, P, Si, B, Se and Ge) with 10 seeds. Table S26 shows the corresponding standard deviation. We find that IEM-V based on video modality achieves the best performance on half of the atom prediction tasks, which shows the advantages of video-based modality. At the same time, we also find that models based on images and geometry images also achieve good performance, such as CGIP-Image and IEM-G (1 conf), indicating that the model can accurately count the number of atoms from the image. Furthermore, we observe that sequence- and graph-based models perform poorly on MBANet$_{atom}$, indicating that the global representations extracted by sequence- and graph-based models are not conducive to atomic-level prediction tasks.

### G.2   bond

Table S27 shows the average RMSE performance of a large number of baselines on MBANet$_{atom}$ for 4 bond distribution prediction tasks (single bond, aromatic bond, double bond, and triple bond) with 10 seeds. We find that the geometry-based TFN model has strong predictive power at the bond level, which suggests that utilizing both molecular geometry and bond information can effectively improve the model's understanding of bond distribution compared to graph models that only use

---

[1] https://github.com/deepmodeling/Uni-Mol/tree/main/unimol#molecular-property-prediction

Table S19: The 4 video-based baselines for benchmarking evaluation.

| No. | Baselines | Descriptions |
|---|---|---|
| 1 | ResNet18-V-R | video-based ResNet18 without pre-training |
| 2 | IEM-V | IEM using video as input |
| 3 | VideoMol-R | VideoMol without pre-training |
| 4 | VideoMol | pre-trained VideoMol, which uses video as input |

Table S20: The ROC-AUC (%) performance on 8 classification tasks with linear probing using BenchMol. The modality types from top to bottom are fingerprint, sequence, graph, geometry graph, image, geometry image, and video. -R means no pre-training, L means the number of layers, -I, -G, and -V mean the modalities are image, geometry image, and video, respectively. Note that the geometry images and videos use the BGR format.

| Model | BBBP | Tox21 | ToxCast | Sider | ClinTox | MUV | HIV | BACE | Avg |
|---|---|---|---|---|---|---|---|---|---|
| mcfp4_2048 | 64.9±0.3 | 73.5±0.1 | 61.9±0.0 | 65.0±0.1 | 83.0±0.2 | 73.4±0.3 | 73.8±0.3 | 77.8±1.0 | 71.66 |
| ecfp4_2048 | 63.9±0.1 | 71.2±0.1 | 59.7±0.1 | 64.2±0.0 | 76.9±0.1 | 70.1±0.1 | 72.0±0.1 | 80.5±0.6 | 69.81 |
| maccs | 66.6±0.3 | 70.1±0.0 | 61.7±0.0 | 63.8±0.1 | 85.1±0.2 | 71.9±0.5 | 70.7±1.5 | 74.4±1.2 | 70.54 |
| physchem | 61.8±0.1 | 62.6±0.1 | 59.4±0.1 | 57.7±0.9 | 66.4±5.7 | 69.6±0.1 | 66.7±0.0 | 61.1±2.5 | 63.16 |
| atompair_2048 | 65.5±0.3 | 73.1±0.3 | 64.8±0.1 | **66.2±0.0** | 63.9±0.4 | 70.4±0.2 | 74.8±0.2 | **83.5±1.3** | 70.28 |
| rdkDes | 59.3±0.5 | 64.1±0.2 | 60.5±0.2 | 54.0±0.7 | 56.2±0.0 | 66.1±1.1 | 68.8±0.0 | 76.5±0.0 | 63.19 |
| BERT-8L-R | 68.7±0.9 | 72.0±0.3 | 62.1±0.2 | 56.6±0.5 | 82.6±1.4 | 64.0±1.8 | 71.1±1.6 | 74.4±1.5 | 68.94 |
| Chem-BERT-8L | 69.2±0.2 | 75.5±0.1 | 62.6±0.1 | 62.6±0.1 | 83.3±0.6 | 77.6±1.2 | 78.2±0.2 | 78.3±0.3 | 73.41 |
| MolFormer-R | **74.6±0.5** | 71.6±0.3 | 61.5±0.3 | 55.9±0.3 | **86.2±0.3** | 67.2±1.6 | 71.2±0.5 | 75.0±1.6 | 70.40 |
| MolFormer | 63.3±0.2 | 72.1±0.1 | 61.4±0.1 | 63.4±0.2 | 68.2±3.3 | 75.4±0.8 | 74.5±0.4 | 78.3±1.8 | 69.58 |
| GIN-R | 57.3±0.3 | 69.3±0.3 | 58.3±0.4 | 56.2±0.8 | 64.9±0.4 | 69.6±1.1 | 66.9±1.2 | 63.2±0.3 | 63.21 |
| EdgePred | 52.1±1.0 | 67.1±0.3 | 56.4±0.0 | 54.5±0.6 | 55.0±2.7 | 65.8±0.2 | 67.6±0.5 | 67.3±2.8 | 60.73 |
| ContextPred | 57.6±0.3 | 70.6±0.1 | 60.7±0.0 | 60.8±1.0 | 58.6±3.2 | 76.4±0.3 | 72.4±1.3 | 78.1±0.2 | 66.90 |
| infomax | 62.4±0.0 | 68.6±0.2 | 59.2±0.1 | 58.5±0.7 | 60.1±2.2 | 76.4±0.3 | 71.9±0.3 | 73.7±1.0 | 66.35 |
| masking | 57.9±0.4 | 68.9±0.2 | 58.2±0.0 | 58.8±0.2 | 52.2±0.4 | 70.7±1.4 | 65.5±0.5 | 68.0±0.2 | 62.53 |
| MolCLR | 63.5±0.1 | 69.0±0.0 | 61.4±0.1 | 58.6±0.3 | 64.2±1.0 | 65.0±1.4 | 72.1±0.3 | 70.2±0.2 | 65.50 |
| MoleBERT | 66.3±0.1 | **77.1±0.1** | 65.0±0.1 | 63.9±0.1 | 74.8±3.4 | 79.7±0.2 | 76.2±0.1 | 75.2±0.9 | 72.28 |
| GraphMVP | 64.2±0.4 | 69.5±0.1 | 60.6±0.1 | 58.6±0.1 | 56.7±1.5 | 68.6±0.4 | 71.6±0.4 | 76.4±0.1 | 65.78 |
| CGIP-Graph | 66.4±0.9 | 71.5±0.3 | 58.6±0.0 | 57.5±0.1 | 70.3±0.2 | 72.5±3.4 | 73.6±0.8 | 66.0±3.7 | 67.05 |
| Uni-Mol-R | 66.3±1.0 | 67.6±0.1 | 62.4±0.1 | 59.2±0.1 | 59.2±0.4 | 59.5±1.6 | 74.7±0.2 | 77.5±0.7 | 65.80 |
| Uni-Mol | 69.8±0.1 | 74.5±0.1 | 65.6±0.4 | 60.2±0.4 | 84.3±0.4 | 78.5±0.2 | 78.4±0.1 | 78.7±0.5 | 73.75 |
| Uni-Mol-R (10 conf) | 64.6±0.4 | 67.7±0.1 | 63.1±0.1 | 59.3±1.2 | 58.6±0.2 | 56.5±1.8 | 77.2±0.1 | 77.2±0.1 | 65.53 |
| Uni-Mol (10 conf) | 69.3±0.6 | 75.2±0.1 | **65.8±0.5** | 61.6±0.4 | 85.1±4.4 | **80.3±0.7** | 77.5±0.1 | 78.2±0.2 | **74.13** |
| ResNet18-R | 52.4±0.1 | 53.8±0.1 | 54.7±0.0 | 55.8±0.1 | 65.3±0.2 | 49.8±1.7 | 52.8±0.2 | 58.1±2.2 | 55.34 |
| ImageMol | 60.5±0.5 | 66.4±0.3 | 59.0±0.3 | 58.2±0.2 | 64.5±2.6 | 61.3±1.3 | 70.8±0.9 | 60.3±1.0 | 62.63 |
| MaskMol | 62.3±1.1 | 65.9±0.1 | 60.1±0.1 | 59.1±0.7 | 56.4±0.7 | 58.8±3.2 | 74.4±0.6 | 67.2±1.2 | 63.03 |
| CGIP-Image | 56.2±0.3 | 66.0±0.0 | 55.6±0.0 | 57.2±0.3 | 68.0±0.3 | 63.1±0.0 | 69.5±0.1 | 59.9±0.3 | 61.94 |
| IEM -I | 59.7±0.1 | 65.8±0.2 | 57.1±0.1 | 56.8±0.9 | 57.8±3.6 | 56.8±1.0 | 72.2±0.5 | 61.4±0.6 | 60.95 |
| ResNet18-G-R | 55.2±0.0 | 58.2±0.2 | 57.4±0.2 | 55.7±0.1 | 60.0±0.4 | 54.3±3.7 | 71.5±0.4 | 57.5±1.9 | 58.73 |
| IEM-G | 64.1±0.2 | 69.0±0.3 | 60.9±0.2 | 56.0±1.0 | 55.8±0.8 | 60.8±2.0 | 75.5±0.7 | 75.2±0.6 | 64.66 |
| ResNet18-G-R (10 conf) | 60.1±0.0 | 62.3±0.1 | 59.8±0.1 | 57.1±0.4 | 67.5±0.1 | 57.0±0.3 | 74.3±0.0 | 71.3±1.3 | 63.68 |
| IEM-G (10 conf) | 68.1±0.4 | 71.9±0.2 | 63.9±0.1 | 61.5±0.8 | 68.6±1.0 | 68.7±1.1 | **80.0±0.4** | 79.6±1.2 | 70.29 |
| VideoMol-R | 59.7±0.2 | 61.3±0.1 | 59.2±0.4 | 55.2±1.4 | 60.0±0.1 | 53.1±1.5 | 75.4±0.4 | 65.4±2.1 | 61.16 |
| VideoMol | 63.4±0.7 | 73.5±0.1 | 63.9±0.2 | 60.3±0.3 | 63.8±2.9 | 75.3±0.9 | 76.8±1.2 | 75.2±0.4 | 69.03 |

bond information. In addition, compared with SE3-Transformer and Uni-Mol, TFN is more suitable for capturing local bond information.

## G.3 BASIC ATTRIBUTES

The Table S28 shows the average RMSE performance of a large number of baselines on MBANet$_{attr}$ for 8 basic attribute prediction tasks (MW, LogP, MR, BalabanJ, #HA, #HD, #VE, TPSA) with 10 seeds. The Table S29 shows the corresponding standard deviation. We find that graph modality achieves the best performance on LogP, BalabanJ, HA and HD and video modality achieves the best performance on MW, MR, VE and TPSA. In particular, IEM-V equipped with video modality achieves the best average performance, which is 54.2% higher than IEM-G using geometry image modality (from 4.916 to 2.254), indicating the superiority of combining IEM with video modality.

Table S21: The RMSE performance on 4 regression tasks with linear probing using BenchMol. Note that the geometry images and videos use the BGR format.

| Model | ESOL | Lipo | Malaria | CEP | Avg |
|---|---|---|---|---|---|
| mcfp4_2048 | 1.510±0.017 | 0.833±0.001 | 1.096±0.001 | **1.582±0.001** | 1.255 |
| ecfp4_2048 | 1.658±0.019 | 0.902±0.001 | 1.088±0.001 | 1.551±0.000 | 1.300 |
| maccs | 1.339±0.014 | 0.980±0.002 | 1.128±0.001 | 1.759±0.002 | 1.302 |
| physchem | 1.713±0.020 | 0.994±0.003 | 1.155±0.001 | 1.953±0.001 | 1.454 |
| atompair_2048 | 1.220±0.006 | **0.817±0.002** | **1.087±0.000** | 1.632±0.003 | 1.189 |
| rdkDes | 1.830±0.001 | 1.067±0.003 | 1.166±0.002 | 2.529±0.001 | 1.648 |
| BERT-8L-R | 1.102±0.041 | 1.005±0.005 | 1.161±0.005 | 1.786±0.008 | 1.264 |
| Chem-BERT-8L | **0.858±0.013** | 0.823±0.003 | 1.106±0.002 | 1.584±0.006 | **1.093** |
| MolFormer-R | 1.278±0.007 | 0.994±0.003 | 1.157±0.005 | 1.845±0.001 | 1.319 |
| MolFormer | 1.350±0.016 | 0.936±0.004 | 1.123±0.004 | 1.764±0.006 | 1.293 |
| GIN-R | 1.780±0.013 | 1.078±0.004 | 1.148±0.002 | 2.299±0.004 | 1.576 |
| EdgePred | 2.396±0.019 | 1.075±0.001 | 1.134±0.002 | 2.053±0.006 | 1.665 |
| ContextPred | 1.520±0.010 | 1.031±0.006 | 1.129±0.003 | 2.165±0.007 | 1.461 |
| infomax | 1.450±0.012 | 1.035±0.007 | 1.131±0.004 | 2.018±0.012 | 1.409 |
| masking | 1.696±0.011 | 1.065±0.002 | 1.130±0.006 | 2.070±0.004 | 1.490 |
| MolCLR | 1.506±0.013 | 0.931±0.006 | 1.114±0.003 | 1.925±0.005 | 1.369 |
| MoleBERT | 1.544±0.006 | 0.897±0.001 | 1.105±0.001 | 1.735±0.004 | 1.320 |
| GraphMVP | 1.623±0.008 | 0.959±0.012 | 1.143±0.002 | 1.879±0.009 | 1.401 |
| CGIP-Graph | 2.494±0.020 | 0.903±0.006 | 1.113±0.004 | 1.696±0.009 | 1.552 |
| Uni-Mol-R | 1.048±0.019 | 0.999±0.004 | 1.146±0.002 | 2.038±0.010 | 1.308 |
| Uni-Mol | 1.003±0.005 | 0.856±0.004 | 1.113±0.001 | 1.676±0.004 | 1.162 |
| Uni-Mol-R (10 conf) | 0.997±0.017 | 0.984±0.008 | 1.149±0.002 | 1.974±0.006 | 1.276 |
| Uni-Mol (10 conf) | 0.978±0.005 | 0.839±0.004 | 1.109±0.002 | 1.648±0.008 | 1.144 |
| ResNet18-R | 1.917±0.004 | 1.108±0.002 | 1.166±0.001 | 2.535±0.000 | 1.682 |
| ImageMol | 1.655±0.021 | 1.053±0.008 | 1.150±0.008 | 2.169±0.003 | 1.507 |
| MaskMol | 1.329±0.034 | 1.056±0.005 | 1.160±0.004 | 2.219±0.006 | 1.441 |
| CGIP-Image | 1.710±0.023 | 1.078±0.001 | 1.149±0.001 | 2.287±0.001 | 1.556 |
| IEM -I | 1.730±0.028 | 1.057±0.005 | 1.156±0.001 | 2.364±0.002 | 1.577 |
| ResNet18-G-R | 1.561±0.015 | 1.073±0.002 | 1.164±0.001 | 2.472±0.001 | 1.568 |
| IEM-G | 1.313±0.008 | 0.974±0.003 | 1.155±0.001 | 2.180±0.001 | 1.406 |
| ResNet18-G-R (10 conf) | 1.359±0.015 | 1.055±0.003 | 1.161±0.002 | 2.364±0.001 | 1.485 |
| IEM-G (10 conf) | 0.936±0.023 | 0.887±0.006 | 1.155±0.001 | 1.868±0.007 | 1.212 |
| VideoMol-R | 1.520±0.017 | 1.058±0.006 | 1.163±0.001 | 2.364±0.003 | 1.526 |
| VideoMol | 1.085±0.011 | 0.887±0.003 | 1.137±0.004 | 1.780±0.003 | 1.222 |

Table S22: The ROC-AUC (%) performance on 8 classification tasks from MoleculeNet with RGB format with linear probing using BenchMol. The first 4 are geometry image-based methods, and the last 2 are video-based methods.

| Model | BBBP | Tox21 | ToxCast | Sider | ClinTox | MUV | HIV | BACE | Avg |
|---|---|---|---|---|---|---|---|---|---|
| ResNet18-G-R | 56.2±0.2 | 59.9±0.1 | 57.1±0.1 | 53.8±0.1 | 59.2±0.5 | 57.9±0.8 | 72.4±0.8 | 63.3±1.4 | 59.98 |
| IEM-G | 64.5±0.6 | 69.5±0.4 | 61.3±0.3 | 56.1±1.0 | 49.7±0.7 | 61.8±0.7 | 75.2±0.8 | 76.5±0.2 | 64.33 |
| ResNet18-G-R (10 conf) | 58.7±0.0 | 62.8±0.1 | 58.5±0.1 | 56.3±0.1 | **68.0±0.1** | 57.3±0.6 | 74.8±0.6 | 69.2±0.6 | 63.20 |
| IEM-G (10 conf) | 65.7±1.3 | 73.2±0.2 | **63.8±0.2** | 59.8±0.2 | 58.3±0.2 | 68.8±0.7 | **78.5±0.0** | **80.3±0.1** | **68.55** |
| VideoMol-R | 60.6±0.3 | 60.6±0.1 | 58.8±0.4 | 57.2±0.2 | 59.1±0.0 | 53.9±1.7 | 76.7±0.4 | 61.0±1.9 | 60.99 |
| VideoMol | **66.5±0.2** | **73.7±0.2** | 63.2±0.2 | **61.8±0.2** | 57.2±2.8 | **74.0±0.5** | 75.2±0.5 | 76.6±0.8 | 68.53 |

# H   EXPANDING MBANET TO 30,000 MOLECULES

In order to study the generalization of MBANet's conclusions, we further expand MBANet to 30,000 molecules, referred to as MBANet$^{30K}$. As shown in Table S30, we find that the results of MBANet$^{30K}$ are not significantly different from the results on the original MBANet (see Table 5). For example, the video modality still achieves the best performance in tasks related to atoms and basic attributes. Therefore, the conclusion of MBANet is effective after further expanding the scale of MBANet to 30K.

Table S23: The RMSE performance on 4 regression tasks with RGB format with linear probing using BenchMol.

| Model | ESOL | Lipo | Malaria | CEP | Avg |
|---|---|---|---|---|---|
| ResNet18-G-R | 1.715±0.007 | 1.080±0.002 | 1.163±0.001 | 2.469±0.000 | 1.607 |
| IEM-G | 1.194±0.007 | 0.981±0.001 | 1.170±0.001 | 2.130±0.002 | 1.369 |
| ResNet18-G-R (10 conf) | 1.395±0.013 | 1.048±0.005 | 1.163±0.001 | 2.392±0.002 | 1.500 |
| IEM-G (10 conf) | **0.968±0.017** | 0.893±0.004 | 1.132±0.005 | 1.874±0.004 | 1.217 |
| VideoMol-R | 1.515±0.025 | 1.050±0.009 | 1.165±0.001 | 2.337±0.003 | 1.517 |
| VideoMol | 1.035±0.005 | **0.867±0.005** | **1.099±0.004** | **1.760±0.006** | **1.190** |

Table S24: The ROC-AUC (%) performance on 6 classification tasks from MoleculeNet with fine-tuning setting under 10 random seeds ranging from 0 to 9. We fine-tune these models using their public code. Specifically, we evaluate Uni-Mol using the officially released no-H pre-trained weight and the corresponding optimal hyperparameters. FT denotes the average performance of fine-tuning on 6 datasets. LP denotes the average result of linear probing, which is obtained by Table S20. $\delta$ denotes $(\frac{FT}{LP} - 1) * 100\%$. Note that IEM and VideoMol use the BGR format.

| | Tox21 | ToxCast | Sider | HIV | BBBP | BACE | FT | LP | $\delta$ |
|---|---|---|---|---|---|---|---|---|---|
| Uni-Mol (1 conf) | 78.3 (0.4) | 68.7 (0.5) | 63.7 (1.3) | 79.2 (1.0) | 69.6 (2.0) | 81.0 (3.9) | 73.4 | 71.2 | ↑ 3.11% |
| Uni-Mol (10 conf) | 78.8 (0.7) | 69.0 (0.5) | 63.6 (1.4) | 79.2 (0.9) | 69.9 (2.7) | 81.7 (3.4) | 73.7 | 71.3 | ↑ 3.41% |
| Molformer | 47.4 (2.1) | 56.2 (1.5) | 61.1 (1.0) | 74.6 (0.9) | 69.5 (1.0) | 80.9 (1.9) | 65.0 | **69.6** | ↓ 6.65% |
| ImageMol | 75.5 (1.0) | 65.6 (0.9) | 64.9 (1.3) | 76.8 (1.3) | 70.5 (1.3) | 78.1 (3.5) | 71.9 | 62.5 | ↑ **14.98%** |
| VideoMol | **78.8 (0.5)** | **66.7 (0.5)** | **66.3 (0.9)** | **79.4 (0.5)** | **70.7 (2.2)** | **82.4 (0.9)** | **74.1** | 69.0 | ↑ 7.27% |

# I  MORE RESULTS ON STRUCTNET

## I.1  ACYCLIC CHAIN MOLECULES

Table S31 and Table S32 show the average RMSE performance and corresponding standard deviation of the baselines on 10 acyclic chain datasets, respectively. Overall, each modality has baselines that make it into the top 5 in terms of performance. From the average performance, we find the effectiveness of graph pre-training strategies on acyclic chain molecules because 4 (MoleBERT, infomax, MolCLR and masking) of the top 5 methods are graph-based pre-training methods. We also find that even without pre-training, SchNet can still achieve good performance on acyclic chain molecules, ranking second, which demonstrates the effectiveness of geometric methods on this type of molecules. Furthermore, we find that the non-pre-trained vision-based ResNet18-G-R and ResNet18-V-R also achieve top-5 performance, which indicates the effectiveness of these visual representations.

## I.2  ACYCLIC MOLECULES

Table S33 and Table S34 show the average RMSE performance and corresponding standard deviation of the baselines on 10 acyclic datasets, respectively. Here, we find the effectiveness of graph pre-training methods because the top 5 methods on performance are all based on graph pre-training methods, such as MoleBERT, ContextPred, masking, CGIP-Graph and infomax. When no pre-training is performed, the geometry-based TFN model achieves the best performance, demonstrating the advantage of geometric methods on acyclic molecules.

## I.3  COMPLETE CHAIN MOLECULES

Table S35 and Table S36 show the average RMSE performance and corresponding standard deviation of the baselines on 10 complete chain datasets, respectively. We find that molecular fingerprint-based maccs achieves the best average performance, surpassing a number of pre-training methods, demonstrating the advantages of maccs on complete chain molecules.

Notably, we observe that a large number of graph-based and image-based pre-training strategies fail on this type of molecules. Specifically, 6 out of 8 graph-based pre-training methods (EdgePred, ContextPred, infomax, masking, MoleBERT and CGIP-Graph) and all 4 image-based pre-training

Table S25: The average RMSE performance on MBANet$_{atom}$ with 10 seeds. The modality types from top to bottom are sequence, graph, geometry graph, image, geometry image, and video. -R means no pre-training, L means the number of layers, -I, -G, and -V mean the modalities are image, geometry image, and video, respectively. The green background represents top-5 performance.

| Models | #C | #N | #O | #F | #S | #Cl | #Br | #P | #Si | #B | #Se | #Ge | Mean |
|---|---|---|---|---|---|---|---|---|---|---|---|---|---|
| BERT-6L | 2.001 | 0.810 | 1.460 | 0.742 | 0.322 | 0.422 | 0.206 | **0.014** | 0.023 | 0.237 | 0.012 | 0.009 | 0.522 |
| BERT-8L | 2.011 | 0.807 | 1.479 | 0.727 | 0.321 | 0.420 | 0.210 | 0.027 | 0.022 | 0.238 | 0.014 | 0.015 | 0.524 |
| RoBERTa-12L | 2.309 | 0.806 | 1.463 | 0.722 | 0.319 | 0.426 | 0.196 | 0.022 | 0.035 | 0.233 | 0.010 | 0.012 | 0.546 |
| molformer-R | 2.288 | 0.819 | 1.514 | 0.702 | 0.321 | 0.423 | 0.204 | 0.025 | 0.036 | 0.241 | 0.019 | 0.021 | 0.551 |
| Chem-BERT-6L | 1.985 | 0.801 | 1.453 | 0.730 | 0.320 | 0.425 | 0.205 | 0.019 | 0.031 | 0.247 | 0.013 | 0.026 | 0.521 |
| Chem-BERT-8L | 2.002 | 0.805 | 1.491 | 0.722 | 0.323 | 0.422 | 0.197 | 0.024 | 0.022 | 0.239 | 0.023 | 0.013 | 0.524 |
| CHEM-RoBERTa-12L | 2.186 | 0.807 | 1.468 | 0.708 | 0.319 | 0.430 | 0.200 | 0.017 | 0.024 | 0.234 | 0.010 | 0.007 | 0.534 |
| Molformer | 2.121 | 0.798 | 1.448 | 0.702 | 0.322 | 0.423 | 0.201 | 0.022 | 0.034 | 0.238 | 0.035 | 0.023 | 0.531 |
| GIN-R | 1.445 | 0.520 | 0.376 | 0.392 | 0.459 | 0.366 | 0.195 | 0.035 | 0.042 | 0.234 | 0.009 | 0.002 | 0.340 |
| EdgePred | 1.396 | 0.586 | 0.354 | 0.442 | 0.197 | 0.300 | 0.189 | 0.036 | 0.045 | 0.234 | 0.011 | 0.002 | 0.316 |
| ContextPred | 1.353 | 0.532 | 0.366 | 0.406 | 0.283 | 0.315 | 0.193 | 0.058 | 0.031 | 0.234 | 0.031 | 0.006 | 0.317 |
| infomax | 1.427 | 0.593 | 0.372 | 0.418 | 0.211 | 0.296 | 0.188 | 0.036 | 0.027 | 0.235 | 0.012 | 0.002 | 0.318 |
| masking | 1.481 | 0.529 | 0.412 | 0.406 | 0.337 | 0.305 | 0.184 | 0.034 | 0.035 | 0.233 | 0.024 | 0.004 | 0.332 |
| MolCLR | 1.375 | 0.533 | 0.395 | 0.344 | 0.395 | 0.355 | 0.187 | 0.034 | 0.037 | 0.233 | 0.012 | **0.001** | 0.325 |
| MoleBERT | 1.552 | 0.487 | 0.378 | 0.380 | 0.197 | 0.260 | 0.191 | 0.045 | 0.044 | 0.233 | 0.019 | 0.016 | 0.317 |
| CGIP-Graph | 1.165 | 0.310 | 0.318 | 0.207 | 0.115 | 0.127 | 0.190 | 0.041 | 0.037 | 0.233 | 0.017 | 0.008 | 0.231 |
| GraphMVP | 1.324 | 0.519 | 0.390 | 0.428 | 0.452 | 0.325 | 0.192 | 0.030 | 0.031 | 0.236 | 0.008 | 0.004 | 0.328 |
| Uni-Mol-R (1 conf) | 1.436 | 0.713 | 0.741 | 0.352 | 0.143 | 0.161 | 0.168 | 0.061 | 0.044 | 0.235 | 0.015 | 0.002 | 0.339 |
| Uni-Mol (1 conf) | 1.466 | 0.784 | 0.756 | 0.362 | 0.144 | 0.156 | 0.156 | 0.043 | 0.029 | 0.225 | 0.020 | 0.005 | 0.346 |
| TFN | 0.602 | 0.337 | 0.713 | 0.290 | 0.117 | 0.149 | **0.139** | 0.047 | 0.037 | 0.176 | 0.015 | 0.010 | 0.177 |
| SE3_Transformer | 1.773 | 0.849 | 1.475 | 0.720 | 0.312 | 0.410 | 0.195 | 0.019 | **0.020** | 0.234 | 0.010 | 0.015 | 0.503 |
| ResNet18-I-R | 1.332 | 0.608 | 0.747 | 0.473 | 0.210 | 0.321 | 0.194 | 0.047 | 0.029 | 0.234 | 0.007 | 0.004 | 0.350 |
| ImageMol | 1.340 | 0.578 | 0.750 | 0.485 | 0.225 | 0.321 | 0.194 | 0.054 | 0.035 | 0.232 | 0.007 | 0.002 | 0.352 |
| CGIP-Image | 1.251 | 0.581 | 0.462 | 0.462 | 0.224 | 0.305 | 0.196 | 0.040 | 0.025 | 0.234 | **0.005** | 0.005 | 0.337 |
| MaskMol | 2.096 | 0.818 | 1.466 | 0.706 | 0.325 | 0.432 | 0.198 | 0.034 | 0.028 | 0.242 | 0.016 | 0.017 | 0.532 |
| IEM-I | 1.324 | 0.580 | 0.738 | 0.489 | 0.210 | 0.326 | 0.195 | 0.043 | 0.025 | 0.233 | 0.009 | 0.003 | 0.348 |
| ResNet18-G-R | 0.939 | 0.349 | 0.467 | 0.409 | 0.165 | 0.343 | 0.198 | 0.045 | 0.056 | 0.223 | 0.012 | 0.013 | 0.268 |
| IEM-G (1 conf) | 0.803 | 0.327 | 0.452 | 0.338 | 0.152 | 0.313 | 0.196 | 0.050 | 0.046 | 0.232 | 0.009 | 0.003 | 0.243 |
| ResNet18-V-R | 0.520 | 0.154 | 0.215 | 0.225 | 0.087 | 0.191 | 0.156 | 0.043 | 0.140 | **0.129** | 0.008 | **0.001** | 0.156 |
| IEM-V | **0.354** | **0.130** | **0.186** | **0.148** | **0.074** | **0.119** | 0.164 | 0.021 | 0.058 | 0.220 | 0.007 | 0.002 | **0.124** |

Table S26: The standard deviation on MBANet$_{atom}$ with 10 seeds. The modality types from top to bottom are sequence, graph, geometry graph, image, geometry image, and video. -R means no pre-training, L means the number of layers, -I, -G, and -V mean the modalities are image, geometry image, and video, respectively.

| Models | #C | #N | #O | #F | #S | #Cl | #Br | #P | #Si | #B | #Se | #Ge |
|---|---|---|---|---|---|---|---|---|---|---|---|---|
| BERT-6L | 0.216 | 0.035 | 0.082 | 0.058 | 0.007 | 0.015 | 0.026 | **0.010** | 0.012 | 0.009 | 0.011 | 0.006 |
| BERT-8L | 0.218 | 0.028 | 0.079 | 0.043 | 0.013 | **0.007** | 0.027 | 0.022 | 0.014 | 0.008 | 0.008 | 0.010 |
| RoBERTa-12L | 0.134 | 0.017 | 0.079 | 0.024 | 0.009 | 0.013 | 0.004 | 0.021 | 0.025 | 0.004 | 0.010 | 0.013 |
| molformer-R | 0.187 | 0.064 | 0.109 | 0.023 | 0.007 | 0.014 | 0.010 | 0.024 | 0.032 | 0.013 | 0.011 | 0.021 |
| Chem-BERT-6L | 0.213 | **0.012** | 0.079 | 0.051 | 0.008 | 0.017 | 0.017 | 0.014 | 0.024 | 0.014 | 0.008 | 0.017 |
| Chem-BERT-8L | 0.177 | 0.028 | 0.070 | 0.038 | 0.009 | 0.013 | 0.004 | 0.022 | 0.021 | 0.007 | 0.026 | 0.012 |
| CHEM-RoBERTa-12L | 0.188 | 0.018 | 0.091 | 0.028 | 0.011 | 0.018 | 0.004 | 0.016 | 0.022 | 0.004 | 0.012 | 0.006 |
| Molformer | 0.162 | **0.012** | 0.093 | **0.016** | 0.010 | 0.015 | 0.006 | 0.025 | 0.018 | 0.011 | 0.037 | 0.020 |
| GIN-R | 0.313 | 0.129 | 0.076 | 0.054 | 0.095 | 0.027 | 0.015 | 0.027 | 0.024 | 0.004 | 0.013 | 0.003 |
| EdgePred | 0.254 | 0.154 | 0.089 | 0.094 | 0.036 | 0.098 | 0.004 | 0.015 | 0.026 | 0.004 | 0.010 | 0.002 |
| ContextPred | 0.211 | 0.167 | 0.077 | 0.128 | 0.121 | 0.073 | 0.010 | 0.030 | 0.017 | 0.004 | 0.029 | 0.008 |
| infomax | 0.172 | 0.088 | 0.102 | 0.187 | 0.065 | 0.097 | 0.004 | 0.031 | 0.018 | 0.004 | 0.011 | 0.002 |
| masking | 0.362 | 0.167 | 0.099 | 0.139 | 0.139 | 0.078 | 0.006 | 0.022 | 0.023 | 0.004 | 0.018 | 0.006 |
| MolCLR | 0.196 | 0.143 | 0.076 | 0.085 | 0.079 | 0.032 | 0.006 | 0.015 | 0.014 | 0.004 | 0.014 | **0.001** |
| MoleBERT | 0.389 | 0.120 | 0.105 | 0.085 | 0.066 | 0.104 | 0.006 | 0.016 | 0.024 | **0.003** | 0.014 | 0.025 |
| CGIP-Graph | 0.175 | 0.055 | 0.046 | 0.040 | 0.035 | 0.036 | 0.008 | 0.020 | 0.016 | 0.009 | 0.010 | 0.005 |
| GraphMVP | 0.248 | 0.144 | 0.103 | 0.146 | 0.078 | 0.021 | 0.008 | 0.021 | 0.018 | 0.005 | 0.009 | 0.006 |
| Uni-Mol-R (1 conf) | 0.133 | 0.086 | 0.057 | 0.025 | 0.031 | 0.030 | 0.018 | 0.021 | 0.029 | 0.013 | 0.005 | 0.002 |
| Uni-Mol (1 conf) | 0.123 | 0.081 | 0.045 | 0.042 | 0.035 | 0.042 | 0.029 | 0.011 | 0.013 | 0.009 | 0.025 | 0.005 |
| TFN | 0.092 | 0.069 | 0.038 | 0.025 | 0.039 | 0.053 | 0.014 | **0.008** | 0.040 | 0.016 | 0.011 | |
| SE3_Transformer | 0.179 | 0.081 | 0.231 | 0.033 | **0.006** | 0.037 | 0.003 | 0.018 | 0.025 | 0.006 | 0.016 | 0.030 |
| ResNet18-I-R | 0.095 | 0.040 | 0.062 | 0.037 | 0.013 | 0.034 | **0.002** | 0.031 | 0.015 | 0.004 | **0.003** | 0.006 |
| ImageMol | 0.116 | 0.037 | 0.052 | 0.029 | 0.026 | 0.046 | 0.003 | 0.027 | 0.017 | **0.003** | 0.005 | 0.002 |
| CGIP-Image | 0.095 | 0.027 | 0.044 | 0.028 | 0.028 | 0.027 | **0.002** | 0.013 | 0.012 | **0.003** | 0.003 | 0.007 |
| MaskMol | 0.136 | 0.023 | 0.115 | 0.017 | 0.010 | 0.015 | 0.004 | 0.021 | 0.022 | 0.010 | 0.013 | 0.013 |
| IEM-I | 0.072 | 0.032 | 0.038 | 0.037 | 0.013 | 0.038 | 0.003 | 0.019 | 0.012 | 0.005 | 0.008 | 0.004 |
| ResNet18-G-R | 0.171 | 0.030 | 0.067 | 0.069 | 0.045 | 0.173 | 0.005 | 0.016 | 0.024 | 0.011 | 0.009 | 0.030 |
| IEM-G (1 conf) | 0.063 | 0.025 | 0.043 | 0.084 | 0.036 | 0.069 | 0.006 | 0.033 | 0.026 | 0.006 | 0.005 | 0.004 |
| ResNet18-V-R | 0.179 | 0.023 | **0.032** | 0.048 | 0.014 | 0.044 | 0.041 | 0.031 | 0.036 | 0.030 | 0.006 | **0.001** |
| IEM-V | **0.062** | 0.041 | 0.038 | 0.052 | 0.020 | 0.042 | 0.047 | 0.015 | 0.065 | 0.022 | 0.005 | 0.003 |

Table S27: The average RMSE (standard deviation) performance on MBANet$_{bond}$ with 10 seeds. The modality types from top to bottom are sequence, graph, geometry graph, image, geometry image, and video. -R means no pre-training, L means the number of layers, -I, -G, and -V mean the modalities are image, geometry image, and video, respectively. The green background represents top-5 performance.

| | Models | #SINGLE | #AROMATIC | #DOUBLE | #TRIPLE | Mean |
|---|---|---|---|---|---|---|
| | BERT-6L | 5.092±0.271 | 3.626±0.176 | 1.240±0.113 | 0.606±0.047 | 2.641 |
| | BERT-8L | 5.254±0.275 | 3.543±0.218 | 1.180±0.107 | 0.616±0.044 | 2.648 |
| | RoBERTa-12L | 5.520±0.313 | 3.652±0.190 | 1.210±0.142 | 0.628±0.075 | 2.753 |
| | molformer-R | 5.623±0.247 | 3.624±0.302 | 1.315±0.114 | 0.592±0.049 | 2.788 |
| SMILES | Chem-BERT-6L | 5.211±0.251 | 3.602±0.163 | 1.271±0.105 | 0.582±0.024 | 2.666 |
| | Chem-BERT-8L | 4.937±0.686 | 3.758±0.436 | 1.237±0.159 | 0.597±0.038 | 2.632 |
| | CHEM-RoBERTa-12L | 5.548±0.246 | 3.645±0.197 | 1.289±0.230 | 0.621±0.041 | 2.776 |
| | Molformer | 5.367±0.266 | 3.621±0.231 | 1.215±0.059 | 0.611±0.032 | 2.703 |
| | GIN-R | 0.988±0.099 | 1.002±0.124 | 0.273±0.060 | 0.145±0.032 | 0.602 |
| | EdgePred | 1.005±0.094 | 0.970±0.135 | 0.256±0.096 | 0.157±0.054 | 0.597 |
| | ContextPred | 1.001±0.236 | 0.965±0.161 | 0.190±0.044 | 0.155±0.045 | 0.578 |
| | infomax | 0.999±0.236 | 0.980±0.110 | 0.219±0.098 | 0.171±0.055 | 0.592 |
| Graph | masking | 0.994±0.158 | 0.967±0.103 | 0.233±0.065 | 0.178±0.064 | 0.593 |
| | MolCLR | 1.072±0.136 | 1.004±0.137 | 0.242±0.059 | 0.206±0.113 | 0.631 |
| | MoleBERT | 1.039±0.210 | 0.910±0.133 | 0.196±0.043 | 0.151±0.050 | 0.574 |
| | CGIP-Graph | 0.851±0.227 | 1.075±0.159 | 0.237±0.045 | 0.190±0.085 | 0.588 |
| | GraphMVP | 1.157±0.196 | 0.969±0.120 | 0.229±0.053 | 0.138±0.044 | 0.623 |
| | Uni-Mol-R (1 conf) | 4.156±0.212 | 3.140±0.203 | 1.151±0.160 | 0.606±0.053 | 2.263 |
| Geometry Graph | Uni-Mol (1 conf) | 4.195±0.263 | 3.203±0.265 | 1.184±0.166 | 0.600±0.039 | 2.296 |
| | TFN | 0.380±0.060 | 0.452±0.203 | 0.280±0.063 | 0.125±0.016 | 0.309 |
| | SE3_Transformer | 2.580±0.939 | 3.432±1.017 | 1.079±0.428 | 0.478±0.217 | 1.892 |
| | ResNet18-I-R | 2.508±0.234 | 2.515±0.192 | 1.010±0.070 | 0.486±0.038 | 1.630 |
| | ImageMol | 2.535±0.183 | 2.358±0.173 | 1.012±0.071 | 0.499±0.049 | 1.601 |
| Image | CGIP-Image | 2.503±0.283 | 2.324±0.153 | 1.025±0.079 | 0.510±0.044 | 1.591 |
| | MaskMol | 5.066±0.329 | 3.720±0.356 | 1.237±0.187 | 0.601±0.036 | 2.656 |
| | IEM-I | 2.655±0.232 | 2.423±0.178 | 1.059±0.078 | 0.543±0.111 | 1.670 |
| Geometry Image | ResNet18-G-R | 2.167±0.205 | 2.519±0.189 | 1.079±0.160 | 0.578±0.036 | 1.586 |
| | IEM-G (1 conf) | 2.236±0.224 | 2.402±0.102 | 1.008±0.093 | 0.591±0.030 | 1.560 |
| Video | ResNet18-V-R | 1.310±0.150 | 1.704±0.122 | 0.740±0.138 | 0.437±0.167 | 1.048 |
| | IEM-V | 1.194±0.157 | 1.596±0.138 | 0.650±0.069 | 0.421±0.131 | 0.965 |

methods (ImageMol, CGIP-Image, MaskMol and IEM-I) produce negative transfer in performance. This shows that for complete chain molecules, we need to further design more suitable pre-training tasks to improve the performance of the model on this type of molecules.

## I.4 MACRO MOLECULES

Table S37 and Table S38 show the average RMSE performance and corresponding standard deviation of the baselines on 10 macro-molecule datasets, respectively. In general, the graph-based methods show great advantages on macromolecules because the green areas are concentrated in the graph-based methods. At the same time, we observe that all molecules in the dataset have rings. Therefore, this indicates that graph-based models are suitable for macro molecules with rings. Furthermore, we find that molecular fingerprints are the second best modality overall compared to the graph modality, indicating that fingerprints are a good alternative for macro molecules with rings. We find that half of the graph-based methods have the problem of negative transfer on macro molecules, such as EdgePred, infomax, masking, MoleBert, which deserves further study in the future.

## I.5 MACROCYCLIC PEPTIDE MOLECULES

Table S39 and Table S40 show the average RMSE performance and corresponding standard deviation of the baselines on 10 macrocyclic peptide datasets, respectively. Overall, image-based methods show great advantages as 4 (CGIP-Image, MaskMol, IEM-I, ResNet18-I-R) out of the top 5 baselines are vision-based methods, which shows the advantages of image modality on macrocyclic peptides. Additionally, we found the limitation of fingerprinting on macrocyclic peptides, with only 2 results out of 66 achieving top-5 performance. If we only observe the non-pretrained methods, we

Table S28: The average RMSE performance on MBANet$_{attr}$ with 10 seeds. The modality types from top to bottom are sequence, graph, geometry graph, image, geometry image, and video. -R means no pre-training, L means the number of layers, -I, -G, and -V mean the modalities are image, geometry image, and video, respectively. The green background represents top-5 performance.

| Models | MW | LogP | MR | BalabanJ | #HA | #HD | #VE | TPSA | Mean |
|---|---|---|---|---|---|---|---|---|---|
| BERT-6L | 39.046 | 1.087 | 9.068 | 0.572 | 1.261 | 0.997 | 15.733 | 23.823 | 11.448 |
| BERT-8L | 39.414 | 1.092 | 8.988 | 0.579 | 1.198 | 0.939 | 15.709 | 23.853 | 11.471 |
| RoBERTa-12L | 47.678 | 0.969 | 10.315 | 0.415 | 1.401 | 0.990 | 18.931 | 24.872 | 13.197 |
| molformer-R | 45.182 | 1.121 | 10.428 | 0.534 | 1.261 | 1.051 | 18.099 | 25.567 | 12.906 |
| Chem-BERT-6L | 38.875 | 1.030 | 8.749 | 0.541 | 1.240 | 0.968 | 15.677 | 23.714 | 11.349 |
| Chem-BERT-8L | 38.660 | 1.043 | 8.813 | 0.600 | 1.170 | 0.941 | 15.539 | 23.422 | 11.273 |
| CHEM-RoBERTa-12L | 47.740 | 0.962 | 10.214 | 0.539 | 1.395 | 0.954 | 18.540 | 24.814 | 13.145 |
| Molformer | 43.787 | 1.105 | 9.434 | 0.624 | 1.295 | 1.096 | 17.079 | 25.100 | 12.440 |
| GIN-R | 30.627 | 0.944 | 9.947 | 0.530 | 0.800 | 0.877 | 13.438 | 10.946 | 8.514 |
| EdgePred | 26.004 | 0.825 | 7.613 | 0.661 | 0.811 | 0.748 | 9.433 | 8.925 | 6.878 |
| ContextPred | 27.664 | 0.822 | 7.389 | 0.599 | 0.773 | 0.754 | 9.318 | 8.387 | 6.963 |
| infomax | 26.379 | 0.761 | 7.152 | 0.585 | 0.721 | 0.694 | 9.176 | 8.827 | 6.787 |
| masking | 25.193 | 0.713 | 6.516 | 0.502 | 0.694 | 0.846 | 9.081 | 8.740 | 6.536 |
| MolCLR | 28.482 | 0.803 | 9.801 | 0.603 | 0.675 | 0.685 | 11.910 | 10.982 | 7.993 |
| MoleBERT | 23.475 | 0.650 | 6.062 | 0.495 | 0.788 | 0.714 | 8.871 | 8.795 | 6.231 |
| CGIP-Graph | 16.019 | 0.674 | 4.833 | 0.384 | 0.614 | 0.447 | 6.075 | 4.991 | 4.255 |
| GraphMVP | 28.781 | 0.761 | 9.264 | 0.704 | 0.662 | 0.813 | 11.588 | 10.157 | 7.841 |
| Uni-Mol-R (1 conf) | 36.526 | 1.081 | 9.017 | 0.640 | 1.103 | 1.171 | 14.486 | 22.852 | 10.859 |
| Uni-Mol (1 conf) | 37.862 | 1.101 | 8.893 | 0.533 | 0.918 | 1.095 | 14.419 | 22.263 | 10.886 |
| TFN | 10.075 | 0.688 | 3.408 | 0.534 | 0.713 | 0.654 | 3.380 | 5.279 | 3.091 |
| SE3_Transformer | 33.948 | 0.888 | 11.641 | 0.568 | 1.126 | 0.920 | 19.913 | 24.265 | 11.659 |
| ResNet18-I-R | 23.070 | 0.921 | 6.251 | 0.601 | 0.935 | 0.746 | 8.721 | 14.364 | 6.951 |
| ImageMol | 21.691 | 0.908 | 5.686 | 0.629 | 0.807 | 0.811 | 7.971 | 13.566 | 6.509 |
| CGIP-Image | 23.269 | 0.955 | 6.382 | 0.494 | 0.851 | 0.801 | 8.844 | 14.549 | 7.018 |
| MaskMol | 40.568 | 1.073 | 9.277 | 0.553 | 1.149 | 0.993 | 16.432 | 24.063 | 11.764 |
| IEM-I | 22.673 | 0.865 | 5.906 | 0.544 | 0.871 | 0.784 | 8.365 | 14.144 | 6.769 |
| ResNet18-G-R | 15.357 | 0.813 | 4.709 | 0.574 | 0.794 | 0.655 | 6.930 | 8.951 | 4.848 |
| IEM-G (1 conf) | 15.694 | 0.745 | 4.834 | 0.468 | 0.713 | 0.717 | 6.960 | 9.193 | 4.916 |
| ResNet18-V-R | 8.121 | 0.750 | 2.661 | 0.582 | 0.711 | 0.656 | 3.870 | 3.929 | 2.660 |
| IEM-V | 6.542 | 0.703 | 2.280 | 0.417 | 0.726 | 0.707 | 2.699 | 3.961 | 2.254 |

find that the 3 vision-based modalities are the best because their performance is in the top 3. This suggests that we can make some further efforts in the future and propose some vision-based methods for the prediction of macrocyclic peptides.

## I.6 RETICULAR MOLECULES

Table S41 and Table S42 show the average RMSE performance and corresponding standard deviation of the baselines on 10 reticular datasets, respectively. We find that image-based methods without any pre-training achieve the best average performance on reticular molecules compared to many pre-trained methods, which suggests that the image modality is suitable for processing reticular molecules. If we only look at the non-pretrained methods, the three vision-based modalities achieve the best top 3 performance. In addition, we find that all vision-based pre-training methods suffer from negative transfer problems on reticular molecules, which deserves to be further studied and explored in the future.

## J COMPUTATIONAL EFFICIENCY

In virtual screening, computational efficiency of models is very important. Here, we analyze the number of parameters of different modal methods and their computational efficiency in training and inference. All evaluation are performed on 1 GeForce RTX 4090 GPU and with a batch size of 8. As shown in Table S43, we find that the video modality takes the most time. This is because a molecular video consists of 60 frames, which greatly increases the time cost. Secondly, we find that the image, SMILES and geometry graph modalities have larger parameter counts, such as MaskMol, CHEM-RoBERTa and Uni-Mol, which is due to the fact that they utilize the architecture of transformer and its variants.

Table S29: The standard deviation performance on MBANet$_{attr}$ with 10 seeds. The modality types from top to bottom are sequence, graph, geometry graph, image, geometry image, and video. -R means no pre-training, L means the number of layers, -I, -G, and -V mean the modalities are image, geometry image, and video, respectively.

| Models | MW | LogP | MR | BalabanJ | #HA | #HD | #VE | TPSA |
|---|---|---|---|---|---|---|---|---|
| BERT-6L | 1.476 | 0.253 | 0.866 | 0.209 | 0.434 | 0.176 | 0.769 | 1.356 |
| BERT-8L | 1.547 | 0.259 | 0.916 | 0.213 | 0.310 | 0.145 | 0.779 | 1.373 |
| RoBERTa-12L | 2.329 | 0.077 | 0.910 | 0.170 | 0.313 | 0.140 | 1.335 | 2.093 |
| molformer-R | 1.529 | 0.213 | 1.332 | 0.247 | 0.252 | 0.277 | 1.360 | 2.695 |
| Chem-BERT-6L | 1.452 | 0.158 | 1.199 | 0.277 | 0.299 | 0.182 | 0.810 | 1.246 |
| Chem-BERT-8L | 1.734 | 0.159 | 0.989 | 0.294 | 0.274 | 0.167 | 0.938 | 1.322 |
| CHEM-RoBERTa-12L | 2.195 | 0.058 | 0.938 | 0.262 | 0.306 | 0.117 | 1.239 | 2.150 |
| Molformer | 1.678 | 0.210 | 1.102 | 0.285 | 0.294 | 0.329 | 1.072 | 2.522 |
| GIN-R | 2.209 | 0.290 | 1.356 | 0.178 | 0.192 | 0.269 | 0.939 | 2.655 |
| EdgePred | 2.390 | 0.213 | 1.193 | 0.231 | 0.161 | 0.110 | 1.481 | 1.197 |
| ContextPred | 2.546 | 0.180 | 1.127 | 0.169 | 0.178 | 0.084 | 0.985 | 1.142 |
| infomax | 3.192 | 0.137 | 0.920 | 0.186 | 0.116 | 0.089 | 0.765 | 1.403 |
| masking | 3.185 | 0.148 | 0.774 | 0.116 | 0.070 | 0.195 | 0.873 | 1.323 |
| MolCLR | 3.530 | 0.254 | 1.126 | 0.220 | 0.100 | 0.104 | 1.155 | 2.636 |
| MoleBERT | 2.107 | 0.045 | 0.824 | 0.163 | 0.138 | 0.080 | 1.129 | 0.654 |
| CGIP-Graph | 1.406 | 0.188 | 0.698 | 0.171 | 0.210 | 0.108 | 0.719 | 0.667 |
| GraphMVP | 3.009 | 0.179 | 1.105 | 0.196 | 0.095 | 0.105 | 1.259 | 1.657 |
| Uni-Mol-R (1 conf) | 4.428 | 0.136 | 1.514 | 0.305 | 0.183 | 0.208 | 2.452 | 2.007 |
| Uni-Mol (1 conf) | 3.155 | 0.155 | 1.411 | 0.244 | 0.098 | 0.297 | 1.957 | 1.797 |
| TFN | 1.837 | 0.122 | 0.679 | 0.244 | 0.092 | 0.065 | 0.506 | 1.458 |
| SE3_Transformer | **0.083** | **0.003** | **0.039** | **0.025** | **0.019** | **0.009** | **0.026** | **0.017** |
| ResNet18-I-R | 0.718 | 0.073 | 0.692 | 0.223 | 0.185 | 0.090 | 0.922 | 1.735 |
| ImageMol | 1.206 | 0.105 | 0.342 | 0.319 | 0.089 | 0.134 | 0.871 | 0.527 |
| CGIP-Image | 0.931 | 0.176 | 0.308 | 0.155 | 0.133 | 0.114 | 0.781 | 1.046 |
| MaskMol | 1.477 | 0.247 | 0.773 | 0.244 | 0.246 | 0.127 | 1.449 | 1.322 |
| IEM-I | 1.057 | 0.065 | 0.380 | 0.238 | 0.106 | 0.078 | 0.974 | 0.717 |
| ResNet18-G-R | 1.190 | 0.156 | 0.484 | 0.162 | 0.132 | 0.085 | 0.616 | 0.499 |
| IEM-G (1 conf) | 1.812 | 0.043 | 0.898 | 0.151 | 0.063 | 0.129 | 1.103 | 0.664 |
| ResNet18-V-R | 1.083 | 0.102 | 0.363 | 0.274 | 0.224 | 0.126 | 0.704 | 0.408 |
| IEM-V | 0.845 | 0.085 | 0.107 | 0.136 | 0.205 | 0.185 | 0.496 | 0.539 |

Given the high temporal cost of the video modality, we further discuss the impact of different numbers of frames on the computational cost. As shown in Table S44, We find that adjusting the number of frames can effectively improve computational efficiency. Therefore, we can try to reduce the number of frames when computing resources are limited.

Next, we analyze how long it takes the model to perform virtual screening on 10,000 molecules. As shown in Table S45, we find that the video modality required more time for virtual screening, while the other modalities took comparable time.

# K  IN-DEPTH ANALYSIS OF INSIGHTS

## K.1  PREDICTION DIVERSITY BETWEEN DIFFERENT MODALITIES ON HIV DATASET

Here, we study the pairwise diversity of different modalities, including feature diversity and prediction diversity, on a single dataset. Especially, we select the dataset with the largest number of samples from MoleculeNet based on single-task classification as an example, namely HIV, because a larger number of samples will provide a more stable conclusion. We use mcfp_2048, Chem-BERT-8L, MoleBERT, Uni-Mol (10 conf), MaskMol, IEM-G, and VideoMol as representatives of each modality (fingerprint, sequence, geometry graph, image, geometry image, video) because they achieve excellent performance on 8 molecular property prediction tasks based on classification tasks. The ROC-AUC performances of mcfp_2048, Chem-BERT-8L, MoleBERT, Uni-Mol (10 conf), MaskMol, IEM-G (10 conf), and VideoMol on the HIV test set are 74.1%, 78.8%, 76.7%, 77.8%, 75.1%, 80.5% and 78.3% respectively.

As shown in Table S47 and Table S48, we find that different modalities have different degrees of differences in the logits of predicting HIV, including RMSE differences and Pearson differences, which provides evidence that fusing different molecular modalities can increase the diversity of

Table S30: The RMSE performance on MBANet$_{atom}^{30K}$, MBANet$_{bond}^{30K}$, MBANet$_{attr}^{30K}$ with 3 seeds. *-R, TFN and SE3_Transformer represent non-pre-trained models, and the others are pre-trained models.

| | Models | MBANet$_{atom}^{30K}$ | MBANet$_{bond}^{30K}$ | MBANet$_{attr}^{30K}$ |
|---|---|---|---|---|
| SMILES-based | BERT-6L | 0.647 | 2.020 | 10.961 |
| | BERT-8L | 0.645 | 2.014 | 10.936 |
| | RoBERTa-12L | 0.644 | 2.030 | 11.223 |
| | molformer-R | 0.646 | 2.042 | 11.159 |
| | Chem-BERT-6L | 0.647 | 2.019 | 10.960 |
| | Chem-BERT-8L | 0.647 | 2.019 | 10.960 |
| | CHEM-RoBERTa-12L | 0.644 | 2.042 | 11.249 |
| | Molformer | 0.646 | 2.046 | 11.031 |
| Graph-based | GIN-R | 0.339 | 0.627 | 6.711 |
| | EdgePred | 0.306 | 0.613 | 6.480 |
| | ContextPred | 0.312 | 0.605 | 6.508 |
| | infomax | 0.325 | 0.612 | 6.313 |
| | masking | 0.315 | 0.606 | 6.323 |
| | MolCLR | 0.329 | 0.627 | 6.873 |
| | MoleBERT | 0.309 | 0.607 | 6.259 |
| | CGIP-Graph | 0.243 | 0.574 | 5.710 |
| | GraphMVP | 0.317 | 0.621 | 6.868 |
| Geometry-based | Uni-Mol-R (1 conf) | 0.504 | 1.977 | 10.299 |
| | Uni-Mol (1 conf) | 0.510 | 1.979 | 10.247 |
| | TFN | 0.187 | **0.239** | 2.913 |
| | SE3_Transformer | 0.642 | 2.070 | 11.087 |
| Image-based | ResNet18-I-R | 0.359 | 1.176 | 6.263 |
| | ImageMol | 0.355 | 1.156 | 6.224 |
| | CGIP-Image | 0.358 | 1.163 | 6.400 |
| | MaskMol | 0.641 | 1.984 | 10.748 |
| | IEM-I | 0.358 | 1.198 | 6.433 |
| Geometry-based | ResNet18-G-R | 0.242 | 1.139 | 5.164 |
| | IEM-G (1 conf) | 0.245 | 1.136 | 5.315 |
| Video-based | ResNet18-V-R | 0.145 | 0.779 | 2.504 |
| | IEM-V | **0.140** | 0.751 | **2.216** |

Table S31: The average RMSE performance on 10 acyclic chain (AC) datasets from Struct-Net. AC#1, AC#2, AC#3, AC#4, AC#5, AC#6, AC#7, AC#8, AC#9 and AC#10 represent CHEMBL1614458_Potency, CHEMBL4513082_Inhibition, CHEMBL4495582_Inhibition, CHEMBL4296187_Inhibition, CHEMBL4296188_Inhibition, CHEMBL1614361_Potency, CHEMBL4303805_Inhibition, CHEMBL4649955_Potency, CHEMBL4649949_Potency and CHEMBL4649948_Potency, respectively. The green background represents top-5 performance.

| Models | AC#1 | AC#2 | AC#3 | AC#4 | AC#5 | AC#6 | AC#7 | AC#8 | AC#9 | AC#10 | Mean |
|---|---|---|---|---|---|---|---|---|---|---|---|
| mcfp4_2048 | 0.932 | 0.143 | 17.976 | 19.208 | 7.337 | 1.226 | 29.222 | 14.469 | 7.497 | 9.194 | 10.720 |
| ecfp4_2048 | 0.954 | 0.147 | 18.022 | 19.186 | 7.399 | 1.290 | 29.415 | 14.468 | 7.482 | 9.194 | 10.756 |
| maccs | 1.044 | 0.151 | 17.191 | 19.148 | 7.091 | 1.159 | 29.277 | 14.473 | 7.468 | 9.190 | 10.619 |
| physchem | 0.795 | 0.155 | 18.122 | 19.271 | 7.318 | 1.159 | 29.234 | 14.483 | 7.517 | 9.202 | 10.726 |
| atompair_2048 | 1.098 | 0.152 | 17.962 | 18.759 | 7.241 | 1.420 | 29.572 | 14.474 | 7.301 | 9.192 | 10.717 |
| rdkDes | 1.049 | 0.133 | 17.167 | 19.349 | 6.984 | 1.317 | 29.046 | 14.470 | 7.497 | 9.186 | 10.620 |
| BERT-6L | 0.757 | 0.400 | 15.976 | 19.173 | 6.818 | 1.175 | 28.848 | 14.499 | 7.594 | 9.379 | 10.462 |
| BERT-8L | 0.757 | 0.392 | 15.977 | 19.179 | 6.822 | 1.167 | 28.892 | 14.502 | 7.585 | 9.350 | 10.462 |
| RoBERTa-12L | 0.758 | 0.370 | 15.959 | 19.219 | 6.817 | 1.177 | 28.730 | 14.500 | 7.603 | 9.367 | 10.450 |
| molformer-R | 0.758 | 0.389 | 15.952 | 19.168 | 6.822 | 1.176 | 28.682 | 14.496 | 7.595 | 9.353 | 10.439 |
| Chem-BERT-6L | 0.757 | 0.442 | 15.980 | 19.164 | 6.821 | 1.171 | 28.918 | 14.502 | 7.597 | 9.348 | 10.470 |
| Chem-BERT-8L | 0.757 | 0.426 | 15.994 | 19.169 | 6.822 | 1.177 | 28.997 | 14.499 | 7.597 | 9.365 | 10.480 |
| CHEM-RoBERTa-12L | 0.760 | 0.416 | 15.984 | 19.196 | 6.818 | 1.174 | 28.965 | 14.502 | 7.601 | 9.359 | 10.478 |
| Molformer | 0.758 | 0.414 | 15.962 | 19.132 | 6.820 | 1.171 | 28.681 | 14.498 | 7.590 | 9.342 | 10.437 |
| GIN-R | 0.774 | 0.261 | 14.889 | 19.617 | 7.059 | 1.187 | 28.948 | 14.524 | 7.767 | 9.564 | 10.459 |
| EdgePred | 0.968 | 0.152 | 16.519 | 19.331 | 6.899 | 1.175 | 29.197 | 14.382 | 7.451 | 9.184 | 10.526 |
| ContextPred | 0.780 | 0.183 | 15.162 | 19.280 | 7.100 | 1.240 | 29.121 | 14.463 | 7.364 | 9.086 | 10.378 |
| infomax | 0.795 | 0.238 | 14.463 | 19.223 | 6.971 | 1.301 | 29.171 | 14.412 | 7.676 | 9.202 | 10.345 |
| masking | 0.757 | 0.185 | 14.693 | 19.193 | 7.122 | 1.208 | 29.123 | 14.463 | 7.699 | 9.224 | 10.367 |
| MolCLR | 0.773 | 0.200 | 14.899 | 19.312 | 6.817 | 1.195 | 28.959 | 14.308 | 7.681 | 9.445 | 10.359 |
| MoleBERT | 0.823 | 0.172 | 14.211 | 19.087 | 6.713 | 1.309 | 29.213 | 14.481 | 7.462 | 9.178 | 10.265 |
| CGIP-Graph | 0.806 | 0.160 | 14.426 | 19.914 | 6.859 | 1.264 | 29.180 | 14.442 | 7.876 | 9.290 | 10.422 |
| GraphMVP | 0.820 | 0.231 | 13.950 | 19.430 | 7.132 | 1.244 | 29.649 | 14.449 | 7.860 | 9.191 | 10.396 |
| SchNet | 0.828 | 0.324 | 13.176 | 19.230 | 7.347 | 1.360 | 29.318 | 14.485 | 7.513 | 9.262 | 10.284 |
| EGNN | 0.759 | 0.217 | 15.898 | 19.134 | 6.818 | 1.212 | 29.331 | 14.446 | 7.534 | 9.318 | 10.467 |
| TFN | 0.883 | 0.469 | 15.922 | 18.951 | 7.101 | 1.448 | 28.928 | 14.382 | 7.608 | 9.275 | 10.497 |
| SE3_Transformer | 0.868 | 0.334 | 15.426 | 18.934 | 6.967 | 1.435 | 29.001 | 14.621 | 7.886 | 9.537 | 10.501 |
| PaiNN | 2.365 | 0.708 | 15.725 | 18.935 | 7.885 | 1.604 | 30.369 | 14.485 | 7.993 | 9.645 | 10.971 |
| Uni-Mol-R (1 conf) | 0.758 | 0.411 | 15.935 | 19.185 | 6.819 | 1.175 | 28.797 | 14.499 | 7.606 | 9.370 | 10.455 |
| Uni-Mol (1 conf) | 0.758 | 0.418 | 15.864 | 19.196 | 6.817 | 1.179 | 28.814 | 14.483 | 7.568 | 9.353 | 10.445 |
| ResNet18-I-R | 0.759 | 0.328 | 16.341 | 19.140 | 6.888 | 1.195 | 29.084 | 14.455 | 7.908 | 9.404 | 10.550 |
| ImageMol | 0.781 | 0.355 | 16.109 | 19.259 | 6.970 | 1.252 | 29.039 | 14.368 | 7.948 | 9.470 | 10.555 |
| CGIP-Image | 0.768 | 0.247 | 16.281 | 19.147 | 6.897 | 1.196 | 28.980 | 14.525 | 8.170 | 9.625 | 10.584 |
| MaskMol | 0.759 | 0.404 | 15.962 | 19.161 | 6.821 | 1.172 | 28.944 | 14.491 | 7.576 | 9.319 | 10.461 |
| IEM-I | 0.762 | 0.199 | 16.288 | 19.166 | 6.819 | 1.229 | 29.279 | 14.421 | 7.879 | 9.498 | 10.554 |
| ResNet18-G-R | 0.775 | 0.334 | 16.538 | 18.802 | 6.737 | 1.139 | 29.115 | 14.524 | 7.560 | 9.290 | 10.481 |
| IEM-G (1 conf) | 0.771 | 0.218 | 15.876 | 19.294 | 6.996 | 1.155 | 28.986 | 14.733 | 7.664 | 9.242 | 10.494 |
| ViT-G-R | 0.758 | 0.359 | 15.962 | 19.165 | 6.827 | 1.176 | 28.662 | 14.462 | 7.582 | 9.344 | 10.430 |
| VideoMol-G | 0.759 | 0.401 | 15.970 | 19.157 | 6.822 | 1.172 | 28.680 | 14.476 | 7.580 | 9.350 | 10.437 |
| ResNet18-V-R | 0.769 | 0.359 | 16.196 | 18.761 | 6.821 | 1.156 | 29.119 | 14.435 | 7.475 | 9.323 | 10.441 |
| IEM-V | 0.771 | 0.210 | 15.798 | 18.949 | 6.813 | 1.175 | 29.280 | 14.432 | 7.460 | 9.311 | 10.420 |

Table S32: The standard deviation on 10 acyclic chain (AC) datasets from StructNet. AC#1, AC#2, AC#3, AC#4, AC#5, AC#6, AC#7, AC#8, AC#9 and AC#10 represent CHEMBL1614458_Potency, CHEMBL4513082_Inhibition, CHEMBL4495582_Inhibition, CHEMBL4296187_Inhibition, CHEMBL4296188_Inhibition, CHEMBL1614361_Potency, CHEMBL4303805_Inhibition, CHEMBL4649955_Potency, CHEMBL4649949_Potency and CHEMBL4649948_Potency, respectively.

| Models | AC#1 | AC#2 | AC#3 | AC#4 | AC#5 | AC#6 | AC#7 | AC#8 | AC#9 | AC#10 |
|---|---|---|---|---|---|---|---|---|---|---|
| mcfp4_2048 | 0.008 | 0.003 | 0.056 | 0.008 | 0.009 | 0.004 | 0.013 | 0.005 | 0.003 | 0.002 |
| ecfp4_2048 | 0.008 | 0.004 | 0.053 | 0.008 | 0.008 | 0.004 | 0.021 | 0.005 | 0.004 | 0.002 |
| maccs | 0.011 | 0.006 | 0.045 | 0.006 | 0.009 | 0.004 | 0.025 | 0.006 | 0.009 | 0.006 |
| physchem | 0.013 | 0.028 | 0.050 | 0.012 | 0.026 | 0.008 | 0.020 | 0.012 | 0.006 | 0.013 |
| atompair_2048 | 0.008 | 0.003 | 0.056 | 0.011 | 0.006 | 0.004 | 0.028 | 0.007 | 0.005 | 0.003 |
| rdkDes | 0.013 | 0.003 | 0.085 | 0.009 | 0.008 | 0.007 | 0.015 | 0.006 | 0.007 | 0.008 |
| BERT-6L | 0.000 | 0.062 | 0.060 | 0.050 | 0.005 | 0.013 | 0.253 | 0.007 | 0.019 | 0.034 |
| BERT-8L | 0.000 | 0.058 | 0.069 | 0.066 | 0.007 | 0.013 | 0.269 | 0.003 | 0.018 | 0.053 |
| RoBERTa-12L | 0.002 | 0.082 | 0.022 | 0.061 | 0.003 | 0.009 | 0.406 | 0.006 | 0.017 | 0.029 |
| molformer-R | 0.002 | 0.082 | 0.005 | 0.034 | 0.005 | 0.004 | 0.216 | 0.008 | 0.016 | 0.042 |
| Chem-BERT-6L | 0.001 | 0.077 | 0.044 | 0.041 | 0.015 | 0.009 | 0.310 | 0.005 | 0.027 | 0.038 |
| Chem-BERT-8L | 0.000 | 0.094 | 0.059 | 0.051 | 0.012 | 0.005 | 0.241 | 0.006 | 0.019 | 0.041 |
| CHEM-RoBERTa-12L | 0.003 | 0.081 | 0.038 | 0.049 | 0.004 | 0.008 | 0.315 | 0.009 | 0.022 | 0.030 |
| Molformer | 0.001 | 0.097 | 0.031 | 0.042 | 0.006 | 0.010 | 0.249 | 0.011 | 0.007 | 0.073 |
| GIN-R | 0.017 | 0.101 | 1.137 | 0.412 | 0.265 | 0.047 | 0.305 | 0.194 | 0.279 | 0.341 |
| EdgePred | 0.558 | 0.029 | 1.804 | 0.107 | 0.109 | 0.033 | 0.073 | 0.257 | 0.199 | 0.156 |
| ContextPred | 0.023 | 0.028 | 0.473 | 0.117 | 0.145 | 0.022 | 0.142 | 0.045 | 0.043 | 0.162 |
| infomax | 0.026 | 0.047 | 0.333 | 0.168 | 0.166 | 0.034 | 0.093 | 0.092 | 0.303 | 0.130 |
| masking | 0.014 | 0.020 | 1.020 | 0.080 | 0.328 | 0.016 | 0.237 | 0.030 | 0.177 | 0.102 |
| MolCLR | 0.028 | 0.049 | 0.570 | 0.215 | 0.269 | 0.024 | 0.430 | 0.176 | 0.145 | 0.271 |
| MoleBERT | 0.045 | 0.020 | 0.555 | 0.172 | 0.073 | 0.031 | 0.128 | 0.026 | 0.069 | 0.019 |
| CGIP-Graph | 0.018 | 0.032 | 1.051 | 0.721 | 0.208 | 0.055 | 0.374 | 0.042 | 0.131 | 0.142 |
| GraphMVP | 0.076 | 0.047 | 1.503 | 0.528 | 0.234 | 0.068 | 0.498 | 0.064 | 0.394 | 0.076 |
| SchNet | 0.030 | 0.124 | 1.182 | 0.634 | 0.283 | 0.076 | 0.395 | 0.080 | 0.096 | 0.172 |
| EGNN | 0.001 | 0.148 | 0.089 | 0.065 | 0.011 | 0.025 | 0.195 | 0.143 | 0.024 | 0.359 |
| TFN | 0.093 | 0.157 | 0.598 | 0.110 | 0.149 | 0.052 | 0.266 | 0.231 | 0.129 | 0.143 |
| SE3_Transformer | 0.045 | 0.098 | 0.721 | 0.182 | 0.182 | 0.058 | 0.472 | 0.267 | 0.303 | 0.231 |
| PaiNN | 0.305 | 0.240 | 0.915 | 0.366 | 0.443 | 0.138 | 0.562 | 0.244 | 0.280 | 0.365 |
| Uni-Mol-R (1 conf) | 0.001 | 0.068 | 0.064 | 0.091 | 0.005 | 0.006 | 0.154 | 0.008 | 0.022 | 0.034 |
| Uni-Mol (1 conf) | 0.001 | 0.089 | 0.161 | 0.067 | 0.026 | 0.015 | 0.178 | 0.071 | 0.050 | 0.049 |
| ResNet18-I-R | 0.010 | 0.113 | 0.217 | 0.174 | 0.095 | 0.038 | 0.399 | 0.119 | 0.326 | 0.181 |
| ImageMol | 0.023 | 0.184 | 0.421 | 0.398 | 0.267 | 0.057 | 0.694 | 0.161 | 0.531 | 0.539 |
| CGIP-Image | 0.013 | 0.099 | 0.663 | 0.602 | 0.248 | 0.036 | 0.337 | 0.111 | 0.419 | 0.650 |
| MaskMol | 0.003 | 0.074 | 0.069 | 0.053 | 0.005 | 0.015 | 0.262 | 0.020 | 0.061 | 0.039 |
| IEM-I | 0.009 | 0.038 | 0.772 | 0.178 | 0.317 | 0.059 | 0.464 | 0.183 | 0.417 | 0.571 |
| ResNet18-G-R | 0.010 | 0.076 | 0.368 | 0.290 | 0.159 | 0.020 | 0.354 | 0.175 | 0.205 | 0.097 |
| IEM-G (1 conf) | 0.026 | 0.110 | 0.268 | 0.607 | 0.231 | 0.021 | 0.381 | 0.634 | 0.189 | 0.172 |
| ViT-G-R | 0.001 | 0.078 | 0.141 | 0.080 | 0.014 | 0.023 | 0.165 | 0.120 | 0.026 | 0.046 |
| VideoMol-G | 0.002 | 0.120 | 0.225 | 0.056 | 0.007 | 0.023 | 0.070 | 0.030 | 0.031 | 0.043 |
| ResNet18-V-R | 0.009 | 0.081 | 0.264 | 0.118 | 0.030 | 0.018 | 0.141 | 0.063 | 0.102 | 0.088 |
| IEM-V | 0.022 | 0.088 | 0.284 | 0.224 | 0.191 | 0.028 | 0.278 | 0.147 | 0.181 | 0.207 |

Table S33: The average RMSE performance on 10 acyclic (A) datasets from StructNet. A#1, A#2, A#3, A#4, A#5, A#6, A#7, A#8, A#9, A#10 represent CHEMBL4513082_Inhibition, CHEMBL4495582_Inhibition, CHEMBL1614458_Potency, CHEMBL4303805_Inhibition, CHEMBL4808149_Inhibition, CHEMBL4296187_Inhibition, CHEMBL4808150_Inhibition, CHEMBL4296188_Inhibition, CHEMBL4649955_Potency, CHEMBL4649949_Potency, respectively. The green background represents top-5 performance.

| Models | A#1 | A#2 | A#3 | A#4 | A#5 | A#6 | A#7 | A#8 | A#9 | A#10 | Mean |
|---|---|---|---|---|---|---|---|---|---|---|---|
| mcfp4_2048 | 0.134 | 17.750 | 0.666 | 15.248 | 26.727 | 17.738 | 14.628 | 6.490 | 13.084 | 13.108 | 12.557 |
| ecfp4_2048 | 0.130 | 17.746 | 0.626 | 15.214 | 26.740 | 17.713 | 14.622 | 6.544 | 13.029 | 13.147 | 12.551 |
| maccs | 0.140 | 17.387 | 0.670 | **15.129** | 26.760 | 17.704 | 14.634 | 6.537 | 13.085 | 13.109 | 12.516 |
| physchem | 0.154 | 17.765 | **0.531** | 15.208 | 26.739 | 17.823 | 14.634 | 6.430 | 13.073 | 13.107 | 12.546 |
| atompair_2048 | 0.135 | 17.996 | 0.782 | 15.394 | 26.721 | 17.548 | 14.626 | 6.431 | 13.104 | 12.989 | 12.573 |
| rdkDes | 0.139 | 17.284 | 0.697 | 15.379 | 26.736 | 17.843 | 14.629 | 6.369 | **12.982** | 13.019 | 12.508 |
| BERT-6L | 0.131 | 17.418 | 0.566 | 15.311 | 27.162 | 17.480 | 14.636 | 6.437 | 13.004 | 13.079 | 12.522 |
| BERT-8L | 0.130 | 17.410 | 0.566 | 15.306 | 27.158 | 17.480 | 14.637 | 6.425 | 13.006 | 13.081 | 12.520 |
| RoBERTa-12L | 0.133 | 17.415 | 0.567 | 15.370 | 27.225 | 17.477 | 14.636 | 6.401 | 13.001 | 13.052 | 12.528 |
| molformer-R | 0.130 | 17.423 | 0.566 | 15.369 | 27.206 | 17.479 | 14.650 | 6.386 | 13.005 | 13.100 | 12.531 |
| Chem-BERT-6L | 0.131 | 17.410 | 0.566 | 15.406 | 27.194 | 17.479 | 14.638 | 6.425 | 13.004 | 13.070 | 12.532 |
| Chem-BERT-8L | 0.132 | 17.417 | 0.567 | 15.302 | 27.178 | 17.479 | 14.639 | 6.425 | 13.001 | 13.066 | 12.521 |
| CHEM-RoBERTa-12L | 0.132 | 17.420 | 0.566 | 15.397 | 27.186 | 17.409 | 14.638 | 6.413 | 13.002 | 13.031 | 12.519 |
| Molformer | 0.131 | 17.421 | 0.566 | 15.277 | 27.167 | 17.476 | 14.637 | 6.380 | 13.003 | 13.045 | 12.510 |
| GIN-R | 0.191 | 15.578 | 0.590 | 15.626 | 27.423 | 18.694 | 14.554 | 6.319 | 13.169 | 13.217 | 12.536 |
| EdgePred | 0.147 | 49.805 | 0.578 | 15.623 | 26.868 | 17.992 | 13.821 | 6.215 | 13.253 | 13.072 | 15.737 |
| ContextPred | 0.185 | 12.660 | 0.585 | 15.560 | 26.788 | 17.876 | 13.928 | 5.881 | 13.180 | 12.875 | 11.952 |
| infomax | 0.258 | 14.447 | 0.545 | 15.551 | 26.784 | 17.795 | 14.156 | 6.176 | 13.288 | **12.475** | 12.147 |
| masking | 0.156 | 13.128 | 0.558 | 15.562 | 26.814 | 17.769 | 14.119 | 6.133 | 13.330 | 13.102 | 12.067 |
| MolCLR | 0.137 | 15.522 | 0.575 | 15.826 | 27.120 | 17.939 | 14.544 | 6.352 | 13.077 | 13.121 | 12.421 |
| MoleBERT | 0.216 | 12.496 | 0.552 | 15.343 | 26.824 | 17.704 | **13.605** | **5.633** | 13.124 | 12.936 | **11.843** |
| CGIP-Graph | 0.198 | 12.395 | 0.603 | 16.262 | **26.659** | 18.565 | 14.677 | 5.770 | 13.118 | 12.598 | 12.085 |
| GraphMVP | 0.142 | **11.831** | 0.599 | 16.323 | 26.701 | 19.174 | 14.528 | 6.195 | 13.233 | 12.997 | 12.172 |
| SchNet | 0.222 | 13.636 | 0.610 | 16.680 | 27.419 | 17.531 | 15.177 | 6.390 | 13.279 | 13.253 | 12.420 |
| EGNN | 0.134 | 16.746 | 0.576 | 16.158 | 27.181 | 17.527 | 14.781 | 6.403 | 13.072 | 13.110 | 12.569 |
| TFN | 0.177 | 14.336 | 0.641 | 15.314 | 27.021 | 17.453 | 14.530 | 6.396 | 13.071 | 12.981 | 12.192 |
| SE3_Transformer | 0.238 | 15.176 | 0.614 | 15.491 | 27.222 | 17.475 | 14.308 | 6.304 | 13.118 | 13.172 | 12.312 |
| PaiNN | 0.696 | 13.416 | 2.167 | 18.349 | 27.323 | **17.384** | 16.141 | 6.584 | 13.122 | 13.161 | 12.834 |
| Uni-Mol-R (1 conf) | 0.132 | 17.340 | 0.566 | 15.348 | 27.166 | 17.481 | 14.639 | 6.433 | 13.004 | 13.070 | 12.518 |
| Uni-Mol (1 conf) | 0.128 | 17.376 | 0.566 | 15.315 | 27.197 | 17.476 | 14.639 | 6.445 | 13.002 | 13.097 | 12.524 |
| ResNet18-I-R | 0.138 | 16.694 | 0.563 | 15.507 | 27.018 | 17.474 | 14.811 | 6.434 | 13.090 | 12.996 | 12.473 |
| ImageMol | 0.151 | 16.214 | 0.579 | 16.242 | 27.165 | 17.776 | 14.850 | 6.419 | 13.295 | 13.152 | 12.584 |
| CGIP-Image | 0.150 | 16.349 | 0.571 | 16.651 | 26.953 | 17.626 | 14.790 | 6.434 | 13.458 | 12.982 | 12.596 |
| MaskMol | 0.131 | 17.421 | 0.566 | 15.463 | 27.129 | 17.481 | 14.651 | 6.419 | 13.002 | 13.091 | 12.535 |
| IEM-I | 0.149 | 15.975 | 0.576 | 16.437 | 27.383 | 17.741 | 15.097 | 6.432 | 13.101 | 13.196 | 12.609 |
| ResNet18-G-R | 0.138 | 16.497 | 0.554 | 15.922 | 27.006 | 17.590 | 14.619 | 6.388 | 13.092 | 13.010 | 12.482 |
| IEM-G (1 conf) | 0.157 | 15.366 | 0.575 | 16.170 | 26.945 | 17.570 | 14.585 | 6.411 | 13.040 | 13.208 | 12.403 |
| ViT-G-R | 0.130 | 17.219 | 0.565 | 15.261 | 27.183 | 17.479 | 14.632 | 6.381 | 13.006 | 13.048 | 12.490 |
| VideoMol-G | **0.127** | 17.197 | 0.562 | 15.282 | 27.145 | 17.457 | 14.637 | 6.422 | 13.008 | 13.042 | 12.488 |
| ResNet18-V-R | 0.137 | 16.421 | 0.551 | 15.785 | 27.007 | 17.519 | 14.554 | 6.388 | 13.019 | 13.063 | 12.444 |
| IEM-V | 0.135 | 13.842 | 0.556 | 16.536 | 26.973 | 17.539 | 14.681 | 6.415 | 13.053 | 13.075 | 12.281 |

Table S34: The standard deviation on 10 acyclic (A) datasets from StructNet. A#1, A#2, A#3, A#4, A#5, A#6, A#7, A#8, A#9, A#10 represent CHEMBL4513082_Inhibition, CHEMBL4495582_Inhibition, CHEMBL1614458_Potency, CHEMBL4303805_Inhibition, CHEMBL4808149_Inhibition, CHEMBL4296187_Inhibition, CHEMBL4808150_Inhibition, CHEMBL4296188_Inhibition, CHEMBL4649955_Potency, CHEMBL4649949_Potency, respectively.

| Models | A#1 | A#2 | A#3 | A#4 | A#5 | A#6 | A#7 | A#8 | A#9 | A#10 |
|---|---|---|---|---|---|---|---|---|---|---|
| mcfp4_2048 | 0.003 | 0.014 | 0.002 | 0.007 | 0.002 | 0.008 | 0.009 | 0.006 | 0.004 | 0.011 |
| ecfp4_2048 | 0.003 | 0.012 | 0.002 | 0.008 | 0.002 | 0.006 | 0.020 | 0.007 | 0.011 | 0.027 |
| maccs | 0.008 | 0.021 | 0.006 | 0.006 | 0.008 | 0.005 | 0.006 | 0.017 | 0.015 | 0.016 |
| physchem | 0.017 | 0.017 | 0.002 | 0.008 | 0.011 | 0.014 | 0.003 | 0.017 | 0.015 | 0.005 |
| atompair_2048 | 0.003 | 0.010 | 0.004 | 0.006 | 0.004 | 0.009 | 0.004 | 0.010 | 0.004 | 0.022 |
| rdkDes | 0.013 | 0.009 | 0.005 | 0.005 | 0.007 | 0.012 | 0.005 | 0.014 | 0.013 | 0.067 |
| BERT-6L | 0.005 | 0.018 | 0.001 | 0.097 | 0.111 | 0.002 | 0.011 | 0.034 | 0.007 | 0.037 |
| BERT-8L | 0.003 | 0.017 | 0.001 | 0.096 | 0.108 | 0.004 | 0.011 | 0.022 | 0.009 | 0.031 |
| RoBERTa-12L | 0.012 | 0.012 | 0.004 | 0.109 | 0.138 | 0.004 | 0.011 | 0.058 | 0.005 | 0.059 |
| molformer-R | 0.003 | 0.009 | 0.001 | 0.140 | 0.144 | 0.004 | 0.018 | 0.047 | 0.010 | 0.076 |
| Chem-BERT-6L | 0.004 | 0.011 | 0.000 | 0.246 | 0.161 | 0.001 | 0.013 | 0.023 | 0.008 | 0.035 |
| Chem-BERT-8L | 0.005 | 0.012 | 0.003 | 0.132 | 0.161 | 0.003 | 0.014 | 0.041 | 0.008 | 0.059 |
| CHEM-RoBERTa-12L | 0.005 | 0.007 | 0.002 | 0.096 | 0.068 | 0.218 | 0.015 | 0.071 | 0.005 | 0.067 |
| Molformer | 0.005 | 0.006 | 0.001 | 0.027 | 0.069 | 0.009 | 0.015 | 0.059 | 0.008 | 0.077 |
| GIN-R | 0.048 | 2.463 | 0.019 | 0.493 | 0.559 | 0.813 | 0.334 | 0.077 | 0.271 | 0.284 |
| EdgePred | 0.014 | 35.577 | 0.020 | 0.302 | 0.155 | 0.082 | 0.460 | 0.215 | 0.110 | 0.123 |
| ContextPred | 0.038 | 2.445 | 0.026 | 0.203 | 0.081 | 0.124 | 0.604 | 0.497 | 0.181 | 0.153 |
| infomax | 0.053 | 3.404 | 0.024 | 0.176 | 0.045 | 0.105 | 0.474 | 0.396 | 0.209 | 0.561 |
| masking | 0.016 | 2.483 | 0.015 | 0.234 | 0.062 | 0.147 | 0.430 | 0.472 | 0.251 | 0.024 |
| MolCLR | 0.014 | 2.662 | 0.018 | 0.549 | 0.317 | 0.412 | 0.265 | 0.183 | 0.204 | 0.300 |
| MoleBERT | 0.082 | 1.515 | 0.022 | 0.114 | 0.094 | 0.074 | 0.270 | 0.310 | 0.064 | 0.187 |
| CGIP-Graph | 0.049 | 1.801 | 0.041 | 0.548 | 0.037 | 0.552 | 0.314 | 0.199 | 0.164 | 0.210 |
| GraphMVP | 0.015 | 2.054 | 0.036 | 0.947 | 0.029 | 0.803 | 0.501 | 0.295 | 0.148 | 0.317 |
| SchNet | 0.053 | 2.075 | 0.037 | 1.866 | 0.644 | 0.347 | 0.735 | 0.144 | 0.337 | 0.604 |
| EGNN | 0.016 | 1.135 | 0.020 | 0.926 | 0.256 | 0.143 | 0.178 | 0.036 | 0.021 | 0.085 |
| TFN | 0.036 | 3.056 | 0.047 | 0.158 | 0.570 | 0.187 | 0.492 | 0.103 | 0.138 | 0.177 |
| SE3_Transformer | 0.035 | 1.728 | 0.037 | 0.228 | 0.259 | 0.059 | 0.173 | 0.120 | 0.108 | 0.158 |
| PaiNN | 0.333 | 2.170 | 0.279 | 1.693 | 0.536 | 0.193 | 0.558 | 0.200 | 0.134 | 0.258 |
| Uni-Mol-R (1 conf) | 0.015 | 0.134 | 0.001 | 0.148 | 0.165 | 0.004 | 0.013 | 0.023 | 0.012 | 0.079 |
| Uni-Mol (1 conf) | 0.006 | 0.075 | 0.017 | 0.161 | 0.190 | 0.012 | 0.013 | 0.069 | 0.011 | 0.082 |
| ResNet18-I-R | 0.011 | 0.972 | 0.026 | 0.181 | 0.089 | 0.106 | 0.170 | 0.072 | 0.078 | 0.178 |
| ImageMol | 0.022 | 1.743 | 0.035 | 0.605 | 0.519 | 0.317 | 0.326 | 0.121 | 0.319 | 0.187 |
| CGIP-Image | 0.019 | 1.598 | 0.030 | 1.923 | 0.278 | 0.307 | 0.107 | 0.202 | 0.753 | 0.175 |
| MaskMol | 0.003 | 0.008 | 0.002 | 0.350 | 0.175 | 0.004 | 0.016 | 0.015 | 0.011 | 0.062 |
| IEM-I | 0.014 | 1.901 | 0.023 | 1.000 | 0.652 | 0.513 | 0.498 | 0.121 | 0.069 | 0.308 |
| ResNet18-G-R | 0.011 | 0.509 | 0.013 | 0.504 | 0.290 | 0.073 | 0.037 | 0.051 | 0.118 | 0.092 |
| IEM-G (1 conf) | 0.025 | 2.382 | 0.020 | 0.934 | 0.404 | 0.147 | 0.309 | 0.083 | 0.219 | 0.274 |
| ViT-G-R | 0.006 | 0.112 | 0.006 | 0.133 | 0.104 | 0.015 | 0.009 | 0.029 | 0.024 | 0.083 |
| VideoMol-G | 0.005 | 0.083 | 0.010 | 0.158 | 0.156 | 0.071 | 0.013 | 0.045 | 0.020 | 0.090 |
| ResNet18-V-R | 0.011 | 0.382 | 0.009 | 0.313 | 0.246 | 0.116 | 0.050 | 0.043 | 0.053 | 0.039 |
| IEM-V | 0.012 | 1.533 | 0.015 | 1.862 | 0.596 | 0.402 | 0.073 | 0.176 | 0.174 | 0.071 |

Table S35: The average RMSE performance on 10 complete chain (CC) datasets from StructNet. CC#1, CC#2, CC#3, CC#4, CC#5, CC#6, CC#7, CC#8, CC#9, CC#10 represent CHEMBL4649949_Potency, CHEMBL4649948_Potency, CHEMBL4649955_Potency, CHEMBL4888485_Inhibition, CHEMBL4296187_Inhibition, CHEMBL4296188_Inhibition, CHEMBL4296802_Inhibition, CHEMBL1614459_Potency, CHEMBL1614458_Potency, CHEMBL1614530_Potency, respectively. The green background represents top-5 performance.

| Models | CC#1 | CC#2 | CC#3 | CC#4 | CC#5 | CC#6 | CC#7 | CC#8 | CC#9 | CC#10 | Mean |
|---|---|---|---|---|---|---|---|---|---|---|---|
| mcfp4_2048 | 17.626 | 14.213 | 11.076 | 9.284 | 18.121 | 5.909 | 14.239 | 1.051 | 0.602 | 0.948 | 9.307 |
| ecfp4_2048 | 17.678 | 14.256 | 11.091 | 9.291 | 18.067 | 5.858 | 14.282 | 1.064 | 0.625 | 0.964 | 9.318 |
| maccs | 17.395 | 14.265 | 11.041 | 9.247 | 18.068 | 5.537 | 14.311 | 0.995 | 0.537 | 0.885 | 9.228 |
| physchem | 17.742 | 14.407 | 11.040 | 9.257 | 18.084 | 5.905 | 14.344 | 0.948 | 0.451 | 0.841 | 9.302 |
| atompair_2048 | 17.660 | 14.087 | 11.088 | 9.285 | 18.097 | 5.817 | 14.034 | 1.064 | 0.534 | 0.874 | 9.254 |
| rdkDes | 17.751 | 14.402 | 11.014 | 9.271 | 18.110 | 5.851 | 14.206 | 0.912 | 0.433 | 0.894 | 9.284 |
| BERT-6L | 17.709 | 14.406 | 11.045 | 9.275 | 18.169 | 5.879 | 14.406 | 0.934 | 0.435 | 0.824 | 9.308 |
| BERT-8L | 17.712 | 14.406 | 11.045 | 9.274 | 18.165 | 5.874 | 14.404 | 0.934 | 0.433 | 0.823 | 9.307 |
| RoBERTa-12L | 17.737 | 14.407 | 11.045 | 9.272 | 18.266 | 5.870 | 14.409 | 0.934 | 0.434 | 0.823 | 9.320 |
| molformer-R | 17.726 | 14.407 | 11.046 | 9.268 | 18.193 | 5.870 | 14.374 | 0.934 | 0.434 | 0.824 | 9.308 |
| Chem-BERT-6L | 17.705 | 14.406 | 11.046 | 9.272 | 18.180 | 5.872 | 14.396 | 0.934 | 0.434 | 0.824 | 9.307 |
| Chem-BERT-8L | 17.713 | 14.406 | 11.045 | 9.271 | 18.159 | 5.874 | 14.388 | 0.934 | 0.433 | 0.823 | 9.305 |
| CHEM-RoBERTa-12L | 17.733 | 14.406 | 11.044 | 9.280 | 18.303 | 5.868 | 14.455 | 0.934 | 0.434 | 0.823 | 9.328 |
| Molformer | 17.718 | 14.407 | 11.046 | 9.276 | 18.197 | 5.870 | 14.383 | 0.934 | 0.432 | 0.823 | 9.309 |
| GIN-R | 17.559 | 14.186 | 11.039 | 9.372 | 18.325 | 5.641 | 14.118 | 0.938 | 0.447 | 0.831 | 9.246 |
| EdgePred | 17.914 | 14.217 | 11.055 | 9.288 | 18.701 | 5.552 | 14.202 | 0.957 | 0.438 | 0.821 | 9.314 |
| ContextPred | 17.841 | 14.436 | 11.087 | 9.291 | 18.628 | 5.543 | 14.196 | 0.972 | 0.429 | 0.821 | 9.324 |
| infomax | 18.285 | 14.604 | 11.040 | 9.211 | 18.331 | 5.527 | 14.115 | 0.995 | 0.438 | 0.814 | 9.336 |
| masking | 18.085 | 14.417 | 11.029 | 9.287 | 18.432 | 5.604 | 14.120 | 0.973 | 0.444 | 0.822 | 9.321 |
| MolCLR | 17.590 | 14.171 | 10.998 | 9.299 | 18.305 | 5.625 | 14.098 | 0.946 | 0.437 | 0.825 | 9.229 |
| MoleBERT | 18.257 | 14.654 | 11.208 | 9.234 | 18.375 | 5.663 | 14.139 | 0.998 | 0.456 | 0.880 | 9.386 |
| CGIP-Graph | 17.800 | 14.195 | 10.993 | 11.150 | 18.333 | 5.657 | 14.238 | 0.980 | 0.429 | 0.840 | 9.462 |
| GraphMVP | 17.589 | 14.122 | 11.025 | 9.320 | 18.422 | 5.567 | 14.088 | 0.946 | 0.441 | 0.827 | 9.235 |
| SchNet | 17.605 | 14.351 | 11.030 | 9.314 | 18.156 | 5.638 | 14.338 | 0.951 | 0.438 | 0.826 | 9.265 |
| EGNN | 17.736 | 14.414 | 11.045 | 9.284 | 18.399 | 5.855 | 14.715 | 0.935 | 0.436 | 0.824 | 9.364 |
| TFN | 17.719 | 14.240 | 11.012 | 9.198 | 17.996 | 5.797 | 14.208 | 0.999 | 0.455 | 0.865 | 9.249 |
| SE3_Transformer | 17.699 | 14.283 | 11.047 | 9.337 | 17.923 | 5.872 | 14.104 | 1.000 | 0.468 | 0.879 | 9.261 |
| PaiNN | - | 14.604 | 11.048 | 9.470 | 18.300 | 5.267 | 14.461 | 1.650 | 0.659 | 1.592 | - |
| Uni-Mol-R (1 conf) | 17.703 | 14.406 | 11.045 | 9.297 | 18.126 | 5.681 | 14.355 | 0.937 | 0.434 | 0.824 | 9.281 |
| Uni-Mol (1 conf) | 17.690 | 14.405 | 11.037 | 9.329 | 18.129 | 5.621 | 14.353 | 0.938 | 0.435 | 0.824 | 9.276 |
| ResNet18-I-R | 17.658 | 14.169 | 11.066 | 9.215 | 18.126 | 5.809 | 14.346 | 0.936 | 0.435 | 0.831 | 9.259 |
| ImageMol | 17.655 | 14.236 | 11.068 | 9.167 | 18.191 | 5.781 | 14.471 | 0.939 | 0.437 | 0.839 | 9.278 |
| CGIP-Image | 17.713 | 14.285 | 11.023 | 9.246 | 18.172 | 5.695 | 14.264 | 0.939 | 0.432 | 0.835 | 9.260 |
| MaskMol | 17.702 | 14.406 | 11.046 | 9.274 | 18.224 | 5.876 | 14.478 | 0.935 | 0.434 | 0.823 | 9.320 |
| IEM-I | 17.659 | 14.279 | 11.029 | 9.270 | 18.099 | 5.757 | 14.319 | 0.941 | 0.439 | 0.826 | 9.262 |
| ResNet18-G-R | 17.744 | 14.659 | 10.991 | 9.284 | 18.198 | 5.802 | 14.534 | 0.935 | 0.435 | 0.823 | 9.340 |
| IEM-G (1 conf) | 17.664 | 14.552 | 11.079 | 9.221 | 18.244 | 5.743 | 14.632 | 0.937 | 0.436 | 0.835 | 9.334 |
| ViT-G-R | 17.722 | 14.410 | 11.043 | 9.253 | 18.166 | 5.873 | 14.397 | 0.934 | 0.434 | 0.824 | 9.306 |
| VideoMol-G | 17.726 | 14.404 | 11.046 | 9.260 | 18.201 | 5.853 | 14.394 | 0.934 | 0.433 | 0.824 | 9.308 |
| ResNet18-V-R | 17.852 | 14.613 | 10.966 | 9.309 | 18.343 | 5.709 | 14.537 | 0.939 | 0.431 | 0.823 | 9.352 |
| IEM-V | 17.768 | 14.611 | 11.010 | 9.282 | 18.352 | 5.627 | 14.474 | 0.939 | 0.434 | 0.825 | 9.332 |

Table S36: The standard deviation on 10 complete chain (CC) datasets from StructNet. CC#1, CC#2, CC#3, CC#4, CC#5, CC#6, CC#7, CC#8, CC#9, CC#10 represent CHEMBL4649949_Potency, CHEMBL4649948_Potency, CHEMBL4649955_Potency, CHEMBL4888485_Inhibition, CHEMBL4296187_Inhibition, CHEMBL4296188_Inhibition, CHEMBL4296802_Inhibition, CHEMBL1614459_Potency, CHEMBL1614458_Potency, CHEMBL1614530_Potency, respectively.

| Models | CC#1 | CC#2 | CC#3 | CC#4 | CC#5 | CC#6 | CC#7 | CC#8 | CC#9 | CC#10 |
|---|---|---|---|---|---|---|---|---|---|---|
| mcfp4_2048 | 0.007 | 0.001 | 0.001 | 0.009 | 0.004 | 0.005 | 0.003 | 0.002 | 0.002 | 0.002 |
| ecfp4_2048 | 0.011 | 0.001 | 0.002 | 0.016 | 0.006 | 0.006 | 0.008 | 0.002 | 0.003 | 0.003 |
| maccs | 0.005 | 0.004 | 0.004 | 0.020 | 0.005 | 0.031 | 0.004 | 0.002 | 0.008 | 0.006 |
| physchem | 0.005 | 0.001 | 0.001 | 0.016 | 0.001 | 0.006 | 0.001 | 0.003 | 0.002 | 0.002 |
| atompair_2048 | 0.005 | 0.004 | 0.005 | 0.015 | 0.001 | 0.131 | 0.005 | 0.013 | 0.003 | 0.005 |
| rdkDes | 0.004 | 0.000 | 0.003 | 0.003 | 0.001 | 0.004 | 0.002 | 0.002 | 0.000 | 0.002 |
| BERT-6L | 0.016 | 0.000 | 0.002 | 0.009 | 0.073 | 0.008 | 0.113 | 0.000 | 0.003 | 0.000 |
| BERT-8L | 0.017 | 0.000 | 0.003 | 0.016 | 0.074 | 0.013 | 0.111 | 0.001 | 0.003 | 0.000 |
| RoBERTa-12L | 0.067 | 0.000 | 0.003 | 0.015 | 0.096 | 0.025 | 0.123 | 0.001 | 0.005 | 0.000 |
| molformer-R | 0.024 | 0.000 | 0.001 | 0.017 | 0.037 | 0.014 | 0.056 | 0.000 | 0.004 | 0.001 |
| Chem-BERT-6L | 0.013 | 0.000 | 0.002 | 0.012 | 0.080 | 0.008 | 0.106 | 0.000 | 0.002 | 0.000 |
| Chem-BERT-8L | 0.018 | 0.000 | 0.002 | 0.012 | 0.084 | 0.012 | 0.109 | 0.000 | 0.003 | 0.000 |
| CHEM-RoBERTa-12L | 0.069 | 0.000 | 0.005 | 0.009 | 0.094 | 0.018 | 0.112 | 0.001 | 0.004 | 0.000 |
| Molformer | 0.027 | 0.000 | 0.003 | 0.013 | 0.044 | 0.014 | 0.047 | 0.000 | 0.003 | 0.000 |
| GIN-R | 0.296 | 0.125 | 0.041 | 0.105 | 0.202 | 0.096 | 0.159 | 0.015 | 0.021 | 0.009 |
| EdgePred | 0.268 | 0.160 | 0.100 | 0.059 | 0.282 | 0.092 | 0.208 | 0.012 | 0.005 | 0.008 |
| ContextPred | 0.383 | 0.256 | 0.108 | 0.078 | 0.276 | 0.073 | 0.221 | 0.014 | 0.006 | 0.010 |
| infomax | 0.479 | 0.205 | 0.087 | 0.072 | 0.169 | 0.082 | 0.171 | 0.025 | 0.009 | 0.018 |
| masking | 0.306 | 0.263 | 0.090 | 0.047 | 0.177 | 0.113 | 0.247 | 0.009 | 0.012 | 0.018 |
| MolCLR | 0.230 | 0.176 | 0.046 | 0.074 | 0.139 | 0.091 | 0.161 | 0.013 | 0.020 | 0.005 |
| MoleBERT | 0.563 | 0.180 | 0.116 | 0.092 | 0.153 | 0.179 | 0.090 | 0.023 | 0.017 | 0.031 |
| CGIP-Graph | 0.282 | 0.187 | 0.065 | 2.305 | 0.183 | 0.153 | 0.338 | 0.020 | 0.007 | 0.037 |
| GraphMVP | 0.233 | 0.122 | 0.048 | 0.083 | 0.291 | 0.097 | 0.224 | 0.018 | 0.015 | 0.009 |
| SchNet | 0.144 | 0.165 | 0.041 | 0.106 | 0.117 | 0.227 | 0.278 | 0.011 | 0.011 | 0.009 |
| EGNN | 0.122 | 0.080 | 0.058 | 0.046 | 0.628 | 0.060 | 0.473 | 0.001 | 0.007 | 0.001 |
| TFN | 0.259 | 0.264 | 0.053 | 0.109 | 0.161 | 0.116 | 0.114 | 0.023 | 0.012 | 0.026 |
| SE3_Transformer | 0.069 | 0.112 | 0.041 | 0.131 | 0.083 | 0.088 | 0.056 | 0.026 | 0.021 | 0.035 |
| PaiNN | - | 0.265 | 0.092 | 0.098 | 0.096 | 0.120 | 0.168 | 0.156 | 0.106 | 0.195 |
| Uni-Mol-R (1 conf) | 0.029 | 0.001 | 0.003 | 0.040 | 0.033 | 0.188 | 0.054 | 0.003 | 0.003 | 0.000 |
| Uni-Mol (1 conf) | 0.041 | 0.004 | 0.014 | 0.103 | 0.036 | 0.146 | 0.058 | 0.005 | 0.003 | 0.000 |
| ResNet18-I-R | 0.296 | 0.139 | 0.051 | 0.130 | 0.120 | 0.075 | 0.198 | 0.007 | 0.012 | 0.008 |
| ImageMol | 0.232 | 0.297 | 0.092 | 0.136 | 0.271 | 0.078 | 0.323 | 0.009 | 0.005 | 0.019 |
| CGIP-Image | 0.217 | 0.162 | 0.025 | 0.119 | 0.122 | 0.100 | 0.143 | 0.009 | 0.005 | 0.016 |
| MaskMol | 0.006 | 0.001 | 0.005 | 0.013 | 0.057 | 0.008 | 0.193 | 0.001 | 0.002 | 0.001 |
| IEM-I | 0.249 | 0.290 | 0.081 | 0.083 | 0.231 | 0.076 | 0.254 | 0.006 | 0.009 | 0.008 |
| ResNet18-G-R | 0.135 | 0.195 | 0.041 | 0.057 | 0.109 | 0.041 | 0.200 | 0.009 | 0.006 | 0.005 |
| IEM-G (1 conf) | 0.145 | 0.121 | 0.113 | 0.056 | 0.114 | 0.094 | 0.391 | 0.009 | 0.007 | 0.014 |
| ViT-G-R | 0.032 | 0.010 | 0.006 | 0.045 | 0.083 | 0.019 | 0.116 | 0.000 | 0.001 | 0.001 |
| VideoMol-G | 0.037 | 0.012 | 0.003 | 0.046 | 0.124 | 0.023 | 0.080 | 0.001 | 0.003 | 0.001 |
| ResNet18-V-R | 0.152 | 0.126 | 0.023 | 0.051 | 0.098 | 0.051 | 0.249 | 0.004 | 0.005 | 0.002 |
| IEM-V | 0.115 | 0.107 | 0.027 | 0.093 | 0.207 | 0.156 | 0.171 | 0.006 | 0.010 | 0.009 |

Table S37: The average RMSE performance on 10 macro (M) datasets from StructNet. M#1, M#2, M#3, M#4, M#5, M#6, M#7, M#8, M#9, M#10 represent CHEMBL4420282_IC50, CHEMBL4419606_IC50, CHEMBL4420281_In, CHEMBL3881498_In, CHEMBL4419605_In, CHEMBL4420271_In, CHEMBL4419595_In, CHEMBL3881499_IC50, CHEMBL4420273_In, CHEMBL4419597_In, respectively. OOM represents the Exception of Out Of Memory (OOM) on single GPU of RTX 4090 Ti (24G). The green background represents top-5 performance.

| Models | M#1 | M#2 | M#3 | M#4 | M#5 | M#6 | M#7 | M#8 | M#9 | M#10 | Mean |
|---|---|---|---|---|---|---|---|---|---|---|---|
| mcfp4_2048 | 0.842 | 0.709 | 36.331 | 35.451 | 36.827 | 26.782 | 26.682 | 0.564 | 10.862 | 10.851 | 18.590 |
| ecfp4_2048 | 0.710 | 0.713 | 36.158 | 33.518 | 36.615 | 26.653 | 26.561 | 0.575 | 10.713 | 10.688 | 18.290 |
| maccs | 0.869 | 0.786 | 36.413 | 34.200 | 36.770 | 26.679 | 26.492 | 0.601 | 10.637 | 10.616 | 18.406 |
| physchem | 1.076 | 0.831 | 35.838 | 32.765 | 36.639 | 26.662 | 26.570 | 0.600 | 10.980 | 10.952 | 18.291 |
| atompair_2048 | 0.612 | 0.628 | 35.734 | 33.803 | 35.983 | 26.669 | 26.585 | 0.503 | 11.601 | 11.564 | 18.368 |
| rdkDes | 1.041 | 0.832 | 40.833 | 36.886 | 41.905 | 29.449 | 29.207 | 0.606 | 11.388 | 11.377 | 20.352 |
| BERT-6L | 1.300 | 0.903 | 36.815 | 35.817 | 36.874 | 26.559 | 26.571 | 0.582 | 11.909 | 11.853 | 18.918 |
| BERT-8L | 1.306 | 0.898 | 36.823 | 35.839 | 36.876 | 26.580 | 26.568 | 0.583 | 11.872 | 11.878 | 18.922 |
| RoBERTa-12L | 1.329 | 0.906 | 37.413 | 36.289 | 37.288 | 26.644 | 26.636 | 0.583 | 11.894 | 11.878 | 19.086 |
| molformer-R | 1.337 | 0.906 | 37.330 | 36.212 | 37.327 | 26.557 | 26.612 | 0.583 | 11.919 | 11.867 | 19.065 |
| Chem-BERT-6L | 1.277 | 0.889 | 36.779 | 35.954 | 36.853 | 26.553 | 26.583 | 0.583 | 11.887 | 11.882 | 18.924 |
| Chem-BERT-8L | 1.290 | 0.887 | 36.737 | 35.725 | 36.862 | 26.594 | 26.552 | 0.582 | 11.894 | 11.883 | 18.901 |
| CHEM-RoBERTa-12L | 1.337 | 0.906 | 37.431 | 36.444 | 37.381 | 26.671 | 26.634 | 0.582 | 11.898 | 11.878 | 19.116 |
| Molformer | 1.304 | 0.898 | 37.262 | 36.103 | 37.371 | 26.536 | 26.597 | 0.583 | 11.893 | 11.862 | 19.041 |
| GIN-R | 0.779 | 0.721 | 35.136 | 32.007 | 35.215 | 26.727 | 26.231 | 0.552 | 11.798 | 11.928 | 18.109 |
| EdgePred | 0.748 | 0.723 | 35.367 | 31.874 | 35.741 | 26.849 | 26.630 | 0.543 | 11.657 | 11.320 | 18.145 |
| ContextPred | 0.746 | 0.721 | 35.374 | 32.078 | 35.355 | 27.409 | 26.911 | 0.543 | 11.047 | 10.866 | 18.105 |
| infomax | 0.703 | 0.719 | 35.799 | 33.396 | 35.739 | 27.274 | 27.130 | 0.540 | 10.738 | 10.743 | 18.278 |
| masking | 0.745 | 0.721 | 35.604 | 33.802 | 36.443 | 27.499 | 28.014 | 0.539 | 10.836 | 10.800 | 18.500 |
| MolCLR | 0.748 | 0.732 | 34.697 | 32.272 | 35.083 | 26.350 | 26.484 | 0.553 | 11.872 | 11.825 | 18.062 |
| MoleBERT | 0.720 | 0.715 | 35.893 | 34.593 | 37.110 | 27.696 | 27.373 | 0.549 | 11.054 | 11.034 | 18.674 |
| CGIP-Graph | 0.833 | 0.669 | 36.427 | 33.055 | 36.285 | 24.291 | 24.490 | 0.541 | 11.836 | 11.771 | 18.020 |
| GraphMVP | 0.653 | 0.666 | 34.532 | 31.745 | 35.121 | 26.295 | 26.539 | 0.544 | 11.773 | 12.108 | 17.998 |
| SchNet | 1.031 | 0.830 | 36.449 | 36.212 | 36.957 | 27.075 | 26.859 | 0.576 | 11.439 | 11.464 | 18.889 |
| EGNN | 1.239 | 0.878 | 36.260 | 35.416 | 37.008 | 26.572 | 26.741 | 0.583 | 11.923 | 11.885 | 18.851 |
| TFN | OOM | OOM | OOM | OOM | OOM | OOM | OOM | OOM | OOM | OOM | OOM |
| SE3_Transformer | OOM | OOM | OOM | OOM | OOM | OOM | OOM | OOM | OOM | OOM | OOM |
| PaiNN | 1.037 | 0.818 | 35.869 | 34.985 | 36.468 | 27.390 | 27.432 | 0.667 | 10.664 | 10.629 | 18.596 |
| Uni-Mol-R (1 conf) | 1.204 | 0.844 | 36.517 | 34.971 | 36.851 | 28.186 | 27.917 | 0.582 | 11.849 | 11.895 | 19.082 |
| Uni-Mol (1 conf) | 1.200 | 0.807 | 36.111 | 34.295 | 36.725 | 28.650 | 28.009 | 0.582 | 11.532 | 11.578 | 18.949 |
| ResNet18-I-R | 1.105 | 0.798 | 37.524 | 36.284 | 37.624 | 27.383 | 28.096 | 0.587 | 10.899 | 11.057 | 19.136 |
| ImageMol | 1.046 | 0.824 | 36.738 | 36.452 | 36.875 | 27.436 | 26.879 | 0.589 | 11.131 | 11.037 | 18.901 |
| CGIP-Image | 1.026 | 0.806 | 37.396 | 35.472 | 36.816 | 27.184 | 26.762 | 0.575 | 11.020 | 10.913 | 18.797 |
| MaskMol | 1.239 | 0.822 | 36.689 | 35.752 | 36.899 | 26.572 | 26.585 | 0.583 | 11.670 | 11.065 | 18.862 |
| IEM-I | 1.059 | 0.813 | 38.558 | 36.142 | 37.955 | 27.290 | 26.979 | 0.587 | 10.958 | 11.065 | 19.141 |
| ResNet18-G-R | 1.149 | 0.879 | 37.156 | 36.673 | 37.427 | 27.010 | 27.011 | 0.587 | 12.009 | 12.243 | 19.214 |
| IEM-G (1 conf) | 1.131 | 0.838 | 36.978 | 36.031 | 37.440 | 27.364 | 27.154 | 0.590 | 11.901 | 11.810 | 19.124 |
| ViT-G-R | 1.301 | 0.896 | 35.611 | 35.991 | 37.137 | 26.564 | 26.600 | 0.583 | 11.900 | 11.877 | 18.846 |
| VideoMol-G | 1.302 | 0.903 | 36.874 | 35.920 | 37.085 | 26.564 | 26.588 | 0.583 | 11.899 | 11.868 | 18.959 |
| ResNet18-V-R | 1.053 | 0.796 | 36.607 | 36.057 | 37.164 | 27.163 | 26.784 | 0.582 | 11.709 | 11.310 | 18.923 |
| IEM-V | 0.994 | 0.793 | 36.746 | 36.019 | 36.957 | 28.258 | 28.066 | 0.583 | 11.485 | 11.987 | 19.189 |

Table S38: The standard deviation on 10 macro (M) datasets from StructNet. M#1, M#2, M#3, M#4, M#5, M#6, M#7, M#8, M#9, M#10 represent CHEMBL4420282_IC50, CHEMBL4419606_IC50, CHEMBL4420281_In, CHEMBL3881498_In, CHEMBL4419605_In, CHEMBL4420271_In, CHEMBL4419595_In, CHEMBL3881499_IC50, CHEMBL4420273_In, CHEMBL4419597_In, respectively. OOM represents the Exception of Out Of Memory (OOM) on single GPU of RTX 4090 Ti (24G).

| Models | M#1 | M#2 | M#3 | M#4 | M#5 | M#6 | M#7 | M#8 | M#9 | M#10 |
|---|---|---|---|---|---|---|---|---|---|---|
| mcfp4_2048 | 0.006 | 0.010 | 0.019 | 0.009 | 0.017 | 0.008 | 0.010 | 0.008 | 0.007 | 0.009 |
| ecfp4_2048 | 0.010 | 0.007 | 0.028 | 0.008 | 0.018 | 0.015 | 0.011 | 0.016 | 0.003 | 0.004 |
| maccs | 0.004 | 0.002 | 0.025 | 0.013 | 0.014 | 0.015 | 0.032 | 0.007 | 0.003 | 0.004 |
| physchem | 0.001 | 0.002 | 0.017 | 0.008 | 0.020 | 0.010 | 0.010 | 0.002 | 0.013 | 0.012 |
| atompair_2048 | 0.004 | 0.013 | 0.007 | 0.016 | 0.006 | 0.047 | 0.005 | 0.003 | 0.016 | 0.022 |
| rdkDes | 0.003 | 0.000 | 0.026 | 0.018 | 0.025 | 0.017 | 0.015 | 0.000 | 0.011 | 0.015 |
| BERT-6L | 0.056 | 0.025 | 0.514 | 0.331 | 0.363 | 0.068 | 0.056 | 0.002 | 0.020 | 0.064 |
| BERT-8L | 0.045 | 0.025 | 0.496 | 0.320 | 0.352 | 0.146 | 0.053 | 0.003 | 0.046 | 0.031 |
| RoBERTa-12L | 0.015 | 0.009 | 0.599 | 0.752 | 0.569 | 0.207 | 0.099 | 0.002 | 0.025 | 0.018 |
| molformer-R | 0.019 | 0.009 | 0.513 | 0.820 | 0.624 | 0.057 | 0.093 | 0.002 | 0.016 | 0.020 |
| Chem-BERT-6L | 0.061 | 0.017 | 0.425 | 0.337 | 0.339 | 0.065 | 0.060 | 0.002 | 0.027 | 0.025 |
| Chem-BERT-8L | 0.058 | 0.013 | 0.364 | 0.346 | 0.320 | 0.124 | 0.044 | 0.002 | 0.038 | 0.031 |
| CHEM-RoBERTa-12L | 0.022 | 0.008 | 0.642 | 0.650 | 0.550 | 0.284 | 0.090 | 0.002 | 0.037 | 0.018 |
| Molformer | 0.064 | 0.018 | 0.501 | 0.567 | 0.512 | 0.043 | 0.076 | 0.003 | 0.051 | 0.036 |
| GIN-R | 0.054 | 0.019 | 1.238 | 1.170 | 1.317 | 0.685 | 1.049 | 0.020 | 0.439 | 0.491 |
| EdgePred | 0.051 | 0.037 | 1.405 | 1.179 | 1.225 | 0.446 | 0.774 | 0.013 | 0.859 | 0.390 |
| ContextPred | 0.030 | 0.032 | 1.071 | 1.146 | 0.721 | 1.136 | 0.778 | 0.007 | 0.206 | 0.118 |
| infomax | 0.023 | 0.020 | 1.333 | 2.169 | 0.763 | 0.955 | 0.588 | 0.012 | 0.118 | 0.146 |
| masking | 0.035 | 0.020 | 1.377 | 1.143 | 0.858 | 1.004 | 1.364 | 0.008 | 0.250 | 0.236 |
| MolCLR | 0.031 | 0.030 | 1.148 | 1.466 | 1.021 | 0.608 | 1.087 | 0.009 | 0.634 | 0.387 |
| MoleBERT | 0.024 | 0.036 | 0.785 | 2.217 | 0.208 | 0.983 | 0.999 | 0.012 | 0.372 | 0.306 |
| CGIP-Graph | 0.030 | 0.018 | 0.971 | 2.610 | 0.898 | 0.328 | 0.684 | 0.015 | 0.624 | 0.789 |
| GraphMVP | 0.010 | 0.024 | 1.335 | 0.634 | 0.642 | 1.095 | 0.554 | 0.011 | 0.485 | 0.773 |
| SchNet | 0.049 | 0.024 | 0.318 | 0.924 | 0.237 | 0.319 | 0.412 | 0.003 | 0.336 | 0.412 |
| EGNN | 0.055 | 0.046 | 0.311 | 0.984 | 0.490 | 0.071 | 0.521 | 0.001 | 0.061 | 0.093 |
| TFN | OOM | OOM | OOM | OOM | OOM | OOM | OOM | OOM | OOM | OOM |
| SE3_Transformer | OOM | OOM | OOM | OOM | OOM | OOM | OOM | OOM | OOM | OOM |
| PaiNN | 0.137 | 0.074 | 0.462 | 0.869 | 0.328 | 0.346 | 0.158 | 0.039 | 0.174 | 0.253 |
| Uni-Mol-R (1 conf) | 0.034 | 0.023 | 0.264 | 0.943 | 0.434 | 0.840 | 1.233 | 0.001 | 0.097 | 0.029 |
| Uni-Mol (1 conf) | 0.082 | 0.021 | 0.616 | 0.706 | 0.429 | 0.499 | 1.006 | 0.002 | 0.397 | 0.315 |
| ResNet18-I-R | 0.103 | 0.033 | 1.086 | 1.646 | 0.763 | 0.727 | 0.773 | 0.012 | 0.316 | 0.467 |
| ImageMol | 0.100 | 0.043 | 1.841 | 1.757 | 1.742 | 1.004 | 1.180 | 0.030 | 0.373 | 0.696 |
| CGIP-Image | 0.062 | 0.055 | 1.532 | 0.893 | 1.185 | 0.890 | 0.833 | 0.020 | 0.594 | 0.591 |
| MaskMol | 0.069 | 0.038 | 0.462 | 0.285 | 0.304 | 0.071 | 0.056 | 0.002 | 0.302 | 0.200 |
| IEM-I | 0.084 | 0.040 | 0.810 | 2.070 | 0.928 | 1.225 | 0.823 | 0.016 | 0.374 | 0.470 |
| ResNet18-G-R | 0.092 | 0.037 | 0.565 | 1.196 | 0.565 | 0.444 | 0.648 | 0.009 | 0.638 | 0.483 |
| IEM-G (1 conf) | 0.073 | 0.029 | 1.173 | 1.450 | 0.720 | 0.948 | 0.665 | 0.020 | 0.595 | 0.336 |
| ViT-G-R | 0.061 | 0.017 | 0.082 | 0.379 | 0.344 | 0.054 | 0.047 | 0.003 | 0.019 | 0.022 |
| VideoMol-G | 0.053 | 0.024 | 0.541 | 0.467 | 0.437 | 0.054 | 0.068 | 0.002 | 0.050 | 0.032 |
| ResNet18-V-R | 0.051 | 0.022 | 0.567 | 0.846 | 0.585 | 0.523 | 0.373 | 0.005 | 0.372 | 0.477 |
| IEM-V | 0.039 | 0.033 | 0.820 | 1.774 | 0.970 | 0.814 | 0.891 | 0.010 | 0.524 | 0.647 |

Table S39: The average RMSE performance on 10 macrocyclic peptide (MP) datasets from StructNet. MP#1, MP#2, MP#3, MP#4, MP#5, MP#6, MP#7, MP#8, MP#9, MP#10 represent CHEMBL4888485_Inhibition, CHEMBL2354301_AC50, CHEMBL3880198_Ki, CHEMBL4420271_Inhibition, CHEMBL4419595_Inhibition, CHEMBL4420282_IC50, CHEMBL3214979_AC50, CHEMBL4420277_Inhibition, CHEMBL4419601_Inhibition, CHEMBL4419606_IC50, respectively. OOM represents the Exception of Out Of Memory (OOM) on single GPU of RTX 4090 Ti (24G). The green background represents top-5 performance.

| Models | MP#1 | MP#2 | MP#3 | MP#4 | MP#5 | MP#6 | MP#7 | MP#8 | MP#9 | MP#10 | Mean |
|---|---|---|---|---|---|---|---|---|---|---|---|
| mcfp4_2048 | 10.134 | 0.405 | 1.012 | 25.835 | 25.835 | 0.838 | 0.293 | 15.057 | 15.057 | 0.970 | 9.544 |
| ecfp4_2048 | 10.227 | 0.624 | 1.055 | 26.031 | 26.031 | 0.874 | 0.467 | 15.113 | 15.113 | 1.020 | 9.656 |
| maccs | 10.172 | 0.374 | 0.923 | 25.612 | 25.612 | 0.904 | 0.274 | 14.935 | 14.935 | 1.001 | 9.474 |
| physchem | 10.157 | 0.374 | 1.037 | 29.923 | 29.923 | 0.964 | 0.564 | 16.133 | 16.133 | 0.998 | 10.621 |
| atompair_2048 | 10.034 | 0.422 | 0.921 | 25.458 | 25.458 | 0.911 | 0.278 | 14.918 | 14.918 | 1.041 | 9.436 |
| rdkDes | 10.169 | 0.381 | 1.044 | 32.574 | 32.574 | 1.050 | 0.933 | 17.745 | 17.745 | 1.085 | 11.530 |
| BERT-6L | 10.129 | 0.371 | 0.866 | 25.152 | 25.152 | 0.963 | 0.271 | 15.017 | 15.017 | 0.976 | 9.391 |
| BERT-8L | 10.124 | 0.371 | 0.866 | 25.132 | 25.132 | 0.963 | 0.272 | 15.005 | 15.005 | 0.980 | 9.385 |
| RoBERTa-12L | 10.124 | 0.371 | 0.873 | 24.697 | 24.697 | 0.963 | 0.270 | 15.522 | 15.522 | 0.942 | 9.398 |
| molformer-R | 10.130 | 0.371 | 0.866 | 24.772 | 24.772 | 0.963 | 0.274 | 15.571 | 15.571 | 0.955 | 9.424 |
| Chem-BERT-6L | 10.130 | 0.371 | 0.863 | 25.203 | 25.203 | 0.963 | 0.272 | 15.009 | 15.009 | 0.985 | 9.401 |
| Chem-BERT-8L | 10.128 | 0.371 | 0.872 | 25.183 | 25.183 | 0.963 | 0.273 | 15.050 | 15.050 | 0.957 | 9.403 |
| CHEM-RoBERTa-12L | 10.120 | 0.371 | 0.872 | 25.312 | 25.312 | 0.963 | 0.267 | 15.500 | 15.500 | 0.934 | 9.515 |
| Molformer | 10.130 | 0.371 | 0.865 | 25.667 | 25.667 | 0.963 | 0.268 | 15.038 | 15.038 | 0.948 | 9.496 |
| GIN-R | 10.526 | 0.385 | 0.771 | 27.702 | 27.007 | 1.191 | 0.261 | 19.136 | 21.544 | 1.034 | 10.956 |
| EdgePred | 10.410 | 0.359 | 0.774 | 25.692 | 25.676 | 0.886 | 0.265 | 14.951 | 14.917 | 0.956 | 9.489 |
| ContextPred | 10.156 | 0.366 | 0.791 | 25.773 | 25.807 | 0.814 | 0.273 | 14.912 | 14.923 | 0.955 | 9.477 |
| infomax | 10.070 | 0.340 | 0.823 | 24.658 | 24.503 | 0.817 | 0.286 | 14.716 | 14.612 | 0.946 | 9.177 |
| masking | 10.241 | 0.359 | 0.793 | 25.620 | 25.734 | 0.841 | 0.276 | 14.906 | 14.905 | 0.933 | 9.461 |
| MolCLR | 10.407 | 0.347 | 0.768 | 25.422 | 25.268 | 0.854 | 0.261 | 15.241 | 15.083 | 0.954 | 9.461 |
| MoleBERT | 10.059 | 0.352 | 0.863 | 25.321 | 25.316 | 0.806 | 0.284 | 14.893 | 14.895 | 0.937 | 9.373 |
| CGIP-Graph | 10.297 | 0.400 | 0.802 | 25.667 | 25.939 | 0.867 | 0.274 | 14.617 | 14.627 | 0.890 | 9.438 |
| GraphMVP | 10.454 | 0.385 | 0.789 | 25.686 | 25.381 | 0.864 | 0.264 | 15.423 | 15.122 | 0.990 | 9.536 |
| SchNet | 10.414 | 0.399 | 0.881 | 25.031 | 25.274 | 0.989 | 0.302 | 15.002 | 14.996 | 0.955 | 9.424 |
| EGNN | 10.117 | 0.371 | 0.877 | 26.386 | 26.386 | 0.963 | 0.272 | 15.245 | 15.245 | 0.929 | 9.679 |
| TFN | 10.106 | 0.364 | 0.868 | OOM | OOM | OOM | 0.363 | OOM | OOM | OOM | - |
| SE3_Transformer | 10.370 | 0.341 | 0.742 | OOM | OOM | OOM | 0.320 | OOM | OOM | OOM | - |
| PaiNN | 10.626 | 0.434 | 0.909 | 25.160 | 25.260 | 1.234 | 0.495 | 14.880 | 14.897 | 1.178 | 9.507 |
| Uni-Mol-R (1 conf) | 10.133 | 0.371 | 0.857 | 25.031 | 25.031 | 0.963 | 0.272 | 15.572 | 15.572 | 0.958 | 9.476 |
| Uni-Mol (1 conf) | 10.134 | 0.369 | 0.864 | 24.991 | 24.991 | 0.963 | 0.273 | 15.457 | 15.457 | 0.955 | 9.445 |
| ResNet18-I-R | 10.024 | 0.377 | 0.898 | 24.988 | 24.988 | 0.963 | 0.286 | 14.862 | 14.862 | 0.964 | 9.321 |
| ImageMol | 10.373 | 0.378 | 0.846 | 24.654 | 24.654 | 0.993 | 0.286 | 15.471 | 15.471 | 0.917 | 9.404 |
| CGIP-Image | 10.222 | 0.370 | 0.877 | 24.718 | 24.718 | 0.962 | 0.284 | 15.036 | 15.036 | 0.857 | 9.308 |
| MaskMol | 10.124 | 0.371 | 0.860 | 24.771 | 24.771 | 0.963 | 0.271 | 15.005 | 15.005 | 0.979 | 9.312 |
| IEM-I | 10.155 | 0.373 | 0.864 | 24.413 | 24.413 | 0.961 | 0.278 | 15.411 | 15.411 | 0.906 | 9.319 |
| ResNet18-G-R | 10.075 | 0.375 | 0.882 | 25.039 | 24.956 | 0.971 | 0.275 | 14.958 | 14.939 | 0.959 | 9.343 |
| IEM-G (1 conf) | 10.139 | 0.377 | 0.910 | 25.727 | 25.148 | 0.997 | 0.278 | 15.290 | 15.137 | 0.969 | 9.497 |
| ViT-G-R | 10.120 | 0.371 | 0.869 | 25.615 | 25.637 | 0.963 | 0.268 | 15.121 | 15.151 | 0.961 | 9.508 |
| VideoMol-G | 10.133 | 0.372 | 0.870 | 24.677 | 24.677 | 0.962 | 0.273 | 15.649 | 15.607 | 0.968 | 9.419 |
| ResNet18-V-R | 9.968 | 0.373 | 0.871 | 24.936 | 25.080 | 0.965 | 0.274 | 14.978 | 14.974 | 0.966 | 9.339 |
| IEM-V | 10.086 | 0.380 | 0.946 | 25.360 | 25.621 | 1.007 | 0.282 | 15.320 | 15.463 | 0.995 | 9.546 |

Table S40: The standard deviation on 10 macrocyclic peptide (MP) datasets from Struct-Net. MP#1, MP#2, MP#3, MP#4, MP#5, MP#6, MP#7, MP#8, MP#9, MP#10 represent CHEMBL4888485_Inhibition, CHEMBL2354301_AC50, CHEMBL3880198_Ki, CHEMBL4420271_Inhibition, CHEMBL4419595_Inhibition, CHEMBL4420282_IC50, CHEMBL3214979_AC50, CHEMBL4420277_Inhibition, CHEMBL4419601_Inhibition, CHEMBL4419606_IC50, respectively. OOM represents the Exception of Out Of Memory (OOM) on single GPU of RTX 4090 Ti (24G).

| Models | MP#1 | MP#2 | MP#3 | MP#4 | MP#5 | MP#6 | MP#7 | MP#8 | MP#9 | MP#10 |
|---|---|---|---|---|---|---|---|---|---|---|
| mcfp4_2048 | 0.015 | 0.005 | 0.014 | 0.057 | 0.057 | 0.002 | 0.002 | 0.003 | 0.003 | 0.008 |
| ecfp4_2048 | 0.012 | 0.026 | 0.018 | 0.008 | 0.008 | 0.001 | 0.004 | 0.005 | 0.005 | 0.003 |
| maccs | 0.008 | 0.007 | 0.014 | 0.042 | 0.042 | 0.004 | 0.004 | 0.006 | 0.006 | 0.017 |
| physchem | 0.006 | 0.001 | 0.005 | 0.050 | 0.050 | 0.000 | 0.043 | 0.035 | 0.035 | 0.012 |
| atompair_2048 | 0.037 | 0.010 | 0.006 | 0.012 | 0.012 | 0.001 | 0.009 | 0.007 | 0.007 | 0.008 |
| rdkDes | 0.002 | 0.000 | 0.004 | 0.026 | 0.026 | 0.011 | 0.033 | 0.022 | 0.022 | 0.018 |
| BERT-6L | 0.006 | 0.000 | 0.012 | 0.189 | 0.189 | 0.000 | 0.003 | 0.110 | 0.110 | 0.038 |
| BERT-8L | 0.014 | 0.000 | 0.018 | 0.198 | 0.198 | 0.000 | 0.004 | 0.136 | 0.136 | 0.035 |
| RoBERTa-12L | 0.006 | 0.001 | 0.022 | 0.020 | 0.020 | 0.000 | 0.017 | 0.357 | 0.357 | 0.033 |
| molformer-R | 0.007 | 0.001 | 0.011 | 0.038 | 0.038 | 0.000 | 0.011 | 0.368 | 0.368 | 0.033 |
| Chem-BERT-6L | 0.007 | 0.001 | 0.013 | 0.170 | 0.170 | 0.000 | 0.003 | 0.106 | 0.106 | 0.019 |
| Chem-BERT-8L | 0.003 | 0.000 | 0.013 | 0.241 | 0.241 | 0.000 | 0.005 | 0.129 | 0.129 | 0.040 |
| CHEM-RoBERTa-12L | 0.010 | 0.001 | 0.011 | 1.380 | 1.380 | 0.000 | 0.015 | 0.471 | 0.471 | 0.051 |
| Molformer | 0.005 | 0.001 | 0.011 | 0.490 | 0.490 | 0.000 | 0.008 | 0.115 | 0.115 | 0.023 |
| GIN-R | 0.366 | 0.030 | 0.077 | 1.631 | 1.546 | 0.153 | 0.014 | 3.520 | 2.601 | 0.039 |
| EdgePred | 0.604 | 0.010 | 0.027 | 0.178 | 0.163 | 0.113 | 0.013 | 0.049 | 0.030 | 0.060 |
| ContextPred | 0.094 | 0.014 | 0.032 | 0.082 | 0.040 | 0.021 | 0.008 | 0.026 | 0.026 | 0.062 |
| infomax | 0.085 | 0.010 | 0.067 | 0.364 | 0.707 | 0.038 | 0.011 | 0.329 | 0.289 | 0.061 |
| masking | 0.458 | 0.013 | 0.084 | 0.929 | 0.170 | 0.043 | 0.014 | 0.025 | 0.020 | 0.038 |
| MolCLR | 0.218 | 0.016 | 0.070 | 0.542 | 0.314 | 0.062 | 0.005 | 0.559 | 0.429 | 0.038 |
| MoleBERT | 0.111 | 0.014 | 0.062 | 0.238 | 0.245 | 0.010 | 0.011 | 0.017 | 0.018 | 0.027 |
| CGIP-Graph | 0.429 | 0.027 | 0.087 | 1.038 | 0.644 | 0.038 | 0.013 | 0.335 | 0.151 | 0.067 |
| GraphMVP | 0.345 | 0.012 | 0.067 | 0.957 | 0.812 | 0.057 | 0.016 | 0.924 | 0.550 | 0.057 |
| SchNet | 0.309 | 0.006 | 0.083 | 0.635 | 0.814 | 0.025 | 0.008 | 0.184 | 0.228 | 0.028 |
| EGNN | 0.141 | 0.001 | 0.047 | 1.582 | 1.582 | 0.000 | 0.004 | 0.430 | 0.430 | 0.046 |
| TFN | 0.161 | 0.023 | 0.128 | OOM | OOM | OOM | 0.032 | OOM | OOM | OOM |
| SE3_Transformer | 0.188 | 0.018 | 0.054 | OOM | OOM | OOM | 0.013 | OOM | OOM | OOM |
| PaiNN | 0.225 | 0.018 | 0.063 | 0.413 | 0.467 | 0.311 | 0.079 | 0.224 | 0.206 | 0.315 |
| Uni-Mol-R (1 conf) | 0.010 | 0.001 | 0.028 | 0.054 | 0.054 | 0.000 | 0.003 | 0.136 | 0.136 | 0.033 |
| Uni-Mol (1 conf) | 0.018 | 0.003 | 0.039 | 0.063 | 0.063 | 0.000 | 0.005 | 0.262 | 0.262 | 0.040 |
| ResNet18-I-R | 0.272 | 0.009 | 0.034 | 0.310 | 0.310 | 0.012 | 0.017 | 0.083 | 0.083 | 0.033 |
| ImageMol | 0.489 | 0.020 | 0.066 | 1.230 | 1.230 | 0.075 | 0.014 | 0.655 | 0.655 | 0.092 |
| CGIP-Image | 0.278 | 0.007 | 0.060 | 0.831 | 0.831 | 0.035 | 0.020 | 0.760 | 0.760 | 0.022 |
| MaskMol | 0.018 | 0.002 | 0.016 | 0.098 | 0.098 | 0.001 | 0.004 | 0.119 | 0.119 | 0.034 |
| IEM-I | 0.288 | 0.010 | 0.055 | 1.126 | 1.126 | 0.032 | 0.010 | 0.564 | 0.564 | 0.048 |
| ResNet18-G-R | 0.073 | 0.008 | 0.036 | 0.235 | 0.296 | 0.011 | 0.010 | 0.112 | 0.150 | 0.035 |
| IEM-G (1 conf) | 0.199 | 0.011 | 0.064 | 0.973 | 0.447 | 0.032 | 0.023 | 0.518 | 0.425 | 0.049 |
| ViT-G-R | 0.055 | 0.002 | 0.015 | 0.412 | 0.514 | 0.000 | 0.006 | 0.080 | 0.099 | 0.039 |
| VideoMol-G | 0.034 | 0.002 | 0.018 | 0.007 | 0.007 | 0.002 | 0.007 | 0.219 | 0.238 | 0.038 |
| ResNet18-V-R | 0.325 | 0.007 | 0.022 | 0.397 | 0.527 | 0.006 | 0.009 | 0.121 | 0.136 | 0.041 |
| IEM-V | 0.320 | 0.015 | 0.094 | 0.420 | 0.566 | 0.032 | 0.014 | 0.457 | 1.033 | 0.050 |

Table S41: The average RMSE performance on 10 reticular (R) datasets from StructNet. R#1, R#2, R#3, R#4, R#5, R#6, R#7, R#8, R#9, R#10 represent CHEMBL4888485_In, CHEMBL1614458_P, CHEMBL1614459_P, CHEMBL1613914_P, CHEMBL1614421_P, CHEMBL1614087_P, CHEMBL1614249_P, CHEMBL1614236_P, CHEMBL1614544_P, CHEMBL1614038_P, respectively. The green background represents top-5 performance.

| Models | R#1 | R#2 | R#3 | R#4 | R#5 | R#6 | R#7 | R#8 | R#9 | R#10 | Mean |
|---|---|---|---|---|---|---|---|---|---|---|---|
| mcfp4_2048 | 19.359 | 0.539 | 0.903 | 0.585 | 0.644 | 0.880 | 0.597 | 0.525 | 0.947 | 0.578 | 2.556 |
| ecfp4_2048 | 19.335 | 0.606 | 0.965 | 0.615 | 0.719 | 0.924 | 0.651 | 0.641 | **0.939** | 0.730 | 2.613 |
| maccs | 19.325 | 0.443 | 0.723 | 0.472 | 0.637 | 0.857 | 0.569 | **0.438** | 0.981 | 0.536 | 2.498 |
| physchem | 19.352 | 0.440 | 0.695 | 0.409 | 0.601 | **0.807** | 0.433 | 0.496 | 0.956 | 0.589 | 2.478 |
| atompair_2048 | 19.358 | 0.557 | 0.907 | 0.734 | 0.675 | 0.961 | 0.649 | 0.574 | 1.039 | 0.559 | 2.601 |
| rdkDes | 19.338 | 0.511 | 0.692 | **0.386** | 0.670 | 0.823 | 0.652 | 0.664 | 1.004 | 0.895 | 2.564 |
| BERT-6L | 19.700 | 0.438 | 0.664 | 0.396 | 0.590 | **0.807** | 0.388 | 0.451 | 0.999 | 0.429 | 2.486 |
| BERT-8L | 19.702 | 0.439 | 0.664 | 0.396 | 0.590 | **0.807** | **0.387** | 0.451 | 1.000 | 0.429 | 2.487 |
| RoBERTa-12L | 19.696 | 0.430 | 0.664 | 0.396 | 0.589 | **0.807** | **0.387** | 0.452 | 0.999 | 0.433 | 2.485 |
| molformer-R | 19.691 | 0.434 | 0.664 | 0.397 | 0.590 | **0.807** | 0.388 | 0.450 | 0.999 | 0.428 | 2.485 |
| Chem-BERT-6L | 19.665 | 0.438 | 0.664 | 0.395 | 0.591 | **0.807** | **0.387** | 0.451 | 1.001 | 0.429 | 2.483 |
| Chem-BERT-8L | 19.697 | 0.437 | 0.664 | 0.396 | 0.589 | **0.807** | **0.387** | 0.451 | 0.999 | 0.429 | 2.486 |
| CHEM-RoBERTa-12L | 19.701 | 0.425 | 0.664 | 0.397 | 0.590 | **0.807** | 0.392 | 0.452 | 0.997 | 0.432 | 2.486 |
| Molformer | 19.688 | 0.438 | 0.664 | 0.396 | 0.589 | **0.807** | 0.388 | 0.450 | 1.001 | 0.429 | 2.485 |
| GIN-R | 19.678 | 0.437 | 0.672 | 0.432 | 0.595 | 0.829 | 0.397 | 0.462 | 1.008 | 0.449 | 2.496 |
| EdgePred | 19.410 | 0.426 | 0.641 | 0.395 | 0.587 | 0.829 | 0.416 | 0.457 | 1.011 | 0.452 | 2.462 |
| ContextPred | 19.442 | 0.458 | 0.667 | 0.407 | 0.581 | 0.841 | 0.394 | 0.449 | 1.035 | **0.407** | 2.468 |
| infomax | 19.379 | 0.450 | 0.684 | 0.427 | 0.593 | 0.871 | 0.435 | 0.496 | 1.064 | 0.447 | 2.485 |
| masking | 19.478 | 0.450 | 0.712 | 0.393 | 0.572 | 0.855 | 0.411 | 0.468 | 1.058 | 0.452 | 2.485 |
| MolCLR | 19.542 | 0.437 | 0.656 | 0.416 | 0.591 | 0.822 | 0.397 | 0.465 | 1.014 | 0.425 | 2.477 |
| MoleBERT | 19.360 | 0.470 | 0.716 | 0.416 | 0.572 | 0.831 | 0.462 | 0.466 | 1.032 | 0.458 | 2.478 |
| CGIP-Graph | **19.240** | 0.493 | **0.633** | 0.410 | 0.585 | 0.841 | 0.445 | 0.479 | 1.051 | 0.443 | 2.462 |
| GraphMVP | 19.271 | 0.472 | 0.666 | 0.422 | 0.586 | 0.839 | 0.406 | 0.476 | 1.026 | 0.439 | 2.460 |
| SchNet | 19.639 | 0.459 | 0.719 | 0.432 | 0.575 | 0.874 | 0.400 | 0.478 | 0.987 | 0.425 | 2.499 |
| EGNN | 19.549 | 0.432 | 0.665 | 0.397 | 0.590 | **0.807** | **0.387** | 0.451 | 0.999 | 0.432 | 2.471 |
| TFN | 19.576 | 0.511 | 0.716 | 0.477 | 0.613 | 0.911 | 0.476 | 0.552 | 1.097 | 0.519 | 2.545 |
| SE3_Transformer | 19.644 | 0.457 | 0.691 | 0.536 | **0.547** | 0.866 | 0.480 | 0.535 | 1.069 | 0.491 | 2.532 |
| PaiNN | 19.317 | 1.385 | 1.393 | 0.978 | 1.160 | 1.377 | 0.962 | 1.326 | 1.594 | 0.888 | 3.038 |
| Uni-Mol-R (1 conf) | 19.565 | 0.431 | 0.664 | 0.396 | 0.590 | **0.807** | 0.388 | 0.450 | 1.001 | 0.428 | 2.472 |
| Uni-Mol (1 conf) | 19.529 | 0.426 | 0.665 | 0.398 | 0.591 | 0.808 | 0.395 | 0.451 | 1.001 | 0.428 | 2.469 |
| ResNet18-I-R | 19.346 | 0.429 | 0.672 | 0.397 | 0.585 | 0.817 | 0.394 | 0.456 | 1.010 | 0.430 | **2.454** |
| ImageMol | 19.618 | 0.447 | 0.751 | 0.423 | 0.605 | 0.835 | 0.431 | 0.487 | 1.052 | 0.459 | 2.511 |
| CGIP-Image | 19.459 | 0.425 | 0.679 | 0.408 | 0.596 | 0.818 | 0.393 | 0.471 | 0.999 | 0.437 | 2.468 |
| MaskMol | 19.660 | 0.430 | 0.664 | 0.396 | 0.588 | **0.807** | 0.388 | 0.450 | 1.000 | 0.426 | 2.481 |
| IEM-I | 19.474 | 0.440 | 0.688 | 0.401 | 0.591 | 0.816 | 0.394 | 0.455 | 1.005 | 0.432 | 2.470 |
| ResNet18-G-R | 19.500 | 0.430 | 0.663 | 0.403 | 0.589 | 0.824 | 0.393 | 0.458 | 0.987 | 0.426 | 2.467 |
| IEM-G (1 conf) | 19.796 | 0.433 | 0.676 | 0.413 | 0.590 | 0.843 | 0.397 | 0.467 | 1.010 | 0.445 | 2.507 |
| ViT-G-R | 19.654 | 0.436 | 0.664 | 0.397 | 0.585 | **0.807** | **0.387** | 0.450 | 0.998 | 0.422 | 2.480 |
| VideoMol-G | 19.667 | 0.434 | 0.665 | 0.397 | 0.589 | 0.809 | 0.390 | 0.451 | 0.999 | 0.428 | 2.483 |
| ResNet18-V-R | 19.529 | **0.424** | 0.676 | 0.395 | 0.585 | 0.822 | 0.390 | 0.457 | 0.984 | 0.430 | 2.469 |
| IEM-V | 19.726 | 0.429 | 0.672 | 0.414 | 0.581 | 0.839 | 0.401 | 0.471 | 1.017 | 0.464 | 2.501 |

Table S42: The standard deviation on 10 reticular (R) datasets from StructNet. R#1, R#2, R#3, R#4, R#5, R#6, R#7, R#8, R#9, R#10 represent CHEMBL4888485_In, CHEMBL1614458_P, CHEMBL1614459_P, CHEMBL1613914_P, CHEMBL1614421_P, CHEMBL1614087_P, CHEMBL1614249_P, CHEMBL1614236_P, CHEMBL1614544_P, CHEMBL1614038_P, respectively.

| Models | R#1 | R#2 | R#3 | R#4 | R#5 | R#6 | R#7 | R#8 | R#9 | R#10 |
|---|---|---|---|---|---|---|---|---|---|---|
| mcfp4_2048 | 0.002 | 0.002 | 0.003 | 0.004 | 0.005 | 0.002 | 0.003 | 0.003 | 0.002 | 0.002 |
| ecfp4_2048 | 0.002 | 0.004 | 0.005 | 0.005 | 0.005 | 0.002 | 0.003 | 0.003 | 0.004 | 0.002 |
| maccs | 0.034 | 0.004 | 0.006 | 0.003 | 0.003 | 0.005 | 0.005 | 0.002 | 0.004 | 0.003 |
| physchem | 0.008 | 0.005 | 0.001 | 0.002 | 0.005 | 0.001 | 0.017 | 0.015 | 0.001 | 0.037 |
| atompair_2048 | 0.006 | 0.003 | 0.004 | 0.007 | 0.002 | 0.003 | 0.003 | 0.003 | 0.003 | 0.002 |
| rdkDes | 0.004 | 0.004 | 0.001 | 0.000 | 0.011 | 0.005 | 0.022 | 0.020 | 0.009 | 0.031 |
| BERT-6L | 0.093 | 0.002 | 0.001 | 0.002 | 0.001 | 0.002 | 0.001 | 0.002 | 0.003 | 0.003 |
| BERT-8L | 0.065 | 0.004 | 0.002 | 0.003 | 0.002 | 0.002 | 0.001 | 0.001 | 0.005 | 0.003 |
| RoBERTa-12L | 0.030 | 0.015 | 0.001 | 0.004 | 0.003 | 0.002 | 0.003 | 0.005 | 0.010 | 0.007 |
| molformer-R | 0.040 | 0.006 | 0.001 | 0.004 | 0.003 | 0.004 | 0.006 | 0.003 | 0.010 | 0.006 |
| Chem-BERT-6L | 0.058 | 0.005 | 0.001 | 0.003 | 0.002 | 0.001 | 0.002 | 0.002 | 0.003 | 0.002 |
| Chem-BERT-8L | 0.046 | 0.007 | 0.003 | 0.001 | 0.002 | 0.002 | 0.002 | 0.001 | 0.005 | 0.004 |
| CHEM-RoBERTa-12L | 0.020 | 0.008 | 0.002 | 0.004 | 0.004 | 0.001 | 0.014 | 0.005 | 0.006 | 0.007 |
| Molformer | 0.060 | 0.010 | 0.001 | 0.003 | 0.001 | 0.003 | 0.003 | 0.003 | 0.007 | 0.004 |
| GIN-R | 0.330 | 0.031 | 0.044 | 0.017 | 0.026 | 0.020 | 0.010 | 0.026 | 0.045 | 0.052 |
| EdgePred | 0.160 | 0.024 | 0.032 | 0.012 | 0.014 | 0.016 | 0.028 | 0.017 | 0.021 | 0.024 |
| ContextPred | 0.063 | 0.035 | 0.075 | 0.016 | 0.022 | 0.009 | 0.007 | 0.012 | 0.014 | 0.013 |
| infomax | 0.202 | 0.022 | 0.090 | 0.016 | 0.029 | 0.017 | 0.021 | 0.023 | 0.037 | 0.013 |
| masking | 0.079 | 0.030 | 0.100 | 0.006 | 0.007 | 0.011 | 0.008 | 0.018 | 0.024 | 0.010 |
| MolCLR | 0.123 | 0.060 | 0.018 | 0.010 | 0.016 | 0.007 | 0.015 | 0.021 | 0.030 | 0.021 |
| MoleBERT | 0.056 | 0.039 | 0.038 | 0.025 | 0.013 | 0.026 | 0.026 | 0.014 | 0.030 | 0.023 |
| CGIP-Graph | 0.200 | 0.053 | 0.024 | 0.023 | 0.026 | 0.021 | 0.025 | 0.015 | 0.049 | 0.029 |
| GraphMVP | 0.239 | 0.075 | 0.033 | 0.019 | 0.037 | 0.027 | 0.022 | 0.014 | 0.037 | 0.026 |
| SchNet | 0.191 | 0.024 | 0.057 | 0.037 | 0.006 | 0.039 | 0.009 | 0.012 | 0.024 | 0.008 |
| EGNN | 0.404 | 0.011 | 0.001 | 0.003 | 0.002 | 0.001 | 0.002 | 0.003 | 0.006 | 0.003 |
| TFN | 0.148 | 0.086 | 0.051 | 0.024 | 0.026 | 0.031 | 0.048 | 0.038 | 0.069 | 0.042 |
| SE3_Transformer | 0.194 | 0.040 | 0.053 | 0.060 | 0.025 | 0.028 | 0.036 | 0.049 | 0.069 | 0.038 |
| PaiNN | 0.197 | 0.124 | 0.141 | 0.104 | 0.050 | 0.131 | 0.047 | 0.113 | 0.115 | 0.042 |
| Uni-Mol-R (1 conf) | 0.121 | 0.007 | 0.001 | 0.002 | 0.002 | 0.003 | 0.002 | 0.002 | 0.002 | 0.003 |
| Uni-Mol (1 conf) | 0.088 | 0.010 | 0.004 | 0.009 | 0.004 | 0.001 | 0.006 | 0.004 | 0.007 | 0.004 |
| ResNet18-I-R | 0.173 | 0.013 | 0.021 | 0.014 | 0.015 | 0.012 | 0.010 | 0.012 | 0.027 | 0.018 |
| ImageMol | 0.533 | 0.036 | 0.094 | 0.021 | 0.041 | 0.022 | 0.032 | 0.047 | 0.052 | 0.028 |
| CGIP-Image | 0.126 | 0.012 | 0.017 | 0.018 | 0.011 | 0.012 | 0.013 | 0.016 | 0.031 | 0.015 |
| MaskMol | 0.066 | 0.007 | 0.004 | 0.005 | 0.003 | 0.002 | 0.003 | 0.003 | 0.006 | 0.003 |
| IEM-I | 0.137 | 0.033 | 0.030 | 0.017 | 0.009 | 0.017 | 0.019 | 0.010 | 0.035 | 0.019 |
| ResNet18-G-R | 0.128 | 0.021 | 0.016 | 0.012 | 0.008 | 0.009 | 0.013 | 0.012 | 0.023 | 0.011 |
| IEM-G (1 conf) | 0.255 | 0.020 | 0.023 | 0.016 | 0.024 | 0.036 | 0.012 | 0.021 | 0.035 | 0.024 |
| ViT-G-R | 0.046 | 0.009 | 0.001 | 0.004 | 0.008 | 0.005 | 0.002 | 0.003 | 0.013 | 0.008 |
| VideoMol-G | 0.065 | 0.019 | 0.002 | 0.005 | 0.010 | 0.004 | 0.003 | 0.005 | 0.008 | 0.011 |
| ResNet18-V-R | 0.171 | 0.016 | 0.007 | 0.006 | 0.005 | 0.008 | 0.006 | 0.009 | 0.016 | 0.014 |
| IEM-V | 0.241 | 0.023 | 0.012 | 0.019 | 0.012 | 0.016 | 0.017 | 0.017 | 0.033 | 0.031 |

Table S43: Parameters and computational costs of different models with a batch size of 8. "time cost (30 epochs)" represents the total time required to train, evaluate, and test the model for 30 epochs on 10,000 molecules. The number of molecules for training, evaluation, and testing are 8,000, 1,000, and 1,000, respectively. "params" represents the number of parameters of the model.

| Modality | model | time cost (30 epochs) | params |
|---|---|---|---|
| SMILES | CHEM-BERT-no-pretrain | 46.391 min | 38.397M |
| | CHEM-BERT | 47.918 min | 38.397M |
| | CHEM-RoBERTa | 98.599 min | 85.495M |
| | CHEM-RoBERTa-no-pretrain | 86.776 min | 85.495M |
| | molformer-no-pretrain | 41.306 min | 25.511M |
| | CHEM-BERT-origin-no-pretrain | 96.216 min | 51.001M |
| | CHEM-BERT-origin | 51.751 min | 51.001M |
| | molformer | 39.692 min | 25.511M |
| Graph | GIN_RANDOM | 20.487 min | 1.862M |
| | EdgePred | 20.780 min | 1.862M |
| | ContextPred | 20.277 min | 1.862M |
| | infomax | 20.425 min | 1.862M |
| | masking | 13.720 min | 1.862M |
| | MolCLR | 20.795 min | 1.862M |
| | MoleBERT | 20.827 min | 1.862M |
| | CGIP-Graph | 25.292 min | 3.793M |
| | GraphMVP | 20.698 min | 1.862M |
| Geometry graph | Uni-Mol-R (1 conf) | 35.763 min | 47.600M |
| | Uni-Mol (1 conf) | 73.444 min | 47.600M |
| | TFN | 273.258 min | 8.663M |
| | SE3_Transformer | 257.265 min | 10.126M |
| Image | ResNet18-I-R | 26.215 min | 11.183M |
| | ImageMol | 26.479 min | 11.183M |
| | CGIP-Image | 27.817 min | 11.183M |
| | MaskMol | 54.007 min | 85.808M |
| | IEM-I | 25.387 min | 11.183M |
| Geometry Image | ResNet18-G-R | 89.537 min | 11.183M |
| | IEM-G (1 conf) | 84.701 min | 11.183M |
| Video | ResNet18-V-R | 750.935 min | 11.183M |
| | IEM-V | 934.883 min | 11.183M |

Table S44: The time required to train the model for 1 epoch using different numbers of frames. The trained model is ResNet18-V-R. "time cost (1 epoch)" is the total time taken to train on 8,000 molecules and evaluate on 2,000 molecules.

| #Frame | time cost (1 epoch) |
|---|---|
| 1 | 2.024 min |
| 3 | 2.786 min |
| 5 | 4.169 min |
| 10 | 5.934 min |
| 30 | 13.836 min |
| 60 | 25.849 min |

Table S45: Time required for the model to virtually screen 10,000 molecules.

| Modality | model | inference time (1,000 molecules) |
|---|---|---|
| SMILES | CHEM-BERT-no-pretrain | 2.301 s |
| | CHEM-BERT | 2.189 s |
| | CHEM-RoBERTa | 3.367 s |
| | CHEM-RoBERTa-no-pretrain | 3.315 s |
| | molformer-no-pretrain | 3.489 s |
| | CHEM-BERT-origin-no-pretrain | 2.652 s |
| | CHEM-BERT-origin | 2.341 s |
| | molformer | 3.441 s |
| Graph | GIN_RANDOM | 2.013 s |
| | EdgePred | 2.241 s |
| | ContextPred | 2.097 s |
| | infomax | 1.822 s |
| | masking | 2.254 s |
| | MolCLR | 1.633 s |
| | MoleBERT | 2.401 s |
| | CGIP-Graph | 1.884 s |
| | GraphMVP | 1.727 s |
| Geometry graph | Uni-Mol-R (1 conf) | 3.301 s |
| | Uni-Mol (1 conf) | 2.924 s |
| | TFN | 6.372 s |
| | SE3_Transformer | 10.180 s |
| Image | ResNet18-I-R | 4.151 s |
| | ImageMol | 2.106 s |
| | CGIP-Image | 2.111 s |
| | MaskMol | 2.600 s |
| | IEM-I | 2.147 s |
| Geometry Image | ResNet18-G-R | 5.769 s |
| | IEM-G (1 conf) | 5.618 s |
| Video | ResNet18-V-R | 73.419 s |
| | IEM-V | 73.163 s |

information to improve performance. It is worth noting that we find that **video have the largest differences with other modalities in RMSE, which may provide a direction for future multi-modal fusion on HIV dataset**.

Table S46: Differences between different modalities on 8 classification datasets from MoleculeNet. The gray background diagonal line is used as the boundary. The lower left corner and upper right corner respectively represent the calculation of the RMSE (the larger the difference, the greater the difference) and Pearson correlation coefficient (the smaller the difference, the greater the difference) between the two modalities. The bold ones represent the top 6 most different modality combinations.

| | Fingerprint | Sequence | Graph | Geometry Graph | Image | Geometry Image | Video |
|---|---|---|---|---|---|---|---|
| Fingerprint | - | 0.425 | 0.530 | 0.439 | **0.238** | **0.367** | 0.369 |
| Sequence | 0.178 | - | 0.388 | 0.524 | **0.308** | 0.443 | 0.426 |
| Graph | 0.125 | 0.190 | - | 0.438 | **0.284** | 0.390 | 0.381 |
| Geometry Graph | 0.143 | 0.192 | 0.151 | - | 0.393 | 0.515 | 0.480 |
| Image | 0.137 | **0.199** | 0.158 | 0.148 | - | **0.362** | **0.325** |
| Geometry Image | **0.193** | **0.235** | **0.195** | 0.152 | **0.191** | - | 0.514 |
| Video | 0.170 | **0.210** | 0.175 | 0.161 | 0.178 | 0.187 | - |

Given the differences in predicted logits between multi-modal molecules, we further conduct an exploratory experiment to find the upper limit of multi-modal fusion. We assume that the test set labels have been obtained. We generate the final prediction logits by determining the minimum difference between the prediction logits of different modalities and the true results. That is, for a certain molecule, the prediction logit of which modality is closest to its true result, we assign this prediction logit to this molecule. Finally, we obtained a ROC-AUC result of 99.7% on the HIV test set. This strong improvement means that **building a routing network or ranking algorithm for selecting results based on models of different molecular modalities is a promising direction**. We also show the contribution of different modalities in Figure S4, which suggests that incorporating video modality into multi-modal representation learning of molecules is promising in the future.

Table S47: The RMSE in prediction results between different modalities on HIV dataset. The larger the RMSE, the greater the difference between the two modes. Green indicates the top three with the greatest differences, and bold indicates the greatest difference.

|  | fingerprint | sequence | graph | geometry_graph | image | geometry_image | video |
|---|---|---|---|---|---|---|---|
| fingerprint | 0 | 0.093046 | 0.100329 | 0.067156 | 0.058782 | 0.102035 | **0.177339** |
| sequence | 0.093046 | 0 | 0.110071 | 0.089303 | 0.103196 | 0.102689 | **0.157157** |
| graph | 0.100329 | 0.110071 | 0 | 0.108321 | 0.117101 | 0.125993 | **0.176838** |
| geometry_graph | 0.067156 | 0.089303 | 0.108321 | 0 | 0.05556 | 0.091146 | **0.166107** |
| image | 0.058782 | 0.103196 | 0.117101 | 0.05556 | 0 | 0.098585 | **0.182885** |
| geometry_image | 0.102035 | 0.102689 | 0.125993 | 0.091146 | 0.098585 | 0 | **0.153324** |
| video | 0.177339 | 0.157157 | 0.176838 | 0.166107 | 0.182885 | 0.153324 | 0 |

Table S48: The Pearson correlation coefficient in prediction results between different modalities on HIV dataset. The smaller the Pearson correlation coefficient, the greater the difference between the two modes. Green indicates the top three with the greatest differences, and bold indicates the greatest difference.

|  | fingerprint | sequence | graph | geometry_graph | image | geometry_image | video |
|---|---|---|---|---|---|---|---|
| fingerprint | 1 | 0.58403 | 0.606879 | 0.519876 | 0.448639 | **0.387169** | 0.410779 |
| sequence | 0.58403 | 1 | 0.576967 | 0.650448 | **0.556317** | 0.572682 | 0.579381 |
| graph | 0.606879 | 0.576967 | 1 | 0.533092 | 0.450626 | **0.404815** | 0.441554 |
| geometry_graph | **0.519876** | 0.650448 | 0.533092 | 1 | 0.660593 | 0.619167 | 0.58209 |
| image | **0.448639** | 0.556317 | 0.450626 | 0.660593 | 1 | 0.564617 | 0.516407 |
| geometry_image | **0.387169** | 0.572682 | 0.404815 | 0.619167 | 0.564617 | 1 | 0.601794 |
| video | **0.410779** | 0.579381 | 0.441554 | 0.58209 | 0.516407 | 0.601794 | 1 |

## K.2 WHY DO UNI-MOL FEATURES ALLOW FOR THE HIGH PERFORMANCE IN MOLECULENET?

We speculate that the high performance of Uni-Mol benefits from its pre-training on 209 M molecular conformations. To verify this speculation, we pre-trained Uni-Mol using 50K, 200K, and 2M molecular conformations respectively and observed their performance on 8 classification datasets from MoleculeNet. In details, we use the official code and parameters provided by Uni-Mol [2], and pretrain with the first 50K, 200K and 2M conformations of pcqm4m-v2-train [3]. The experimental settings for linear probing on MoleculeNet are consistent with our paper. As shown in Table S49, we find that the amount of pre-training data is very important for Uni-Mol to achieve good performance on MoleculeNet.

Table S49: ROC-AUC performance of Uni-Mol pre-trained with different data sizes on 8 classification tasks from MoleculeNet with linear probing and 10 different run seeds. In linear probing, 10 conformations per molecule are used. #Conf indicates the amount of data used for pre-training Uni-Mol.

|  | #Conf | BBBP | Tox21 | ToxCast | Sider | ClinTox | MUV | HIV | BACE | Avg |
|---|---|---|---|---|---|---|---|---|---|---|
|  | 50K | 68.0±0.2 | 71.3±0.1 | 63.1±0.4 | 60.4±0.2 | 68.0±3.2 | 71.3±0.5 | 78.0±0.4 | 80.7±0.7 | 70.1 |
|  | 200K | 65.0±0.9 | 73.9±0.2 | 64.0±0.8 | 62.6±0.3 | 62.6±1.5 | 72.4±1.0 | 76.2±1.0 | 79.2±0.7 | 69.5 |
| Uni-Mol | 2M | 67.3±0.2 | 73.2±0.2 | 63.7±0.7 | 60.6±0.3 | 64.8±1.3 | 73.9±0.7 | 77.1±0.3 | 77.9±0.3 | 69.8 |
|  | 209M | 69.3±0.6 | 75.2±0.1 | 65.8±0.5 | 61.6±0.4 | 85.1±4.4 | 80.3±0.7 | 77.5±0.1 | 78.2±0.2 | 74.1 |

## K.3 WHY MULTIPLE CONFORMATIONS CAN SIGNIFICANTLY IMPROVE GEOMETRY IMAGE PERFORMANCE?

Tables S20 and S21 show that multiple conformations can significantly improve the performance of the geometric image modality (such as IEM-G) with little gain for geometric graph modality (such as Uni-Mol). Here, we try to answer why this phenomenon occurs. We select BBBP, Tox21, ClinTox and ToxCast and analyze them in terms of the direction and scale of the features from different

---

[2]https://github.com/deepmodeling/Uni-Mol/blob/main/unimol/README.md

[3]https://ogb.stanford.edu/docs/lsc/pcqm4mv2/

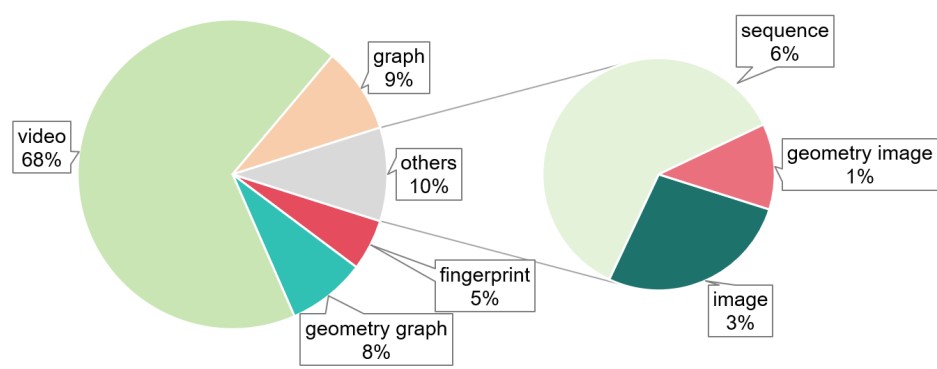

Figure S4: Contribution of different modalities to prediction on the HIV test set.

conformations. We use IEM-G and Uni-Mol to extract features of 10 conformations from these 4 datasets.

- **Direction between features from different conformations.** Cosine similarity represents the similarity between features in direction. As shown in Figure S5, the The lower left corner and upper left corner show the difference between Uni-Mol and IEM-G in cosine similarity, respectively. Obviously, we find that the cosine similarity of IEM-G is lower than that of Uni-Mol, which means that IEM-G has more diversity of direction in extracting features of different conformations. The cosine similarities of Uni-Mol are all above 99%, which means that multiple conformations are difficult to provide additional information.

- **Scale between features from different conformations.** Euclidean distance represents the difference in feature scale. As shown in Figure S6, the lower left corner and upper left corner show the differences between Uni-Mol and IEM-G in Euclidean distance, respectively. It is easy to observe that IEM-G has a larger Euclidean distance than Uni-Mol, which means that IEM-G has a larger difference in feature scale than Uni-Mol. It may bring more degrees of freedom to IEM-G in feature space.

Furthermore, we plot the distribution of cosine similarity and Euclidean distance in Figure S7 . We can see that the distribution of IEM-G is much flatter than that of Uni-Mol. In general, Figures S5, S6, S7 show that **the multi-conformational features extracted by IEM-G have more directional diversity and scale diversity**.

### K.4 IMPORTANCE OF NUMBER OF LAYERS FOR SEQUENCE MODELS ON MOLECULENET

In the MoleculeNet experiment, we see that even without any pre-training, the features of sequence-based MolFormer-R can still achieve excellent performance on MoleculeNet. Therefore, in order to find out whether there is a correlation between the number of layers of MolFormer-R and its performance, we perform ablation on the number of layers of the MolFormer-R. In MolFormer-R, the default number of layers is 6. Here, we further set the number of layers to 1, 2, 4, and 8. As shown in Table S50, we find that with the increase of the number of layers, except for the average performance of the 6th layer, the average performance of other layers increased from 69.91% of the 1st layer to 71.55% of the 8-layer, which suggests that further improving the performance by increasing the number of layers of the sequence model is a possible direction to try.

### K.5 WHY DOES THE SOME MODALITIES HAVE STRONG PERFORMANCE EVEN WITHOUT TRAINING?

In the linear probing experiment of MoleculeNet, we find that some modalities (sequence and geometry) can still achieve good performance even without any training, especially the MolFormer-R based on the sequence modality, which achieves an average ROC-AUC of 70.40% on 8 classification tasks. We speculate that this is related to the inductive bias of the modality. Given the excellent

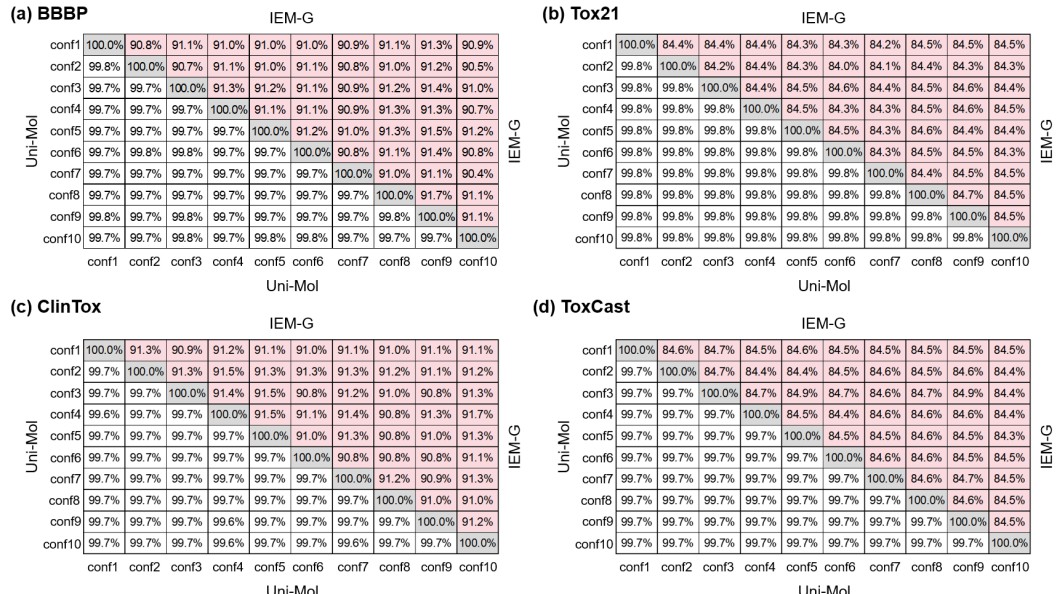

Figure S5: Cosine similarity between different conformations on BBBP, Tox21, ClinTox, and Tox-Cast. The white background in the lower left corner represents the features of different conformations extracted using Uni-Mol and the cosine similarity calculated, while the pink background in the upper right corner represents those of IEM-G.

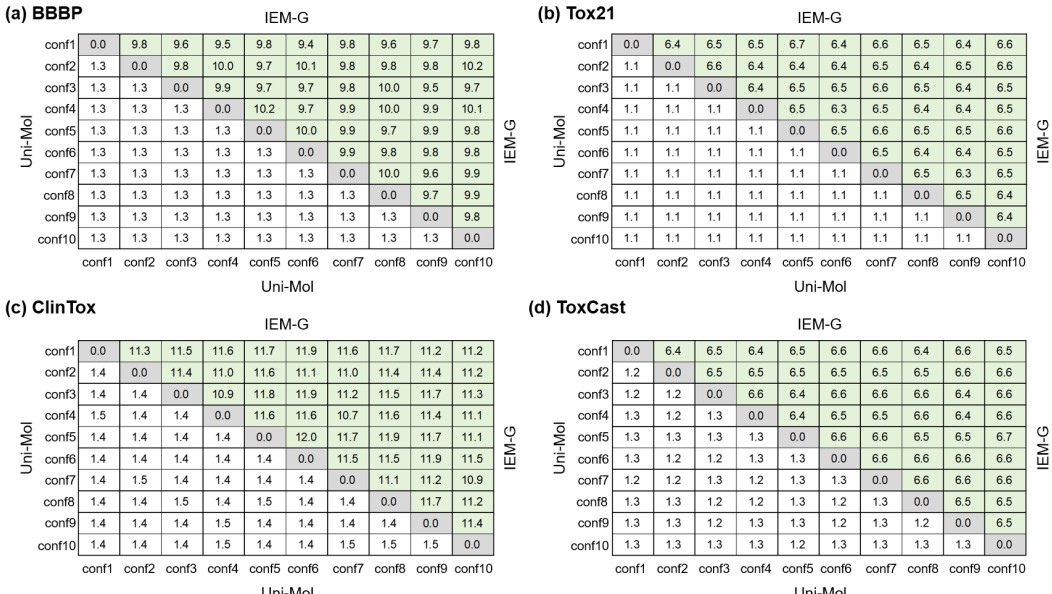

Figure S6: Euclidean distance between different conformations on BBBP, Tox21, ClinTox, and ToxCast. The white background in the lower left corner represents the features of different conformations extracted using Uni-Mol and the cosine similarity calculated, while the green background in the upper right corner represents those of IEM-G. "conf" 1 to "conf 10" refer to 10 different molecular conformations.

performance of the sequence modality without training, we speculate that the model's ability to recognize molecular substructures is the key to its performance.

To verify this conjecture, we use 6 fingerprints (mcfp4_2048, atompair_2048, ecfp4_2048, maccs, physchem, rdkDes) and randomly initialized models from 6 different modalities (sequence, graph

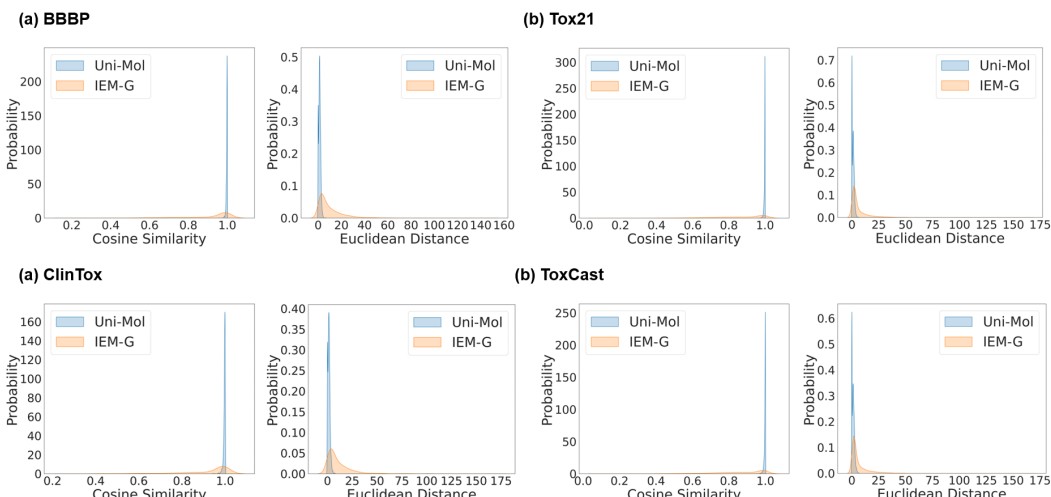

Figure S7: The distribution of cosine similarity and Euclidean distance between different conformations on BBBP, Tox21, ClinTox, and ToxCast datasets with Uni-Mol and IEM-G. The distribution here is calculated by randomly sampling 2 from 10 conformations.

Table S50: Ablation study of the number of MolFormer-R layers using ROC-AUC metric and 10 runs. #layers indicates the number of layers in the setting.

| | #layers | BBBP | Tox21 | ToxCast | Sider | ClinTox | MUV | HIV | BACE | Avg |
|---|---|---|---|---|---|---|---|---|---|---|
| MolFormer-R | 1 | 68.9±0.4 | 71.0±0.1 | 61.6±0.4 | 56.9±0.3 | 82.6±0.8 | 68.8±1.7 | 70.2±0.7 | 79.3±0.9 | 69.91 |
| | 2 | 71.5±0.9 | 71.1±0.1 | 61.8±0.4 | 57.2±0.5 | 81.4±0.6 | 68.1±1.5 | 70.8±0.7 | 78.5±1.3 | 70.05 |
| | 4 | 69.4±0.8 | 70.9±0.2 | 62.3±0.2 | 57.2±0.2 | 82.4±0.2 | 67.1±1.9 | 72.0±0.6 | 89.7±0.7 | 71.38 |
| | 6 | 74.6±0.5 | 71.6±0.3 | 61.5±0.3 | 55.9±0.3 | 86.2±0.3 | 67.2±1.6 | 71.2±0.5 | 75.0±1.6 | 70.40 |
| | 8 | 75.1±0.6 | 71.2±0.2 | 61.7±0.3 | 55.4±0.4 | 79.8±1.6 | 67.8±1.9 | 70.4±0.8 | 91.0±2.0 | **71.55** |

geometry graph, image, geometry image, video) for feature extraction on 8 classification datasets from MoleculeNet. In particular, the models from sequence, graph, geometry graph, image, geometry image and video are MolFormer-R, GIN-R, Uni-Mol-R, ResNet18-R, ResNet-G-R, VideoMol-R, respectively. Subsequently, we extract the scaffolds of the molecules and select the top 10 scaffolds with the highest counts as the substructures of the molecules. Finally, we sample 1,000 molecules from each of the selected substructures and use t-SNE (Van der Maaten & Hinton, 2008) for dimensionality reduction and clustering visualization. Note that all molecules will be sampled if the number of molecules containing a certain substructure is less than 1,000. To quantitatively analyze the model's ability to identify substructures, we calculate the Davis-Bouldin (DB) index (Davies & Bouldin, 1979) between the features of dimensionality reduction and the substructure labels, which is used to evaluate the clustering performance. DB index is a quantitative metric used to evaluate clustering quality. The lower the value, the better the clustering quality.

Table S51 shows the DB Index for different methods. We calculate the Pearson correlation coefficient using the results of DB index in Table S51 and the results of ROC-AUC in Table S20 . Taking BBBP as an example, we concatenate the results of BBBP on 12 methods in Table S51 into one vector as the vector of DB index. Then, we find the corresponding 12 methods in Table S20 and concatenate them into a ROC-AUC vector. Figure S8 shows the Pearson correlation coefficient calculated using the DB index vector and the ROC-AUC vector. Obviously, we can draw a conclusion: DB index is inversely proportional to ROC-AUC. This means that **the inductive bias of identifying substructures is important for predicting the properties of molecules**.

Figure S9 shows t-SNE visualizations of different methods from 7 modalities on the HIV dataset. We find that fingerprint-based mcfp4_2024, sequence-based MolFormer, graph-based GIN-R, and Uni-Mol-R have a good inductive bias for identifying substructures, while the three vision-based modalities (image-based ResNet18-R, geometric image-based ResNet-G-R, and video-based VideoMol-R) need to rely on post-training. In particular, we find that MolFormer-R is consistent with mcfp4_2048 in the clustering distribution of some substructures, such as the red and cyan clusters, which may

Table S51: The DB index of different models in 10 molecular substructures.

| | BBBP | Sider | Tox21 | ToxCast | HIV | MUV | ClinTox | BACE |
|---|---|---|---|---|---|---|---|---|
| mcfp4_2048 | 1.4224 | 1.5532 | 6.0667 | 5.0185 | 3.1764 | 10.3910 | 7.2231 | 0.4158 |
| atompair_2048 | 1.8361 | 2.6594 | 2.9751 | 2.8674 | 4.8532 | 2.8937 | 3.0130 | 0.3366 |
| ecfp4_2048 | 5.2635 | 1.1886 | 5.9853 | 3.3846 | 1.5788 | 1.9715 | 2.2368 | 0.3092 |
| maccs | 1.4097 | 2.9680 | 22.9463 | 7.9525 | 3.8023 | 3.4787 | 3.4251 | 0.5501 |
| physchem | 6.2917 | 4.3235 | 9.3082 | 10.4890 | 14.9467 | 20.1449 | 17.7473 | 7.9298 |
| rdkDes | 7.9706 | 20.7415 | 6.2466 | 7.1217 | 15.3582 | 24.6885 | 5.0376 | 4.8888 |
| MolFormer-R | 2.0754 | 1.8347 | 3.5817 | 4.1549 | 7.7935 | 17.2643 | 2.3998 | 1.1253 |
| GIN-R | 1.2580 | 1.5947 | 7.4661 | 4.2741 | 6.4118 | 5.4392 | 3.5611 | 1.2960 |
| Uni-Mol-R | 1.5010 | 2.7910 | 4.4361 | 5.2003 | 6.0031 | 3.7354 | 3.9110 | 1.1853 |
| ResNet18-R | 14.1199 | 13.5737 | 34.2620 | 25.3775 | 28.6812 | 36.1045 | 16.4683 | 16.3196 |
| ResNet-G-R | 8.9312 | 14.8282 | 12.5358 | 39.0406 | 13.5395 | 16.4114 | 9.5849 | 8.7852 |
| VideoMol-R | 12.6038 | 6.4966 | 11.0884 | 10.6541 | 10.0499 | 6.7280 | 11.6074 | 12.0046 |

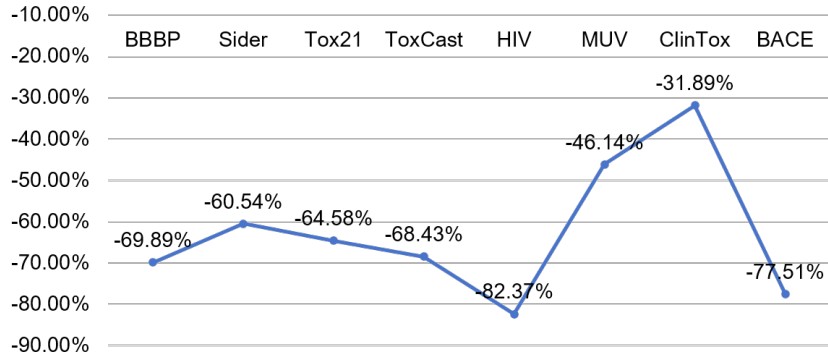

Figure S8: The Pearson correlation coefficient between the DB index vector and the ROC-AUC vector on 8 classification datasets from MoleculeNet.

provide evidence for the similar performance of MolFormer-R and mcfp4_2048 on 8 classification tasks from MoleculeNet, with average ROC-AUC performance of 70.40% and 71.66%, respectively.

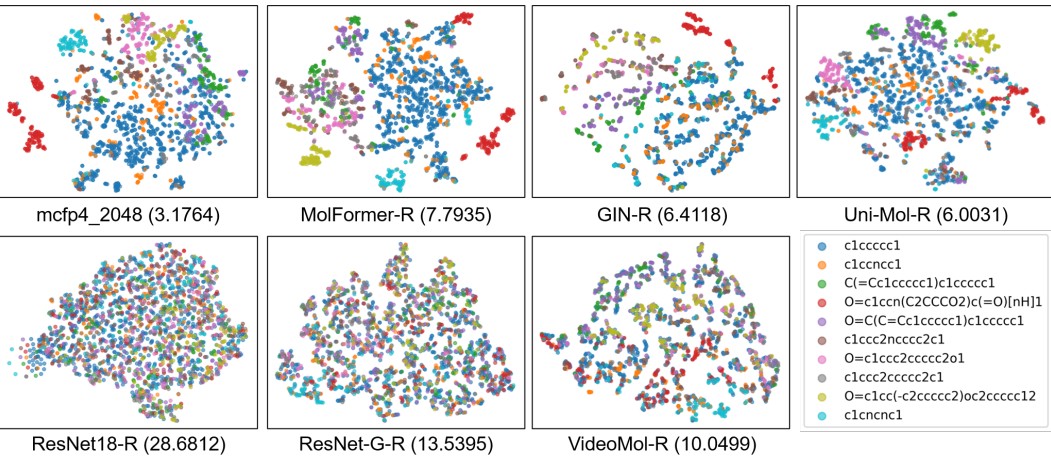

Figure S9: The t-SNE visualization of methods from 7 modalities on HIV dataset with 10 substructures. The value in brackets is the DB index.

### K.6 WHY ARE GEOMETRIC IMAGES AND VIDEOS NOT SENSITIVE TO RGB AND BGR FORMATS?

For images, RGB and BGR are two similar formats. Early OpenCV used BGR format by default and current PIL uses RGB format by default. Here we study the impact of RGB and BGR on the performance of geometry images and videos. Table S22 and Table S23 show the results for RGB. We find no significant difference between RGB and BGR for geometry images and videos. For example, VideoMol uses BGR (69.03%) better than RGB (68.53%) on classification tasks and RGB (1.190) is better than BGR (1.222) on regression tasks. The relationship between BGR and RGB is similar to image augmentation and the combination between them may further improve performance, just like TTA (Test Time Augmentation) (Kimura, 2021).

A natural hypothesis about why geometric images and videos are insensitive to RGB and BGR formats is that the features extracted from RGB and BGR images have high similarity. To verify this hypothesis, we evaluate the cosine similarity between RGB features and BGR features extracted by IEM-G on 12 molecular property prediction datasets. As shown in Figure S10, we find that IEM-G has high similarity between RGB and BGR features ranging from 78.4% to 98.5%, which verifies the rationality of our hypothesis. Furthermore, we also find that RGB images and BGR images still have certain differences in features. Inspired by Appendix K.3, it is a **promising direction to use different formats of images to increase the diversity of molecules and further improve the performance of the model by fusing images of different formats**.

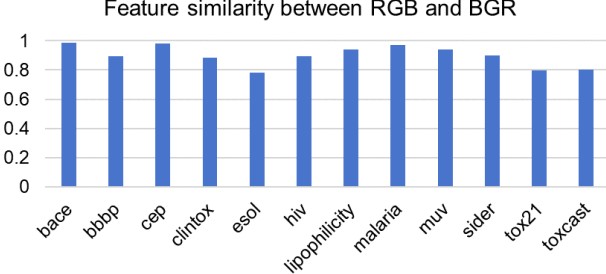

Figure S10: The cosine similarity between RGB and BGR features extracted by IEM-G on 12 molecular property prediction datasets.

### K.7 WHY IS VIDEO MODALITY SO GOOD AT TASKS RELATED TO ATOMS AND FUNDAMENTAL PROPERTIES?

For fairness, we select the unpretrained method with best performance in each modality as the representative of the performance of that modality, namely BERT-6L (sequence modality), GIN-R (graph modality), TFN (geometry modality), ResNet18-I-R (image modality), ResNet18-G-R (geometry image modality), and ResNet18-V-R (video modality). We take the atomic distribution prediction task in MBANet as an example for analysis.

In order to verify the robustness of the conclusion, we use coefficient of determination ($R^2$) (Di Buc-chianico, 2008) and Kullback-Leibler Divergence (KLD) (Kullback & Leibler, 1951) to evaluate these methods on 6 common atomic distribution prediction tasks, including C, N, O, F, S, Cl. $R^2$ and KLD are used to evaluate the explanatory power of the predicted results for the true results (goodness of fit) and the difference between the predicted distribution and the true distribution, re-spectively. As shown in Table S52 and Table S53, we can further conclude that molecular video representation has advantages in atomic distribution prediction due to its superior performance on $R^2$ and KLD. Figure S11 also shows that the predicted results of the video modality have very similar probability distributions to the true results.

Next, we analyze the principle behind video to achieve better results. Since sequence-, graph-, and geometry-based methods have a similar learning paradigm, namely, message passing between tokens/nodes through attention, we choose the graph modality-based GIN-R method as their repre-sentative.

Table S52: The $R^2$ performance (The higher the better) between predicted results and true results on 6 common atomic distribution prediction tasks from MBANet. The sequence, graph, geometry graph, image, geometry image, and video represent BERT-6L, GIN-R, TFN, ResNet18-I-R, ResNet18-G-R, and ResNet18-V-R, respectively.

|  | C | N | O | F | S | Cl |
|---|---|---|---|---|---|---|
| sequence | -0.07169458 | -0.319058707 | -0.998515564 | -0.605519986 | -0.064993676 | -0.012769986 |
| graph | -0.126330018 | 0.638550341 | 0.864461839 | 0.609151781 | -0.945517063 | 0.24113214 |
| geometry graph | 0.763227582 | 0.819823027 | 0.887618303 | **0.900081873** | 0.818889618 | **0.876054287** |
| image | 0.202817709 | 0.416600559 | 0.498178795 | 0.55820686 | 0.454666281 | 0.531525685 |
| geometry image | 0.406215768 | 0.801209417 | 0.802587877 | 0.729413738 | 0.407258332 | 0.499188822 |
| video | **0.914670507** | **0.939035378** | **0.956766033** | 0.806362244 | **0.906643885** | 0.705495937 |

Table S53: The KLD performance (the smaller the better) between predicted results and true results on 6 common atomic distribution prediction tasks from MBANet. The sequence, graph, geometry graph, image, geometry image, and video represent BERT-6L, GIN-R, TFN, ResNet18-I-R, ResNet18-G-R, and ResNet18-V-R, respectively.

|  | C | N | O | F | S | Cl |
|---|---|---|---|---|---|---|
| sequence | 11.36941061 | 10.57263088 | 5.290512691 | 5.109889215 | 4.78485022 | 5.506043385 |
| graph | 1.649138246 | 0.802423623 | 0.361164172 | 0.545743367 | 0.929395787 | 0.601528091 |
| geometry graph | 0.184020462 | 0.976098698 | 1.227725127 | 0.430602355 | 1.102794542 | 0.850205376 |
| image | 0.220310553 | 0.613314328 | 1.830226786 | 0.334507478 | 0.1362694 | 0.459256267 |
| geometry image | 0.187400906 | 0.604782544 | 0.591080359 | 0.343508344 | **0.132136985** | 0.585412658 |
| video | **0.010882909** | **0.245992622** | **0.377055353** | **0.169471655** | 0.186108709 | **0.225095687** |

We first analyze why GIN-R performs poorly on the atom prediction task, which may be a chain reaction affecting the estimation of molecular weight in MBANet$_{attr}$. We conjecture that graph message passing is not conducive to the model learning the semantics of a single node because the representation of a node are determined by the representation of its neighbors. As shown in Table S54, there is no relationship between the same atoms, which means that the information represented by atoms is more affected by their neighbors than by themselves. For example, C#4 is at most 86% similar to other carbon atoms (such as C#6), but is at most 95% similar to other types of atoms (such as N#2). Therefore, the inductive bias towards capturing structural information causes the graph modality to lose the ability to focus on the nodes themselves.

Table S54: Cosine similarity of pairwise atomic representations using GIN-R on the molecule 'N#Cc1cccc2nnc(C(F)(F)F)n12'. # indicates the atom number.

|  | C#1 | C#2 | C#3 | C#4 | C#5 | C#6 | C#7 | C#8 | F#1 | F#2 | F#3 | N#1 | N#2 | N#3 | N#4 |
|---|---|---|---|---|---|---|---|---|---|---|---|---|---|---|---|
| C#1 | 1.00 | 1.00 | 0.96 | 0.45 | 0.81 | 0.66 | -0.04 | -0.07 | -0.18 | -0.18 | -0.18 | 0.22 | 0.34 | 0.13 | 0.61 |
| C#2 | 1.00 | 1.00 | 0.98 | 0.51 | 0.84 | 0.71 | 0.00 | -0.05 | -0.18 | -0.18 | -0.18 | 0.28 | 0.40 | 0.19 | 0.66 |
| C#3 | 0.96 | 0.98 | 1.00 | 0.65 | 0.92 | 0.83 | 0.12 | -0.01 | -0.16 | -0.16 | -0.16 | 0.41 | 0.57 | 0.36 | 0.78 |
| C#4 | 0.45 | 0.51 | 0.65 | 1.00 | 0.82 | 0.86 | 0.51 | 0.15 | 0.05 | 0.05 | 0.05 | 0.89 | 0.95 | 0.84 | 0.87 |
| C#5 | 0.81 | 0.84 | 0.92 | 0.82 | 1.00 | 0.98 | 0.32 | 0.08 | -0.16 | -0.16 | -0.16 | 0.51 | 0.80 | 0.59 | 0.94 |
| C#6 | 0.66 | 0.71 | 0.83 | 0.86 | 0.98 | 1.00 | 0.42 | 0.12 | -0.13 | -0.13 | -0.13 | 0.55 | 0.89 | 0.70 | 0.98 |
| C#7 | -0.04 | 0.00 | 0.12 | 0.51 | 0.32 | 0.42 | 1.00 | 0.81 | 0.69 | 0.69 | 0.69 | 0.47 | 0.59 | 0.80 | 0.56 |
| C#8 | -0.07 | -0.05 | -0.01 | 0.15 | 0.08 | 0.12 | 0.81 | 1.00 | 0.91 | 0.91 | 0.91 | 0.13 | 0.20 | 0.35 | 0.21 |
| F#1 | -0.18 | -0.18 | -0.16 | 0.05 | -0.16 | -0.13 | 0.69 | 0.91 | 1.00 | 1.00 | 1.00 | 0.22 | 0.03 | 0.25 | -0.03 |
| F#2 | -0.18 | -0.18 | -0.16 | 0.05 | -0.16 | -0.13 | 0.69 | 0.91 | 1.00 | 1.00 | 1.00 | 0.22 | 0.03 | 0.25 | -0.03 |
| F#3 | -0.18 | -0.18 | -0.16 | 0.05 | -0.16 | -0.13 | 0.69 | 0.91 | 1.00 | 1.00 | 1.00 | 0.22 | 0.03 | 0.25 | -0.03 |
| N#1 | 0.22 | 0.28 | 0.41 | 0.89 | 0.51 | 0.55 | 0.47 | 0.13 | 0.22 | 0.22 | 0.22 | 1.00 | 0.78 | 0.77 | 0.59 |
| N#2 | 0.34 | 0.40 | 0.57 | 0.95 | 0.80 | 0.89 | 0.59 | 0.20 | 0.03 | 0.03 | 0.03 | 0.78 | 1.00 | 0.91 | 0.92 |
| N#3 | 0.13 | 0.19 | 0.36 | 0.84 | 0.59 | 0.70 | 0.80 | 0.35 | 0.25 | 0.25 | 0.25 | 0.77 | 0.91 | 1.00 | 0.81 |
| N#4 | 0.61 | 0.66 | 0.78 | 0.87 | 0.94 | 0.98 | 0.56 | 0.21 | -0.03 | -0.03 | -0.03 | 0.59 | 0.92 | 0.81 | 1.00 |

Different from graphs, the inductive bias of molecular videos focuses on learning local patterns. Figure S12 shows the GradCAM attention of ResNet18-V-R on MBANet$_{atom}$. We find that video-based ResNet18-V-R can obtain accurate atomic distribution information based on the local information. In particular, ResNet18-V-R accurately locates the positions of atoms with attention area in the video frame. Based on the located atoms, we can count the correct atomic distribution.

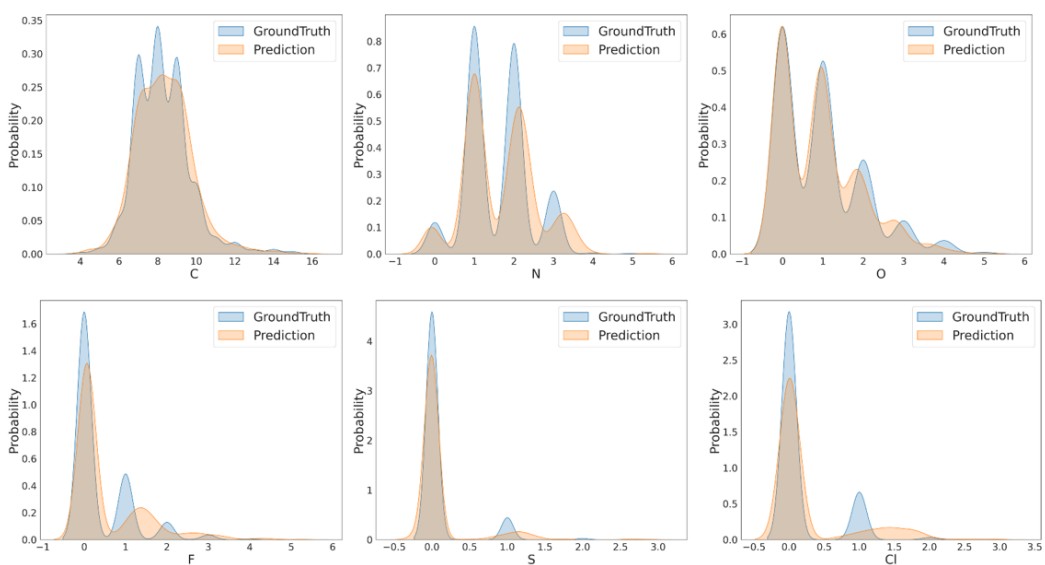

Figure S11: The probability distribution of 6 atoms (C, N, O, F, S, Cl) with video modality-based IEM-V. GroundTruth and Prediction represent the true results and the predicted results of ResNet-V-R. The title of $x$-axis represents the name of atom.

In general, **the advantage of molecular videos is that they can learn local information of molecules with high degrees of freedom, while graphs are limited by the message passing of neighbors, which weakens the extraction of local atomic-level information.**

### K.8    SIGNIFICANCE TEST FOR MODAL PREFERENCE IN STRUCTNET

In Table 6, we obtain the preferences of different modalities for molecular types. Since the means in the table are obtained from 100 results (10 runs on 10 data sets), the conclusions have good validity. Our conclusions on simplifying preferences are as follows:

- **Preference#1.** The geometry graph modality prefers acyclic (AC and A) molecules.
- **Preference#2.** The fingerprint and graph modalities prefer cyclic (CC and M) molecules.
- **Preference#3.** The visual-based modalities (image, geometry image, and video) prefer macro-cyclic peptide (MP) and reticular (R) molecules.

In order to further test the robustness of the conclusions, we conduct significance tests on comparisons between different modalities. Specifically, we use a two-sided Mann-Whitney U test (Mann & Whitney, 1947) to evaluate whether the results between modalities are significantly different.

Table S55, Table S56 and Table S57 show the results of the significance test of Preference#1, Preference#2 and Preference#3 respectively. In Preference#1, we find that geometric graph prefer AC with significant differences compared with geometric images and video modalities. In Preference#2, We find that fingerprint and graph preferences are significantly different in M compared to all other modalities. In Preference#3, modalities of image, geometry image and video prefer MP and R showing significant differences in most cases.

Overall, the average results reported in Table 6 are statistically significant and in most cases the results are significantly different.

Table S55: Results of two-sided Mann-Whitney U test between the graph modality and other modalities (fingerprint, sequence, graph, image, geometry image, video) with a significance level $p < 0.05$ on acyclic chain molecules (AC) and acyclic molecules (A) of StructNet. Green background indicates significant differences in results.

| Modality | Molecular type | Fingerprint | Sequence | Graph | Image | Geometry Image | Video |
|----------|---------------|-------------|----------|-------|-------|----------------|-------|
| Geometry Graph | AC | 0.709404 | 0.072866 | 0.314039 | 0.176196 | 0.036353 | 0.000247 |
| | A | 0.086259 | 0.464252 | 0.089914 | 0.760945 | 0.906625 | 0.720347 |

Table S56: Results of two-sided Mann-Whitney U test between the fingerprint, graph modalities and other modalities (sequence, geometry graph, image, geometry image, video) with a significance level $p < 0.05$ on acyclic cyclic chain molecules (CC) and macro molecules (M) of StructNet. Green background indicates significant differences in results.

| Molecular type | Modality | Sequence | Geometry Graph | Image | Geometry Image | Video |
|----------------|----------|----------|----------------|-------|----------------|-------|
| CC | Fingerprint | 0.020995 | 0.036789 | 0.009321 | 0.460525 | 0.444346 |
| | Graph | 0.269363 | 0.030572 | 0.757225 | 0.242288 | 0.283926 |
| M | Fingerprint | 0 | 0 | 0 | 0 | 0 |
| | Graph | 0 | 0.00003 | 0 | 0 | 0 |

Table S57: Results of two-sided Mann-Whitney U test between the image, geometry image, video modalities and other modalities (fingerprint, sequence, graph, geometry graph) with a significance level $p < 0.05$ on macrocyclic peptide molecules (MP) and acyclic reticular molecules (R) of StructNet. Green background indicates significant differences in results.

| Molecular type | Modality | Fingerprint | Sequence | Graph | Geometry Graph |
|----------------|----------|-------------|----------|-------|----------------|
| MP | Image | 0.000044 | 0.322914 | 0 | 0.000122 |
| | Geometry Image | 0.001465 | 0.255839 | 0 | 0.00024 |
| | Video | 0.000897 | 0.202102 | 0 | 0.000185 |
| R | Image | 0.000001 | 0.724008 | 0.005462 | 0.295615 |
| | Geometry Image | 0.000187 | 0.493057 | 0.007428 | 0.17973 |
| | Video | 0.000007 | 0.672471 | 0.001746 | 0.342439 |

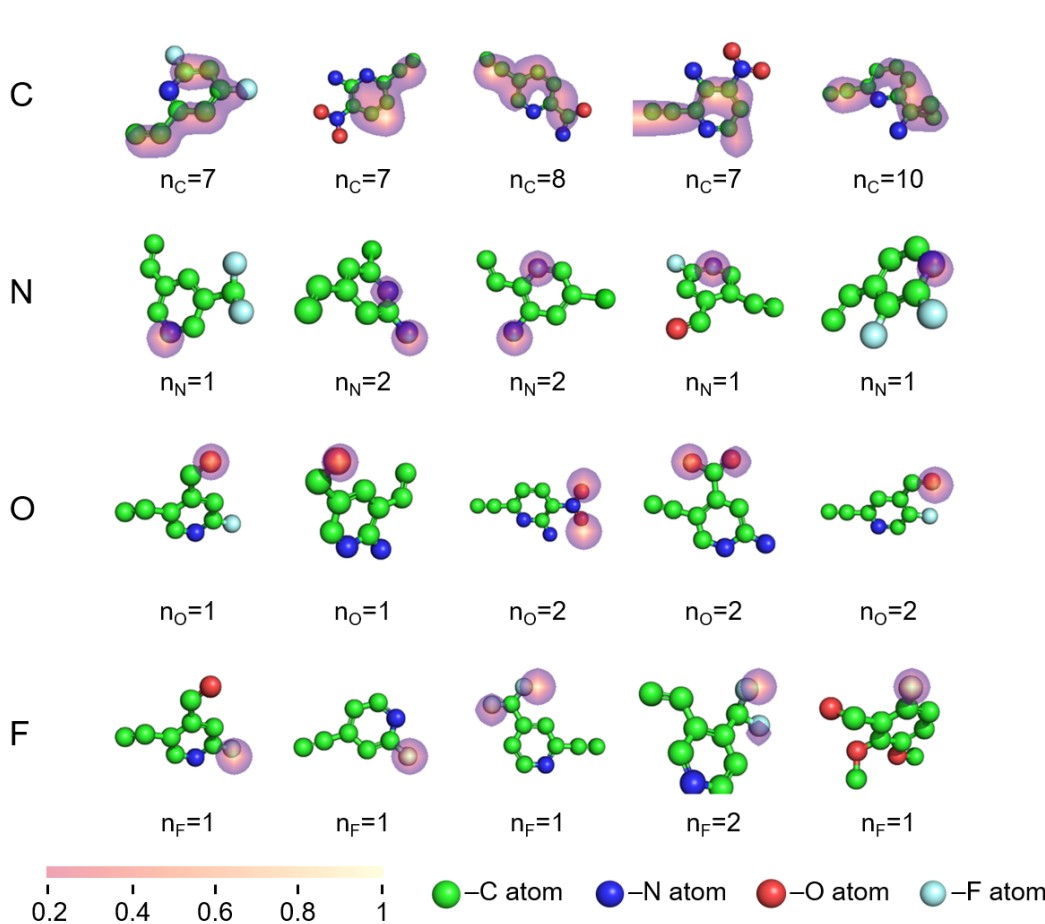

Figure S12: Grad-CAM visualization of ResNet18-V-R on frames of molecular videos. The first to fourth rows show the corresponding GradCAM visualizations of molecules when ResNet18-V-R predicts the distribution of C, N, O, and F atoms, respectively. We use 0.2 as the threshold for visualization, that is, set the importance lower than 0.2 to 0. $n_C$, $n_N$, $n_O$, $n_F$ represent the number of C, N, O, F atoms, respectively.

