# OpenReview forum: "BenchMol: A Multi-Modality Benchmarking Platform for Molecular Representation Learning"
_ICLR.cc/2025/Conference — ICLR 2025 Conference Withdrawn Submission_

### Official Review · Reviewer_LcfM · 2024-10-31

**Soundness:** 2
**Presentation:** 2
**Contribution:** 1
**Rating:** 5
**Confidence:** 5

**Summary:**

This paper proposes a multi-modality molecular property benchmark for molecular representation learning (MRL) methods. It generates various modalities, including fingerprint, sequence, graph, geometry, image, geometry-based image, and video, and constructs new benchmarks using data from PCQM4Mv2 and CHEMBL 34. A range of single-modality methods are evaluated on both MoleculeNet and the newly constructed benchmarks.

**Strengths:**

1. The paper is well-written and easy to understand.
2. The multi-modality methods are extensive, covering a broad range of modalities.
3. The conclusions from the linear probing experiments on different modality methods in MoleculeNet (Section 5.2) are insightful and interesting.

**Weaknesses:**

1. The labels in MBANet, consisting only of atom types, bond types, and basic molecular attributes, are relatively simple and lack practical value for a comprehensive molecular property benchmark.
2. I am curious about the rationale for splitting the data into different types (such as acyclic, complete chain, acyclic chain, macrocyclic peptide, macromolecule, and reticular) based on their 2D structural patterns. This approach implies an assumption that these distinctions are meaningful and that different modality models would clearly favor specific 2D graph patterns. However, the performance differences among various modality methods in Table 6 are minor and do not reflect the significance of such a split.
3. The molecular image and video modalities are generated from 2D or 3D structures. It would be helpful to clarify why these modalities are important and which tasks specifically benefit from such artificial representations.
4. Why do 3D modality-based methods, such as Uni-Mol, outperform other modalities on MoleculeNet tasks? Are there any insights or reasons behind this?

**Questions:**

See weakness

---

> ### Comment · Reviewer_LcfM · 2024-11-24
>
> Dear authors:
>
> Thanks for the responses, some of my concerns have been resolved. However, I still find the properties presented lack practical significance, and the images or videos do not provide additional insights into the molecule itself. I will make a final decision after further discussion with the other reviewers.

---

### Official Review · Reviewer_7P4y · 2024-11-04

**Soundness:** 1
**Presentation:** 2
**Contribution:** 1
**Rating:** 3
**Confidence:** 4

**Summary:**

The paper introduces BenchMol, a comprehensive and multi-modality platform specifically designed for molecular representation learning. TThe authors introduce two novel datasets and corresponding benchmarks, namely MBANet and StructNet on newly defined MRL tasks.

**Strengths:**

The author conducts extensive experiments in evaluating property prediction performance of MRL methods on existing datasets.

The paper re-evaluate a large amount of existing molecular representation methods on moleculeNet.

**Weaknesses:**

1. The authors propose a multi-modality benchmarking platform, yet the study predominantly focuses on the performance comparison of single-modality molecular representation learning
methods and missing multimodal molecular representation learning methods (E.g. [1]), which is a critical weakness point considering the data scope as introduced in this paper.

2. The rationale for providing a multi-modality dataset that compares single modality MRL methods is not clear, given that existing packages such as RDKit and OpenBabel already facilitate the conversion between different modalities for a given molecule(E.g converting SMILES to 2D molecular graph). This raises questions about the contributions of the proposed benchmarks compared to readily available tools.

3. It’s better to demonstrate what kind of research gap in machine learning for chemistry this paper is trying to address. What certain type of chemistry questions is this paper trying to address, that may benefit the AI4Science community. For example, in section E, what specific chemistry problems does the atom distribution prediction task try to solve? How does a correction prediction of the atom distribution can benefit the chemistry community?

4. The provided link for accessing the datasets is currently non-functional, as attempts to access the OneDrive URL listed in the README file result in a 'This site can’t be reached' error. Therefore, I am not able to reproduce some of the experiments.

---

Minor concerns.

5. The presentation of Figures S3 (Pg. 21) is somewhat disorganized, notably, the font size on the x-axis of figure c and f is inconsistent with the rest.

6. The organization of the manuscript could be improved for better readability; specifically, the description of the molecular print method is positioned on Page 24, while other molecular MRL methods summarized on Page 6. In addition, it is better to put a reference or hyperlink of the MRL method within each table.

---

For improving this dataset and benchmark paper, [2] can be possibly considered as a reference.

[1] Wang Z, Jiang T, Wang J, et al. Multi-Modal Representation Learning for Molecular Property Prediction: Sequence, Graph, Geometry[J]. arXiv preprint arXiv:2401.03369, 2024.

[2] Velez-Arce A, Huang K, Li MM, Lin X, Gao W, Fu T, Kellis M, Pentelute BL, Zitnik M. TDC-2: Multimodal Foundation for Therapeutic Science. bioRxiv [Preprint]. 2024 Jun 21:2024.06.12.598655. doi: 10.1101/2024.06.12.598655. PMID: 38948789; PMCID: PMC11212894.

**Questions:**

My questions is listed in the Weakness part.

---

### Official Review · Reviewer_Sees · 2024-11-04

**Soundness:** 3
**Presentation:** 2
**Contribution:** 2
**Rating:** 5
**Confidence:** 4

**Summary:**

The paper introduces BenchMol, a comprehensive and unified platform for molecular representation learning (MRL). BenchMol integrates seven major molecular modalities (fingerprint, sequence, graph, geometry, image, geometry image, and video) and evaluates 23 mainstream MRL methods across various tasks. The authors propose two new benchmarks, MBANet and StructNet, to systematically assess the performance and preferences of different modalities. Through extensive experiments, the paper provides findings into the strengths and weaknesses of various modalities and their suitability for different molecular types and tasks.

The paper is in general interesting to read, while there are a few concerns that need to addressed.

**Strengths:**

- The paper gives a good survey for MRL methods using different modalities of molecular data.
- The paper proposes two benchmark datasets and tests various methods on them. Interesting conclusions are made according to the results.
- Given the numbers of experiments the paper have conducted, it is evident that the authors have put numerous efforts into this work, unifying code from different methods.

**Weaknesses:**

- The evaluations in the paper mainly focus on prediction accuracies. However, in many scenarios, such as virtual screening, computational cost is also very important. This is especially relevant in comparing methods using different modalities, while the paper completely ignores this aspect.
- The paper's presentation quality falls short of expectations. While most contents are still comprehensible, many minor issues exist in the current version of the paper. For example, in Figure 2(b), the word wrapping of “preproc/essing” in is strange; also, what is “combing”? For another example, the paper does not provide good definitions for image, geometry image and video in the MRL context. Minor issues like these truly affect the understanding of the paper.
- The findings in the experiments are interesting, but many of them are potentially superficial. They are often observations of the performance numbers, but fail to develop into more insightful discussions. For example, in section 5.3 “Fine-tuning of MBANet”, the paper mentions that models using the video modality significantly outperform those using other modalities. But *why* is that? The *findings* of this paper would be much more interesting if they can take one step further to develop into *insights*.
- The design of the benchmarks seems questionable. The dataset contains only 10,000 molecules, which is a small size considering the vast chemical space. In this case, the video modality seems to be advantageous because the video models can see more frames of the molecules. For fairer comparison, models using other modalities should be also able to access the same amount of molecular conformations.

**Questions:**

* To summarize the findings of the paper, could you give a concise conclusion on which model/modality to choose in MRL-related tasks?

---

### Official Review · Reviewer_KqA8 · 2024-11-06

**Soundness:** 1
**Presentation:** 2
**Contribution:** 1
**Rating:** 1
**Confidence:** 4

**Summary:**

The authors describe BenchMol, a benchmark for discriminative tasks in the molecular context with a focus on comparing methods for molecular representation learning.

**Strengths:**

The authors point out that benchmarking in molecular representation learning is littered with problems, such as unfair comparisons of methods arising from difference evaluation strategies (e.g. splitting differences) and the absence of a convenient unified benchmarking platform.

The authors perform a novel task in predicting basic attributes of molecules given their molecular representation of choice. It is surprising that this apparently simple task results in many failures, perhaps this aspect could be made a focus of the paper in the context of different molecular representations.

**Weaknesses:**

The authors have benchmarked methods for molecular representation, however true multi-modality comes from the underlying modality of the data, such as binary labels vs continuous labels vs 3D data vs dynamics data, and as such this benchmark is not a benchmark for multi-modal models in the way one would expect - the models themselves are single modality.

Since this benchmark is not evaluating multi-modal molecular algorithms, there is no specific need addressed by this new benchmark that isn't already serviced by existing molecular benchmarks, e.g. QM9, MoleculeNet, OGB, etc.

**Questions:**

1. The behaviour of different molecular representations is an important question, despite not being related to multi-modality. Would the authors consider an angle that examined the performance on the toy task described across molecular representations. Failures of representations to perform simple operations would be highly impactful.

---

### Official Review · Reviewer_CyMa · 2024-11-08

**Soundness:** 4
**Presentation:** 4
**Contribution:** 4
**Rating:** 10
**Confidence:** 5

**Summary:**

This study is a comprehensive examination of molecular representation learning (MRL) methods benchmarking, covering multiple molecular modalities (1D string representation, 1D fingerprints, 2D Graphs, 2D Images, 3D geometries, 3D geometry images, and video.

The study proposes 3 separate sets of datasets to evaluate all these modalities: MoleculeNet (pre-existing), MBANet (newly created), and StructNet (newly created). The first benchmark covers broad application in the biomolecular domain, the second benchmark evaluates the ability of MRL methods to capture basic molecular attributes, the third benchmark allows for discerning which MRL are more appropriate for which molecular types.

The study evaluates multiple MRL techniques and pre-trained models and draws 9 main insights from this extensive and large-scale examination, which I summarise as follows:

1. All modalities are useful for almost every task (models from 6 modalities are the top 6 in performance).
2. Ensembling multiple conformers can improve image-based MRLs.
3. Sequence-based models (transformers and similar architectures) perform well even when randomly initialised and without fine-tuning, which suggests that they have good inductive biases.
4. Geometry images and videos are resilient to different image formatting conventions (RGB vs BGR).
5. Video modality is the best for recovering basic molecular information (MBANet benchmark).
6. Pre-training models improve performance on recovering basic molecular information (MBANet), therefore, pre-training tasks are useful for this purpose.
7. Performance on MBANet within MRLs leveraging the same modality is similar
8. Modality determines whether the model is best performing at different types of molecules (StructNet benchmark).
9. Certain pre-trained models perform worse against certain molecular types than their randomly initalised counterparts. Therefore, certain pre-training tasks might be better suited for certain molecular types and will be detrimental for others.

Finally, the study presents tools for standarising multi-modal benchmarking on these datasets, provides splits for replicating results and utilities that are likely to accelerate multi-modal representation learning strategies.

**Strengths:**

Main strengths
---
1. The paper tackles a really important issue in molecular representation learning, provides a sound and comprehensive benchmark of existing methods, provides a clear way to compare the strengths and weaknesses of available MRL techniques and allows for a clear and fair comparison across modalities. Further they provide the utilities for reproducing their results easy to use. This research will significantly move the field forward and constitutes a strong baseline from which new multimodal research can build upon.
2. The paper is clearly written, results are concisely conveyed, the methodology is sound and detailed enough as to allow for the reproduction of the results.
3. Tables and data displayed clearly demonstrate and support the main claims and insights drawn by the authors.
4. Supplementary information is rich and comprehensive.

Minor strengths
----
_(Details of the paper that do not have a direct bearing on my evaluation, but I think are commendable)_

1. The insight regarding sequence-based models having enough inductive bias even when randomly initialised and with linear probing is highly interesting and could merit its further exploration.
2. The experiments with multiple conformations for the image modality are really interesting, the insights drawn highly informative, and they go beyond what the scope of the paper was to give a really comprehensive evaluation of the benefits and idiosyncrasies of different modalities.
3. Visual design in the Figures and Tables is crisp and facilitates the comprehension of the paper.

**Weaknesses:**

Main weakness
---
In Appendix B1, Figure S1, the histograms for the atom counts of certain atoms like Si (c), Br (f), P (g), S (h), Cl (i), B (j), Se (k), and particularly Ge (l), seem to be completely skewed and quite limited in the independent variable values (0, 1, 2). It seems that they'd be better suited for a classification task. I'd argue that the Ge task introduces only noise to the final metric as there is only one count of value 1, the rest are value 0.

Therefore, it will either be in training and will not be tested; or it will be in the testing and the model will not know that such a value is even possible. I see that the issue of transforming them into classification tasks would be that the the average of the classification and regression metrics would not make that much sense and this could be alleviated by using correlation metrics like Matthew's correlation coefficient (or a multi-class generalisation thereof) for classifcation, and Spearman's or Pearson's correlation coefficient for regression. Another alternative, probably simpler alternative, could be to remove the subtasks that are too skewed to be useful. I am not sure which option is best and I am open to the authors to explain their rationale for including them in their study. I think that, the limitation of this specific part of the benchmark should be at least acknowledged in the main text.

This is also applicable to Figure S2 - b.

This is the only major weakness in an otherwise excellent study.

**Questions:**

1. The experimental results with the randomly initialised sequence-based models are quite intriguing and seem a bit counterintuitive, particularly as it pertains the linear probing, do you have any further intuition of what may be the underlying mechanism that provides them with such a remarkable inductive bias. Have you seen any dependence between model size and model performance in this particular setting?
2. Some datasets can be quite dirty, a single molecule can be translated into multiple SMILES strings depending on the specific algorithm used, this leads to some datasets having the same molecule duplicated, but with different SMILES which makes it difficult to distinguish. Have you done any tests to detect these duplications (e.g., canonicalising the SMILES or using InChIKeys)?

---

> ### Comment · Reviewer_CyMa · 2024-11-25
>
> I thank the authors for their prompt response to my concerns. I particularly appreciate the follow up studies on the potential mechanism behind the inductive bias and the impact of model size on performance. Regarding the inductive bias, I personally am not familiar with the Davis-Bouldin Index and think that a brief introduction would be useful. Regarding the number of layers, the results in Table S49, I do not think that the conclusion that perfomance increases as number of layers increases is well supported as number of layers 4 is greater than 6, though I agree that there is an upwards trend, I would recommend softening the statement.
>
> I also disagree with the authors in the inclusion of the skewed distributions as 1) I don't think it shows meaningful learning to just repeat 0, a blind function that always outputs 0 would outperform anything in that setting, 2) the argument that regression allows for extrapolation I think is not supported by common wisdom in the field, the deeper a network is the more it will reproduce the data distribution it has trained on, if value 10 never appeared in its training set it is highly unlikely that it will know to extrapolate to it. In practice, I don't think that regression offers any significant advantage over classification with regards to extrapolation. I find the argument in favour of RMSE compelling.
>
> Overall, I stand by previous assessment of the quality of the paper. It is a strong work that fills a necessary gap in the literature by offering a comprehensive cross-modal evaluation of state-of-the-art models and provides substantial baselines upon which to build new solutions. The wrappers for the multiple modalities provide an easy interface to facilitate calculation, but I do not think they are significant technical contributions. The analyses of the multiple modalities and the effects on performance are through. The points I'm discussing with the authors are, in my opinion, minor and the authors are right that they do not have a significant impact on the validity of the results, I think they formal aspects that are open to different interpretations.
>
> Finally, I'd like to congratulate the authors for their outstanding working.

---

### Author Response · Authors · 2024-11-23
**General Response to All the Reviewers**

First, We sincerely thank all the reviewers for their insightful and constructive feedback on our manuscript. We are happy to hear that **our benchmark platform is comprehensive** ($\frac{4}{5}$ CyMa, Sees, 7P4y, LcfM), and **our experiments are interesting, insightful, and surprising** (reviewers: $\frac{4}{5}$ CyMa, Sees, LcfM, KqA8). In addition, They think **our paper is well-written and easy to understand** ($\frac{2}{5}$ CyMa, LcfM).

Second, we have substantially revised the paper and all revised content are highlighted in red. We summarize the primary revisions made to the paper:

- **In-depth Analysis of Insights:** We conducted extensive and substantial in-depth analysis of the presented insights, which are included in **Appendices K.1-K.8** to provide further insights.
- **Clarification of Multi-Modality:** Our multi-modality only describes that the number of modalities supported by BenchMol is large and the goal of the study is to study the differences between different modalities. Since the word "multi-modal" can easily lead to misunderstandings about the evaluation of multi-modal methods, we have revised a lot of multi-modal vocabulary in the paper;
- **Title Correction:** We changed the title to "Benchmarking Single-Modal Molecular Representations Across Diverse Modalities" to avoid misunderstanding of the paper.
- **Clarity of Visual Modality:** We rewrote the definition of visual modality in Section 3 and added more details of visual rendering and visual importance in **Appendix B and Appendix C.3** respectively.
- **Motivation and Potential Impact of BenchMol:** We discuss it in **Appendix C.1**.
- **Motivation, Practicality and Limitation of Benchmarks:** We discuss it in **Appendix C.2 and D.1**.
- **Extension of MBANet Scale:** We expand the data scale of MBANet in the experiment to reflect the robustness of the conclusion.
- **Computational Cost:** Additional information on model size, training cost, and inference speed has been included in **Appendix J**, providing more insight into the practicality and efficiency of different approaches.
- **Presentation Corrections:** We addressed disorganized presentation to improve overall readability and clarity.

Third, we have carefully addressed all the reviewers' comments and provided detailed responses to each point. The revised manuscript has been uploaded. If there are any further questions or concerns, please don’t hesitate to reach out. We remain committed to improving the quality of this work and welcome further discussions.

Thank you once again for your valuable feedback and support!

---

### Note · Authors · 2025-01-19

**Comment:**

We sincerely thank all the reviewers for spending their valuable time reading our paper and providing constructive feedback. We appreciate the insights shared, which we believe will help strengthen the core aspects of the paper. We also thank the reviewers for their positive comments on this paper, which we find encouraging. We are committed to carefully considering these comments and making necessary improvements to the presentation of the paper and the key research proposed.

After careful consideration, we have decided to withdraw this paper from consideration. Thanks again to the reviewers and the Associate Editor for their time and feedback.

**Withdrawal Confirmation:**

I have read and agree with the venue's withdrawal policy on behalf of myself and my co-authors.